# BIDIRECTIONAL COMMUNICATION-EFFICIENT NON-CONVEX ADAPTIVE FEDERATED LEARNING

## ABSTRACT

Within the framework of federated learning, we introduce two novel strategies: New Lazy Aggregation (NLA) and Accelerated Aggregation (AA). The NLA strategy reduces communication and computational costs through adaptive gradient skipping, while the AA strategy accelerates computation and decreases communication costs via adaptive gradient accumulation. Building upon these innovative strategies and compression techniques, we propose two new algorithms: FedBN-LACA and FedBACA, aimed at minimizing bidirectional communication costs. We provide theoretical guarantees for client participation (either full or partial) in these algorithms under non-convex settings and heterogeneous data. In the context of non-convex optimization with full client participation, our proposed FedBN-LACA and FedBACA algorithms achieve the same convergence rate of $\mathcal{O}(1/T)$ as their non-tight counterparts. Extensive experimental results demonstrate that our protocols facilitate effective training in non-convex environments and exhibit robustness across a wide range of devices, partial participation, and imbalanced data.

## 1 INTRODUCTION

Federated Learning (FL) is a promising machine learning (ML) framework that enables collaborative model training without sharing data. This approach ensures that data privacy is protected, as data does not need to leave the owner's possession McMahan et al. (2017). FL models involve edge clients or devices, such as smartphones and personal computers (PCs), training local ML models using their own data without sharing. Instead of sending raw data to a central server (e.g. cloud server), the edge client updates the parameters of the local server model (e.g. smartphone, PC updates). The server then generates a shared global ML model by aggregating local parameter updates. In FL, traditional distributed stochastic gradient descent (SGD) is unsuitable for challenging federated learning scenarios, where data does not follow independent homogeneous distributions and only a small fraction of clients participate in each communication round. To address this issue, many federated optimization methods use local client updates. To reduce significantly the amount of communication required to train the model, local clients update their models multiple times before communicating with the server. One of the most popular FL methods is the FedAvg algorithm proposed by McMahan et al. (2017), in which the global model is updated by averaging over multiple local SGD update steps.

Although FedAvg algorithm can be trained without sharing data and achieve good results, it in practice still presents challenges in FL. 1) Lack of adaptivity. As SGD-based updates may not be suitable for stochastic gradient noise with heavy-tailed distributions, which usually occur when training large models Devlin et al. (2019); Brown et al. (2020); 2) Unaffordable communication. Repeated synchronization of the uplink and downlink between the client and the server leads to significant communication overhead, but some of these parameters are passed non-essential. Here, the uplink represents a transmission from clients to servers and the downlink represents a transmission from server to clients.

Many works have researched the aforementioned issues. 1) For adaptivity in the federated learning framework, Reddi et al. (2020) proposed the FedAdam algorithm and its variants Tong et al. (2020); Wang et al. (2022a); Wu et al. (2023), which integrate the adaptive gradient approach. 2) Recently, to reduce communication costs, three approaches have been developed: (i) Local approach Huang

et al. (2023); Li & Wang (2019); Li et al. (2019b); Mishchenko et al. (2022); Karimireddy et al. (2019a). The local devices implement multiple rounds of optimization before sending the information to the server. This reduces the rounds of global communication, thereby having an overall lower communication cost. (ii) Compression methods Reisizadeh et al. (2020); Chen et al. (2021); Richtárik et al. (2023); Beznosikov et al. (2023). During each round of communication, the local devices send compressed information to the server, reducing communication costs through some compression mechanisms. (iii) Lazy aggregation algorithmChen et al. (2018b); Sun et al. (2019); Ghadikolaei et al. (2021); Mishchenko et al. (2022). Some parameters may be similar to those of the previous round, so they do not need to be transmitted (i.e., the parameters of the previous round are used in the current round), which also effectively reduces the communication cost.

However, although the lazy aggregation algorithm provides a strategy to reduce communication costs, it is more difficult to implement in practice, especially in joint learning where only some clients are involved in the training. Although these algorithms have achieved great success in reducing communication costs, they are unidirectional in the sense that they are uplink communication, not downlink communication. A natural thought is: **How do we design a simple and more efficient strategy to improve communication efficiency based on bidirectional communication?** In this paper, we initially introduce two innovative updated strategies, namely NLA and AA. These strategies are subsequently applied to the FedAMS and FedCAMS algorithms as outlined in Wang et al. (2022a). Furthermore, we advance the development of bidirectional communication-efficient adaptive algorithms within a non-convex framework and in the context of heterogeneous data.

**Main contributions**

• We present two novel communication strategies for joint learning in federated learning (FL), referred to as NLA and AA, which improve upon the existing LAG algorithmChen et al. (2018b); Sun et al. (2019); Mishchenko et al. (2022). In contrast to conventional approaches, NLA and AA do not require parameters from multiple iterations; rather, they utilize only the parameters from the preceding iteration along with those from the current iteration. This characteristic enhances their adaptability and simplifies their implementation.

• We propose two novel and efficient adaptive optimization techniques for communication: FedNLACA and FedACA . Both methods enhance communication efficiency and adaptability within the context of joint learning. The FedNLACA algorithm notably decreases communication costs by employing real error feedback, a compression strategy, and our proposed NLA strategy, all while preserving high accuracy. Similarly, the FedACA algorithm effectively reduces communication costs and sustains high accuracy through the use of real error feedback, a compression strategy, and our proposed AA strategy. It is important to highlight that FedNLACA and FedACA attains a convergence rate of $\mathcal{O}(1/T)$.

• We have developed a novel cross-device compatible adaptive joint optimization method, referred to as FedBNLACA and FedBACA, which employs two strategic approaches to facilitate a reduction in bidirectional communication costs for both uplink and downlink transmissions. A convergence analysis was performed under conditions of general non-convexity and data heterogeneity, demonstrating that both FedBNLACA and FedBACA can also attain a convergence rate of $\mathcal{O}(1/T)$.

• Extensive experiments on various benchmarks showed that our proposed algorithms are well adaptive in training real-world machine learning models.

## 2 RELATED WORK

**Adaptive Gradient Methods:** Adaptive gradients (Zeiler, 2012; Duchi et al., 2011; Kingma & Ba, 2014; Reddi et al., 2018) are a series of algorithms that effectively reduce the relatively slow convergence and over-sensitivity to parameters of gradient descent methods in the face of heavy-tailed stochastic gradients, and are heavily used to train large networks.

**Federated Learning:** FedAvg was introduced by McMahan et al. (2017) as the inaugural algorithm for federated learning (FL). By employing periodic model averaging, this approach significantly mitigates communication overheads. Initial studies focused on the analysis of FL algorithms within a homogeneous data framework, while more recent investigations have expanded the scope of federated learning to encompass heterogeneous data environments (non-iid) and non-convex models (Li &

Wang, 2019; Li et al., 2019a; Sahu et al., 2018; Yang et al., 2019; Wang et al., 2022a)). (Li et al., 2019a; Sahu et al., 2018) proposed FedProx and FedDANE algorithms for federated optimization against heterogeneity. Wang et al. (2022a)gave theoretical results of data heterogeneous federated learning under the conditions of adaptive methods. In this paper, we follow the ideas of Wang et al. (2022a). Numerous studies have been conducted that build upon the FedAvg framework, including notable contributions such as FedNova Wang et al. (2020), and SCAFFOLD Karimireddy et al. (2019a), as well as various other adaptations of FedAvg (Yang et al., 2021b; Wang et al., 2022a). In a recent advancement, Reddi et al. (2020) introduced several adaptive federated optimization techniques, including FedAdagrad, FedYogi, and FedAdam, aimed at addressing the convergence challenges associated with FedAvg. Additionally, Chen et al. (2020) presented Local AMSGrad, while Tong et al. (2020) introduced a suite of federated adaptive gradient methods that incorporate calibration mechanisms.

**Effective methods of communication:** Many methods to reduce communication costs have been proposed in federated learning, three main ideas are 1) multiple rounds of local iteration methods (Elgabli et al., 2022; McMahan et al., 2017; Li et al., 2019a; Sahu et al., 2018), 2) compression and error feedback (Reisizadeh et al., 2020; Elgabli et al., 2022; Haddadpour et al., 2020; Wang et al., 2022a), 3) lazy aggregation (Sun et al., 2020; Mao et al., 2022; Chen et al., 2018b; Sun et al., 2019). Most of the above methods are only unidirectional (uplink), and although (Sun et al., 2020; Wang et al., 2022b) considered bidirectional algorithms, they did not consider the heterogeneity problem in FL, the involvement of some of the servers in the training as well as the related theoretical guarantee, and the adaptive problem. Moreover, their strategies to reduce the communication cost are difficult to implement under heterogeneous partial clients participation. Therefore, how to give simple strategies to reduce communication cost, and how to design adaptively efficient algorithms for bidirectional communication with partial clients participation is the focus of this paper.

**Notation:** For vectors $\mathbf{x}, \mathbf{y} \in \mathbb{R}^d$, $\sqrt{\mathbf{x}}, \mathbf{x}^2, \mathbf{x}/\mathbf{y}$ denote the $n$ element-wise square root, square, and division of the vectors. For vector $\mathbf{x}$ and matrix $A$, $\|\cdot\|$ denotes the $\ell_2$ norm of vector/matrix, i.e., $\|\mathbf{x}\| = \|\mathbf{x}\|_2$ and $\|A\| = \|A\|_2$. In algorithms, $t$ denotes the $t$-th iteration. The $m$ is the number of all clients/devices.

## 3 PRELIMINARY

This paper aims to study the federated learning non-convex optimization problem, which is formulated as follows:

$$\min_{\theta \in \mathbb{R}^d} f(\theta) = \frac{1}{m} \sum_{i=1}^{m} F_i(\theta),$$

where $F_i(\theta) = \mathbb{E}_{\xi_i \sim \mathcal{D}_i} F_i(\theta, \xi_i)$, $F_i(\theta)$ denotes the local non-convex loss, $\mathcal{D}_i$ represents the data distribution on $i$ clients. $m$ represents the number of all clients, $d$ denotes the dimension of the model parameters. In the non i.i.d setting, distributions $\mathcal{D}_i$ and $\mathcal{D}_j$ can vary from each other, i.e., $\mathcal{D}_i \neq \mathcal{D}_j$, $\forall i \neq j$. In the stochastic setting, one can only get unbiased estimates of $F_i(\theta)$, i.e., the stochastic gradient $\mathbf{g}_t^i = \nabla F_i(\theta, \xi_i)$.

**Assumption 3.1.** (Smoothness). There exists an $L$ such that each loss function on the $i$-th worker $F_i(\theta)$ satisfies the following equation, $\forall \mathbf{x}, \mathbf{y} \in \mathbb{R}^d$,

$$|F_i(\mathbf{x}) - F_i(\mathbf{y}) - \langle \nabla F_i(\mathbf{y}), \mathbf{x} - \mathbf{y} \rangle| \leq \frac{L}{2} \|\mathbf{x} - \mathbf{y}\|^2.$$

The smoothness of $F_i$ also means that the $L$-gradient Lipschitz condition, i.e., $\|\nabla F_i(\mathbf{x}) - \nabla F_i(\mathbf{y})\| \leq L\|\mathbf{x} - \mathbf{y}\|$. This assumption is widely used Yang et al. (2021a); Reddi et al. (2018).

**Assumption 3.2.** (Bounded Gradient). Each loss function on the $i$-th worker $F_i(\theta)$ has $G$-bounded stochastic gradient on $\ell_2$, i.e., for all $\xi$, $E\|\nabla f_i(\theta, \xi)\| \leq G$. In addition, we also assume that $\|\theta\| \leq H$. The assumption of bounded gradient is usually adopted in adaptive gradient methods Reddi et al. (2018); Chen et al. (2018a).

**Assumption 3.3.** (Bounded Variance). The bounded local variance, i.e. for all $\theta$, $i \in [m]$, $\mathbb{E}[\|\nabla f_i(\theta, \xi) - \nabla F_i(\theta)\|^2] \leq \sigma_l^2$; and global variance constraint, i.e. $\frac{1}{m} \sum_{i=1}^{m} \|\nabla F_i(\theta) - \nabla f(\theta)\|^2 \leq \sigma_g^2$, where $\sigma_l^2$ and $\sigma_g^2$ are some positive constants.

The assumption of bounded variance also is usually adopted in adaptive gradient methods Yang et al. (2021a); Reddi et al. (2020). The bounded local variance represents the randomness of stochastic gradients, while the bounded global variance represents the heterogeneity of data between clients. It is important to note that these variances are bounded. The value of $\sigma_g = 0$ indicates the i.i.d setting, where datasets from each client have the same distribution.

**Assumption 3.4.** (Biased Compressor). Consider a biased operator $\mathcal{C} : \mathbb{R}^d \to \mathbb{R}^d$ : for $\forall \theta \in \mathbb{R}^d$, there exists constant $0 \leq q \leq 1$ such that

$$\|\mathcal{C}(\theta) - \theta\| \leq q\|\theta\|, \forall \theta \in \mathbb{R}^d.$$

Note that $q = 0$ means no compression to $\theta$. Here are two examples: scaled-sign compressor and top-$k$ compressor.

**Top-$k$** Shi et al. (2019); Stich et al. (2018): For $1 \leq k \leq d$ and $\forall \theta \in \mathbb{R}^d$, the coordinate of $\theta$ is ordered by the magnitude $|\theta_{(1)}| \leq |\theta_{(2)}| \leq \cdots \leq |\theta_{(d)}|$. Denote $h_1, h_2, ..., h_d$ as standard unit basis vectors in $\mathbb{R}^d$. The compressor $\mathcal{C}_{\text{top}} : \mathbb{R}^d \to \mathbb{R}^d$ is defined as: $\mathcal{C}_{\text{top}}(\theta) = \sum_{i=d-k+1}^{d} \theta_{(i)} h_{(i)}$.

Define the compression ratio as $r = k/d$. It can be shown that $\|C_{\text{top}}(\theta) - \theta\|^2 \leq (1 - k/d)\|\theta\|^2 = (1 - r)\|\theta\|^2$, and thus we have $q = \sqrt{1 - r}$.

**Scaled sign** Karimireddy et al. (2019b): For $1 \leq k \leq d$ and $\forall \theta \in \mathbb{R}^d$, the compressor $\mathcal{C}_{\text{sign}} : \mathbb{R}^d \to \mathbb{R}^d$ is defined as

$$\mathcal{C}_{\text{sign}}(\theta) = \|\theta\|_1 \cdot \text{sign}(\theta)/d.$$

For scaled sign compressor, when $\|C_{\text{sign}}(\theta) - \theta\|^2 = (1 - \|\theta\|_1^2/d\|\theta\|^2)\|\theta\|^2$, thus $q = \sqrt{1 - \|\theta\|_1^2/d\|\theta\|^2}$.

### 3.1 Two Strategies

Prior to the presentation of my strategy, it is essential to examine the LAG algorithmChen et al. (2018b); Sun et al. (2019); Mishchenko et al. (2022):

$$\left\|\nabla F_m(\boldsymbol{\theta}_m^{t-1}) - \nabla F_m(\boldsymbol{\theta}^t)\right\|^2 \leq \frac{1}{\alpha^2 m^2} \sum_{r=1}^{R} \xi_R \left\|\boldsymbol{\theta}^{t+1-R} - \boldsymbol{\theta}^{t-R}\right\|^2, \tag{1}$$

$$L_m^2 \left\|\boldsymbol{\theta}_m^{t-1} - \boldsymbol{\theta}^t\right\|^2 \leq \frac{1}{\alpha^2 m^2} \sum_{r=1}^{R} \xi_R \left\|\boldsymbol{\theta}^{t+1-R} - \boldsymbol{\theta}^{t-R}\right\|^2. \tag{2}$$

While the aforementioned concept is innovative, it necessitates the establishment of parameters for the initial R iterations. Determining the appropriate value for R poses a challenge, and furthermore, the inclusion of parameters from all R iterations in the computation may lead to complications in data storage. Therefore, we propose two new aggregation strategies.

Here we introduce the meanings of some parameter representations. The $S_t$ denotes the sum of all participating training clients at the $t$-th iteration, $C, D, \alpha$ are some postive constants, $m$ represents the number of all clients, here we first introduce the meanings of some parameter representations, R represents the number of iterations selected, $L_m$ represents the $L$-gradient Lipschitz condition of $m$.

**NLA Strategy** (**N**ew **L**azy **A**ggregation). For any $\mathbf{x}$, let $\rho_t = \mathbf{x}_t - \mathbf{x}_{t-1}$. If

$$\|\rho_t\| \leq \frac{C}{\alpha S_t}\|\mathbf{x}_{t-1}\| : \mathbf{x}_t \leftarrow \mathbf{x}_{t-1}, else : \mathbf{x}_t \leftarrow \mathbf{x}_t. \tag{3}$$

**Example 1:** Let $q_t^i = \Delta_t^i - \Delta_{t-1}^i$, if $\|q_t^i\| \leq \frac{C}{\alpha S_t}\|\Delta_{t-1}^i\|, i \in M_t : \tilde{\Delta}_t^i \leftarrow \Delta_{t-1}^i$, else: $\tilde{\Delta}_t^i \leftarrow \Delta_t^i$,the specific parameters are given in Algorithm 1.

**Example 2:** Let $\mathcal{C}(q_t^i) = \widehat{\Delta}_t^i - \widehat{\Delta}_{t-1}^i$, if $\|\mathcal{C}(q_t^i)\| \leq \frac{C}{\alpha S_t}\|\widehat{\Delta}_{t-1}^i\|, i \in M_t : \widehat{\widetilde{\Delta}}_t^i \leftarrow \widehat{\Delta}_{t-1}^i$, else: $\widehat{\widetilde{\Delta}}_t^i \leftarrow \widehat{\Delta}_t^i$,the specific parameters are given in Algorithm 3.

**Example 3:** Let $\mathcal{C}(Q_t^i) = \widehat{\theta}_t^i - \widehat{\theta}_{t-1}^i$, if $\|\mathcal{C}(Q_t^i)\| \leq \frac{C}{\alpha S_t}\|\widehat{\theta}_{t-1}^i\|, i \in M_t : \widehat{\widetilde{\theta}}_t^i \leftarrow \widehat{\theta}_{t-1}^i$, else: $\widehat{\widetilde{\theta}}_t^i \leftarrow \widehat{\theta}_t^i$,the specific parameters are given in Algorithm 2.

*Remark* 3.1. The NLA Strategy presented herein represents a modification of the lazy aggregation method as described in previous works Chen et al. (2018b); Sun et al. (2019). This approach is characterized by its operational simplicity, necessitating only a comparison with the parameters from the preceding iteration. As an innovative lazy aggregation technique, the NLA Strategy effectively diminishes communication costs by minimizing the number of communication parameters required. (The NLA strategy assesses whether the difference between the parameters from the $(t-1)$th and $t$-th iterations falls within a minimal threshold, indicating that these parameters are closely aligned. If this proximity is confirmed, the parameter from the $(t-1)$th iteration is retained in place of the parameter from the $t$-th iteration.)

**AA Strategy** (**A**ccelerated **A**ggregation). For any $\mathbf{x}$, let $\rho_t = \mathbf{x}_t - \mathbf{x}_{t-1}$. if

$$\|\rho_t\| \leq \frac{D}{\alpha S_t}\|\mathbf{x}_{t-1}\| : \mathbf{x}_t \leftarrow \mathbf{x}_{t-1} + \mathbf{x}_t, else : \mathbf{x}_t \leftarrow \mathbf{x}_t. \tag{4}$$

**Example 4:** Let $q_t^i = \Delta_t^i - \Delta_{t-1}^i$, if $\|q_t^i\| \leq \frac{D}{\alpha S_t}\|\Delta_{t-1}^i\|, i \in M_t : \tilde{\Delta}_t^i \leftarrow \Delta_{t-1}^i + \Delta_t^i$, else: $\tilde{\Delta}_t^i \leftarrow \Delta_t^i$, the specific parameters are given in Algorithm 1

**Example 5:** Let $\mathcal{C}(q_t^i) = \widehat{\Delta}_t^i - \widehat{\Delta}_{t-1}^i$, if $\|\mathcal{C}(q_t^i)\| \leq \frac{D}{\alpha S_t}\|\widehat{\Delta}_{t-1}^i\|, i \in M_t : \widehat{\tilde{\Delta}}_t^i \leftarrow \widehat{\Delta}_{t-1}^i + \widehat{\Delta}_t^i$, else: $\widehat{\tilde{\Delta}}_t^i \leftarrow \widehat{\Delta}_t^i$, the specific parameters are given in Algorithm 3.

**Example 6:** Let $\mathcal{C}(Q_t^i) = \widehat{\theta}_t^i - \widehat{\theta}_{t-1}^i$, if $\|\mathcal{C}(Q_t^i)\| \leq \frac{D}{\alpha S_t}\|\widehat{\theta}_{t-1}^i\|, i \in M_t : \widehat{\tilde{\theta}}_t^i \leftarrow \widehat{\theta}_{t-1}^i + \widehat{\theta}_t^i$, else: $\widehat{\tilde{\theta}}_t^i \leftarrow \widehat{\theta}_t^i$, the specific parameters are given in Algorithm 2.

*Remark* 3.2. The AA strategy is a novel acceleration method designed to reduce communication costs by speeding up the process. The proposal of this strategy is based on a fundamental motivation: when the parameters of the $(t-1)$th iteration are very close to the parameters of the $t$th iteration, it is possible to achieve an update of both steps at once (i.e., by adding the parameters of the $(t-1)$th and $t$th iterations). This core idea is consistent with the principles of the NLA algorithm. Through adaptive accelerated iterative descent, the AA strategy can effectively reduce communication costs. It is worth noting that even without the AA strategy, the iterative process of conventional algorithms can still achieve results similar to those of the AA strategy, but a more in-depth analysis of the AA strategy will be reserved for future research.

## 4 METHODS

### 4.1 FEDERATED NEW LAZY AGGREGATION AMSGRAD AND FEDERATED ACCELERATION AMSGRAD

In this section, we present two new frameworks of the adaptive algorithm: **Fed**erated **N**ew **L**azy **A**ggregation **AMSG**rad (FedNLAA) and **Fed**erated **A**ccelerated **AMSG**rad (FedAA). In FedNLAA and FedAA algorithms. $\theta_t$ is the $t$-th iteration $t$ of the global model parameters. At iteration $t$, the participating client $i$ in the selected subset $S_t$ (of size $n$) receives the model $\theta_t$ from the server, i.e., $\theta_{t,0}^i = \theta_t$. Then, the client performs $K$ steps of local SGD updating with local learning rate $\eta_l$ to get the local model $\theta_{t,K}^i$, judges whether the client $i$ model difference $\Delta_t^i = \theta_{t,K}^i - \theta_t$ satisfies the strategy NLA (in FedNLAA) or AA (in FedAA), and then send the judged model difference $\widehat{\Delta}_t^i$ to the server. The server updates the global model difference $\Delta_t$ by simply averaging the local model differences $\widehat{\Delta}_t^i$. Algorithm 1 gives the detailed procedure for FedNLAA and FedAA.

### 4.2 CONVERGENCE ANALYSIS FOR FEDNLAA AND FEDAA

**Full Participation:** All clients participate in training, i.e., $|\mathcal{S}_t| = m, \forall t \in [t]$.

**Theorem 4.1.** *Under Assumptions 3.1-3.3, if learning rate $\eta_l$ satisfies the following condition: $\eta_l \leq$*

$$\min\left\{\frac{1}{8KL}, \frac{\epsilon}{K\sqrt{\beta_2 K^2 G^2 + \epsilon}[(3+C_1^2)\eta L + 2\sqrt{2(1-\beta_2)}G]}\right\}, \textit{then FedLAA in Algorithm 1 under the full}$$

---

**Algorithm 1** FedNLAA and FedAA

---

**Input:** Initial value $\theta_1$, local step size $\eta_l$, global step size $\eta$, constants $\beta_1$, $\beta_2$ and $\epsilon$, $\Delta_0^i = 0$

1: $\mathbf{m}_0 \leftarrow 0, \mathbf{v}_0 \leftarrow 0$
2: **for** $t = 1$ **to** $T$ **do**
3:      Server randomly selects a subset of clients $S_t$ and transmits $\theta_t$ to the subset of clients $S_t$.
4:      $\theta_{t,0}^i \leftarrow \theta_t$
5:      **for** each client $i \in S_t$ in parallel **do**
6:         **for** $k = 0, ..., K-1$ **do**
7:            Compute local stochastic gradient:
             $\mathbf{g}_{t,k}^i = \nabla F_i(\theta_{t,k}^i; \xi_{t,k}^i)$,
8:            Update $\theta_{t,k+1}^i = \theta_{t,k}^i - \eta_l \mathbf{g}_{t,k}^i$.
9:         **end for**
10:         Compute $\Delta_t^i = \theta_{t,K}^i - \theta_t$ , $q_t^i = \Delta_t^i - \Delta_{t-1}^i$,
11:         Judges: If $q_t^i$ satisfies NLA (Example 1) or AA (Example 4).
12:         Outputs: $\tilde{\Delta}_t^i$ .
13:      **end for**
14:      Server aggregates: $\tilde{\Delta}_t = \frac{1}{|S_t|} \sum_{i \in S_t} \tilde{\Delta}_t^i$,
15:      Update: $\mathbf{m}_t = \beta_1 \mathbf{m}_{t-1} + (1 - \beta_1)\tilde{\Delta}_t$,
16:      Update: $\mathbf{v}_t = \beta_2 \mathbf{v}_{t-1} + (1 - \beta_2)\tilde{\Delta}_t^2$,
       $[\widehat{\mathbf{v}}_t = \max(\widehat{\mathbf{v}}_{t-1}, \mathbf{v}_t, \epsilon)$ and $\theta_{t+1} = \theta_t + \eta\frac{\mathbf{m}}{\sqrt{\mathbf{v}_t}}]$,
       or
       $[\widehat{\mathbf{v}}_t = \max(\widehat{\mathbf{v}}_{t-1}, \mathbf{v}_t)$ and $\theta_{t+1} = \theta_t + \eta\frac{\mathbf{m}}{\sqrt{\mathbf{v}_t}+\epsilon}]$.
17: **end for**

---

*participation has*

$$\min \mathbb{E}[\|\nabla f(\theta_t)\|^2] \leq 4\sqrt{\beta_2\eta_l^2 K^2 G^2 + \epsilon} \cdot \left[\frac{f_0 - f_*}{\eta\eta_l KT} + \frac{\Xi}{T} + \Omega\right],$$

*where* $\Xi = \frac{C_1 G^2 d}{\sqrt{\epsilon}} + \frac{2C_1^2\eta\eta_l KLG^2 d}{\epsilon}, \Omega = \frac{5\eta_l^2 K^2 L^2}{\sqrt{2\epsilon}}(\sigma_l^2 + 6K\sigma_g^2) + (3 + C_1^2)\eta^2 L + 2\sqrt{2(1-\beta_2)}\eta G](\frac{2\eta_l}{m\eta\epsilon}\sigma_l^2 + \frac{2KC^2\eta_l G^2}{\alpha^2\eta m^2\epsilon}) + \frac{\sqrt{2}GC}{\alpha m\epsilon}$, *and* $C_1 = \frac{\beta_1}{1-\beta_1}$.

**Theorem 4.2.** *Under Assumptions 3.1-3.3, if learning rate $\eta_l$ satisfies the following condition:* $\eta_l \leq \min\left\{\frac{1}{8KL}, \frac{\epsilon}{K\sqrt{\beta_2 K^2 G^2 + \epsilon}[(3+C_1^2)\eta L + 2\sqrt{2(1-\beta_2)}G]}\right\}$, *then FedAA in Algorithm 1 under the full participation has*

$$\min \mathbb{E}[\|\nabla f(\theta_t)\|^2] \leq 4\sqrt{\beta_2\eta_l^2 K^2 G^2 + \epsilon} \cdot \left[\frac{f_0 - f_*}{\eta\eta_l KT} + \frac{\Xi}{T} + \Omega\right],$$

*where* $\Xi = \frac{C_1 G^2 d}{\sqrt{\epsilon}} + \frac{2C_1^2\eta\eta_l KLG^2 d}{\epsilon}, \Omega = \frac{5\eta_l^2 K^2 L^2}{\sqrt{2\epsilon}}(\sigma_l^2 + 6K\sigma_g^2) + (3 + C_1^2)\eta^2 L + 2\sqrt{2(1-\beta_2)}\eta G](\frac{2\eta_l}{m\eta\epsilon}\sigma_l^2 + \frac{2K\eta_l G^2}{\eta\epsilon}) + \frac{\sqrt{2}G}{\epsilon}$, *and* $C_1 = \frac{\beta_1}{1-\beta_1}$.

*Remark* 4.1. When the parameters $C = D, \frac{C}{\alpha m} = 1$, the result of Theorem 4.1 becomes the result of Theorem 4.2. The upper bound for $\min_{t\in[T]}\mathbb{E}[\|\nabla f(\theta_t)\|^2]$ contains three parts: The first two terms decrease as $T$ increases, and this term tends to zero as $t$ tends to infinity. The last term relates to the local stochastic variance $\sigma_l$ and global variance $\sigma_g$. In the i.i.d setting, where the global variance is zero and each worker has the same data distribution, i.e., $\sigma_g = 0$, the variance term $\Omega$ will be smaller.

**Corollary 4.1.** *Suppose choose local learning rate $\eta_l = \Theta(\frac{1}{\sqrt{TK}})$ and global learning rate $\eta = \Theta(\sqrt{Km})$, when $T$ is sufficiently large, i.e.,$T \geq Km$, the convergence rate for FedNLAA and FedAA in Algorithm 2 under full participation has*

$$\min_{t\in[T]} \mathbb{E}[\|\nabla f(\theta_t)\|^2] = \mathcal{O}\left(\frac{1}{\sqrt{TKm}}\right).$$

*Remark* 4.2. Corollary 4.1 suggests that with sufficient large $T$, when $T = \mathcal{O}(Km)$, FedNLAA and FedAA achieve a convergence rate of $\mathcal{O}(\frac{1}{T})$, which matches the result for general federated non-convex optimization methods such as FedAMS Wang et al. (2022a) and FedAdam Reddi et al. (2020).

**Partial Participation:** We assume that only $n$ of $m$ workers participate the local updating and communicate with the central server on each step $t$, i.e., $|S_t| = n, \forall t \in [1, T]$. The partial participation includes the randomness of sampling, and the coefficient varies for different sampling methods. Here we consider the random sampling without replacement. At the $t$-th iteration, we randomly sample a subset $S_t$ contains $n$ workers for local updating, for any two workers $i, j \in S_t$, the probability of being sampled to participate in the model update are $\mathbb{P}\{i \in \mathcal{S}_t\} = \frac{n}{m}$ and $\mathbb{P}\{i, j \in \mathcal{S}_t\} = \frac{n(n-1)}{m(m-1)}$.

**Theorem 4.3.** *Under Assumptions 3.1-3.3, if $\eta_l$ satisfies: $\eta_l \quad \leq \quad \min\big\{\frac{1}{8KL},$*

$$\frac{n(m-1)\epsilon}{48m(n-1)K\sqrt{\beta_2 K^2 G^2 + \epsilon} \cdot [3\eta L + C_1^2 \eta L + 2\sqrt{2(1-\beta_2)}G]}\bigg\}, \text{ then FedLAA in Algorithm 1 under partial}$$

*participation has*

$$\min \mathbb{E}[\|\nabla f(\theta_t)\|^2] \leq 8\sqrt{\beta_2 \eta_l^2 K^2 G^2 + \epsilon}\left[\frac{f_0 - f_*}{\eta \eta_l KT} + \frac{\Xi}{T} + \Omega\right],$$

*where* $\Xi = \frac{C_1 G^2 d}{\sqrt{\epsilon}} + \frac{2G_1^2 \eta \eta KLG^2 d}{\epsilon}$, $\Omega = \frac{5\eta^2 KL^2}{\sqrt{2}\epsilon}(\sigma_l^2 + 6K\sigma_q^2) + [(3 + C_l^2)\eta L + 2\sqrt{2(1-\beta_2)}G](\frac{\eta_l}{n\eta\epsilon}\sigma_l^2 + \frac{2\eta_l C^2 K^2 G^2}{\alpha^2 \eta n^2 \epsilon}) + [(3+C_1^2)\eta L + 2\sqrt{2(1-\beta_2)}G]\frac{\eta_l(m-n)}{2n(m-1)\epsilon}[15K^2 L^2 \eta_l^2(\sigma_l^2 + 6K\sigma_g^2) + 3K\sigma_g^2]\frac{1}{\eta\eta_l K} + \frac{\sqrt{2}GC}{\alpha n\epsilon}$ *and* $C_1 = \frac{\beta_1}{1-\beta_1}$.

**Theorem 4.4.** *Under Assumptions 3.1-3.3, if $\eta_l$ satisfies: $\eta_l \quad \leq \quad \min\big\{\frac{1}{8KL},$*

$$\frac{n(m-1)\epsilon}{48m(n-1)K\sqrt{\beta_2 K^2 G^2 + \epsilon} \cdot [3\eta L + C_1^2 \eta L + 2\sqrt{2(1-\beta_2)}G]}\bigg\}, \text{ then FedAA in Algorithm 1 under partial}$$

*participation has*

$$\min \mathbb{E}[\|\nabla f(\theta_t)\|^2] \leq 8\sqrt{\beta_2 \eta_l^2 K^2 G^2 + \epsilon}\left[\frac{f_0 - f_*}{\eta \eta_l KT} + \frac{\Xi}{T} + \Omega\right],$$

*where* $\Xi = \frac{C_1 G^2 d}{\sqrt{\epsilon}} + \frac{2G_1^2 \eta \eta KLG^2 d}{\epsilon}$, $\Omega = \frac{5\eta^2 KL^2}{\sqrt{2}\epsilon}(\sigma_l^2 + 6K\sigma_q^2) + [(3 + C_l^2)\eta L + 2\sqrt{2(1-\beta_2)}G](\frac{\eta_l}{n\eta\epsilon}\sigma_l^2 + \frac{2\eta_l K^2 G^2}{\eta\epsilon}) + [(3+C_1^2)\eta L + 2\sqrt{2(1-\beta_2)}G]\frac{\eta_l(m-n)}{2n(m-1)\epsilon}[15K^2 L^2 \eta_l^2(\sigma_l^2 + 6K\sigma_g^2) + 3K\sigma_g^2]\frac{1}{\eta\eta_l K} + \frac{\sqrt{2}G}{\epsilon}$ *and* $C_1 = \frac{\beta_1}{1-\beta_1}$.

*Remark* 4.3. When the parameters $C = D$ and $\frac{C}{\alpha n} = 1$, result of Theorem 4.3 becomes the one of Theorem 4.4. The upper bound for $\min_{t \in [T]}\mathbb{E}[\|\nabla f(\theta_t)\|^2]$ contains three terms: The first two terms decrease as $T$ increases, and this term tends to zero as $t$ tends to infinity. The last term relates to the local stochastic variance $\sigma_l$ and global variance $\sigma_g$. In the i.i.d setting, where the global variance is zero and each worker has the same data distribution, i.e., $\sigma_g = 0$, the variance term $\Omega$ will be smaller.

**Corollary 4.2.** *Suppose choose local learning rate $\eta_l = \Theta(\frac{1}{\sqrt{TK}})$ and global learning rate $\eta = \Theta(\sqrt{Kn})$, the convergence rate for FedNLAA and FedAA in Algorithm 1 under partial participation without replacement sampling is*

$$\min_{t \in [T]} \mathbb{E}\big[\|\nabla f(\theta_t)\|^2\big] = \mathcal{O}\left(\frac{\sqrt{K}}{\sqrt{Tn}}\right).$$

*Remark* 4.4. Note that Corollary 4.2 suggests that Theorems 4.3 and 4.4 directly relate to the global variance $\sigma_q^2$. Such convergence rate is consistent with the partial participation result of FedAvg in the non-i.i.d case in Yang et al. (2021a). It is shown that the global variance has more influence on the convergence behavior in partial participation cases. This is especially true for highly non-i.i.d cases where $\sigma_g$ is large. The effect of the number of local updates, $K$, is complex. In partial participation settings, the larger value of $K$ results in a slower convergence, while full participation suggests the opposite. A similar slowdown was also seen in Wang et al. (2022a).

---

**Algorithm 2** FedBNLACA and FedBACA.

**Input:** initial value $\theta_1, \theta_0 = 0$, local step size $\eta_l$, global step size $\eta$, constants $\beta_1, \beta_2$ and $\epsilon$, for each client $i \in S_t$, $\Delta_0^i = 0$, compressor $C(\cdot)$.

1: $\mathbf{m}_0 \leftarrow 0, \mathbf{v}_0 \leftarrow 0, \mathbf{e}_1^i = 0, \mathbf{E}_1^i = 0$.
2: **for** $t = 1$ **to** $T$ **do**
3:     On the server: Server randomly selects a subset of clients $S_t$
4:     **for** each client $j \in S_t$ in parallel **do**
5:         $\theta_t^i = \theta_t$
6:         Compress: $\widehat{\theta}_t^i = C(\theta_t^i + \mathbf{E}_t^i), C(Q_t^i) = \widehat{\theta}_t^i - \widehat{\theta}_{t-1}^i$,
7:         Judge: If $C(Q_t^i)$ satisfies NLA ( Example 3) or AA (Example 6), output: $\widehat{\widetilde{\theta}}_t^i$
8:         Update: $\mathbf{E}_{t+1}^i = \theta_t^i + \mathbf{E}_t^i - \widehat{\widetilde{\theta}}_t^i$,
9:     **end for**
10:     **for** each client $j \notin S_t$ in parallel **do**
11:         maintain stale compression error $\mathbf{E}_{t+1}^j = \mathbf{E}_t^j$.
12:     **end for**
13:     On the clients: $\theta_{t,0}^i = \widehat{\widetilde{\theta}}_t^i$
14:     **for** each client $i \in S_t$ in parallel **do**
15:         **for** for $k = 0, ..., K - 1$ **do**
16:             Compute local SGD: $\mathbf{g}_{t,k}^i = \nabla F_i(\theta_{t,k}^i; \xi_{t,k}^i)$,
17:             $\theta_{t,k+1}^i = \theta_{t,k}^i - \eta_l \mathbf{g}_{t,k}^i$.
18:         **end for**
19:         $\Delta_t^i = \theta_{t,K}^i - \widehat{\widetilde{\theta}}_t^i$
20:         Following the same way as in Algorithm 3 (Line 11-13)
21:     **end for**
22:     Following the same way as in Algorithm 3 (Line 14-18)
23: **end for**

---

In the following, we will give Algorithms 2 and 3. Due to space constraints, we will only give Algorithm 2 here, and Algorithm 3 is moved into the appendix A. Algorithm 2, which is the most important one in this paper, is an effective adaptive federated learning algorithm for non-convex heterogeneity with bidirectional communication.

### 4.3 FEDBNLACA AND FEDBACA ALGORITHMS

Algorithm 3 gives only a one-way algorithm to reduce the cost of communication (unplink). In the section, we propose two bidirectional communication algorithms (uplink and downlink) with efficiently adaptive non-convex optimization: **Fed**erated **B**idirectional **N**ew **L**azy **A**ggregation **C**ompression **A**MSGrad (FedBNLACA) and **Fed**erated **B**idirectional **A**ccelerated **C**ompression **A**MSGrad (FedBACA). The detailed procedure is given in Algorithm 2.

Next, we show the convergence analysis for FedBNLACA and FedBACA. Due to the space limit, we only show the full participation setting and leave the partial participation setting in Appendix E.2.

**Theorem 4.5.** *Under Assumptions 3.1-3.3, if the local learning rate $\eta_l$ satisfies:* $\eta_l \leq \min\big\{\frac{1}{8KL},$

$\frac{\epsilon}{KC_{\beta,q}[3\eta L + 2C_2\eta_l L + 2\sqrt{2(1-\beta_2)}G]}\big\}$, *where* $C_{\beta,q} = \sqrt{4\beta_2(1+q^2)^3(1-q^2)^{-2}K^2G^2 + \epsilon}$, *then FedBNLACA in Algorithm 2 under partial participation has*

$$\min \mathbb{E}[\|\nabla f(\theta_t)\|^2] \leq 4\sqrt{4\beta_2 \frac{(1+q^2)^3}{(1-q^2)^2}\eta_l^2 K^2 G^2 + \epsilon}\Big[\frac{f_0 - f_*}{\eta\eta_l KT} + \frac{\Xi}{T} + \Omega\Big],$$

*where* $\Xi = \frac{C_1 G^2 d}{\sqrt{\epsilon}} + \frac{2C_1^2\eta\eta_l KLG^2 d}{\epsilon}$, $\Omega = \Big[G + \frac{L\eta\eta_l KG}{\sqrt{\epsilon}} + \frac{L\eta\eta_l C_1 KGd}{\epsilon}\Big]\frac{\eta(\gamma + \frac{C}{\alpha m})H}{(1-\beta)\sqrt{\epsilon}} + \frac{2L\eta^2(\gamma^2 + \frac{C^2}{\alpha^2 m^2})H^2}{(1-\beta)^2\epsilon} + \frac{2L\eta^2(\gamma^2 + \frac{C^2}{\alpha^2 m^2})H^2}{(1-\beta)\sqrt{\epsilon}}\frac{5\eta^2 KL^2}{\sqrt{2\epsilon}}(\sigma_l^2 + 6K\sigma_g^2) + [(3 + 2C_2)\eta L +$

$2\sqrt{2(1-\beta_2)}G]\frac{\eta_l}{2m\eta\epsilon}\sigma_l^2$, $C_1 = \frac{\beta_1}{1-\beta_1} + \sqrt{\frac{12q^2}{(1-q^2)^2} + \frac{(1-q^2)^2C^2}{\alpha^2m^2q^2}} + \frac{2TL^2\eta^2(\gamma^2+\frac{C^2}{\alpha^2m^2})H^2}{(1-\beta)^2\epsilon}(\frac{C^2}{\alpha^2m^2}+1)$

and $C_2 = \frac{\beta_1^2}{(1-\beta_1)^2} + \frac{4(q+\gamma+\frac{\lambda C}{\alpha m})^2}{(1-q^2)^2}$.

**Theorem 4.6.** *Under Assumptions 3.1-3.3, if the local learning rate $\eta_l$ satisfies: $\eta_l \leq \min\left\{\frac{1}{8KL},\right.$*

$\left.\frac{\epsilon}{KC_{\beta,q}[3\eta L+2C_2\eta L+2\sqrt{2(1-\beta_2)}G]}\right\}$, *where* $C_{\beta,q} = \sqrt{4\beta_2(1+q^2)^3(1-q^2)^{-2}K^2G^2 + \epsilon}$, *then FedBACA in Algorithm 2 under partial participation has*

$$\min \mathbb{E}[\|\nabla f(\theta_t)\|^2] \leq 4\sqrt{4\beta_2\frac{(1+q^2)^3}{(1-q^2)^2}\eta_l^2K^2G^2 + \epsilon}\left[\frac{f_0-f_*}{\eta\eta_lKT} + \frac{\Xi}{T} + \Omega\right],$$

*where* $\Xi = \frac{C_1G^2d}{\sqrt{\epsilon}} + \frac{2C_1^2\eta\eta KLG^2d}{\epsilon}$, $\Omega = \left[G + \frac{L\eta\eta_lKG}{\sqrt{\epsilon}} + \frac{L\eta\eta_lC_1KGd}{\epsilon}\right] \cdot \frac{\eta(\gamma+\frac{C}{\alpha m})H}{(1-\beta)\sqrt{\epsilon}} + \frac{2L\eta^2(\gamma^2+\frac{C^2}{\alpha^2m^2})H^2}{(1-\beta)^2\epsilon} + \frac{2L\eta^2(\gamma^2+\frac{C^2}{\alpha^2m^2})H^2}{(1-\beta)\sqrt{\epsilon}}\frac{5\eta^2KL^2}{\sqrt{2\epsilon}}(\sigma_l^2 + 6K\sigma_g^2) + [(3 + 2C_2)\eta L + 2\sqrt{2(1-\beta_2)}G]\frac{\eta_l}{2m\eta\epsilon}\sigma_l^2$, $C_1 = \frac{\beta_1}{1-\beta_1} + \sqrt{\frac{12q^2}{(1-q^2)^2} + \frac{(1-q^2)^2C^2}{\alpha^2m^2q^2}} + \frac{2TL^2\eta^2(\gamma^2+\frac{C^2}{\alpha^2m^2})H^2}{(1-\beta)^2\epsilon}(\frac{C^2}{\alpha^2m^2}+1)$

*and* $C_2 = \frac{\beta_1^2}{(1-\beta_1)^2} + \frac{4(q+\gamma+\frac{\lambda C}{\alpha m})^2}{(1-q^2)^2}$.

*Remark* 4.5. When the parameters $C = D$ and $\frac{C}{\alpha m} = 1$, result of Theorem 4.5 becomes the one of Theorem 4.6. The upper bound for $\min_{t\in[T]}\mathbb{E}[\|\nabla f(\theta_t)\|^2]$ contains three terms: The first two terms decrease as $T$ increases, and this term tends to zero as $t$ tends to infinity. The last term relates to the local stochastic variance $\sigma_l$ and global variance $\sigma_g$. In the i.i.d setting, where the global variance is zero and each worker has the same data distribution, i.e., $\sigma_g = 0$, the variance term $\Omega$ will be smaller.

**Corollary 4.3.** *Suppose choose local learning rate $\eta_l = \Theta(\frac{1}{\sqrt{TK}})$ and global learning rate $\eta = \Theta(\sqrt{Km})$, when $T$ is sufficiently large, i.e.,$T = \mathcal{O}(Km)$, the convergence rate for FedNLAA and FedAA in Algorithm 1 under full participation has*

$$\min_{t\in[T]} \mathbb{E}\left[\|\nabla f(\theta_t)\|^2\right] = \mathcal{O}\left(\frac{1}{T}\right).$$

*Remark* 4.6. Corollary 4.3 suggests that with sufficient large $T$, FedBNLACA and FedBACA achieve a convergence rate of $\mathcal{O}(\frac{1}{T})$, which matches the result for general federated non-convex optimization methods such as FedAMS Wang et al. (2022a) and FedAdam Reddi et al. (2020).

# 5 EXPERIMENTS

We compare our proposed algorithms with several state-of-the-art algorithms (FedAvgMcMahan et al. (2017), FedAMS, FedCAMS(Wang et al. (2022a)), Fedadam (Reddi et al. (2020)). We use MNIST and Fashion-MNIST datasets, and models by MLP and CNN, respectively. A total of 100 clients for all federated training experiments are used. Set the partial participation rate to 0.5, i.e. in each round, the server selects 50 clients out of 100 to participate in communication and model updating. In each round, the client completes 3 local epochs with batch size 32. In experiments, we respectively sample Independent Identical Distribution (I.I.D.) and non-I.I.D. client data from the dataset. Choose compression rate in Top$k$ to be $1/8$ and $1/128$. For parameters $C$ and $D$, the values are not too large. From theoretical analysis, the larger the values of $C$ and $D$, the larger the errors. In addition, $C$ and $D$ are too small, and basically does not help to reduce communication cost. We also verify this result. We suggest that $C$ and $D$ in the vicinity of $\frac{\sqrt{\alpha\lg m}}{m}$.

Figures 1-2 represent the relationship between the accuracy of prediction and communication Bits when the model is CNN and the I.I.D. client data is sampled from Fashion-MNIST dataset. From Figure 1, we can find that (i) the proposed algorithm FedNLAA not only communicates fewer Bits than the other three state-of-the-art algorithms, but also has higher accuracy; (ii) our proposed algorithms (FedNLACA and FedBNLACA) require only few communication Bits to achieve good accuracy, especially the bidirectional compression algorithm FedBNLACA requires even fewer Bits. These shows that our proposed algorithms are communication efficient. From Figure 2, we can find that (i) the proposed algorithm FedAA algorithm can converge more quickly than the other three

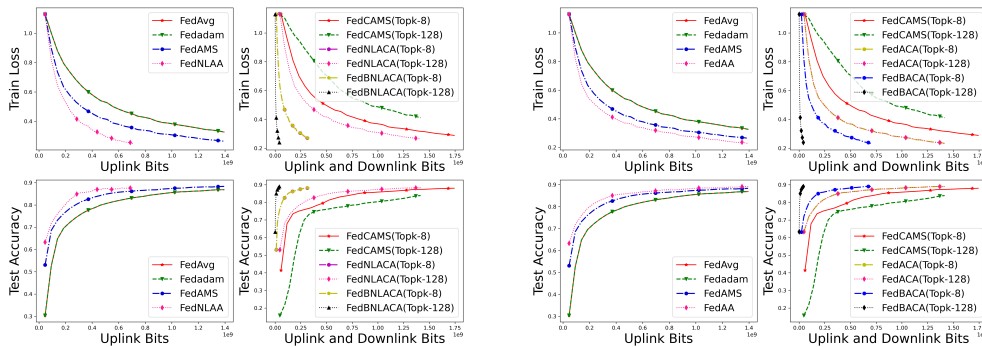

Figure 1: NLA strategy, on Fashion-MNIST via CNN model, and the I.I.D. client sampling.

Figure 2: AA strategy, on Fashion-MNIST via CNN model, and the I.I.D. client sampling.

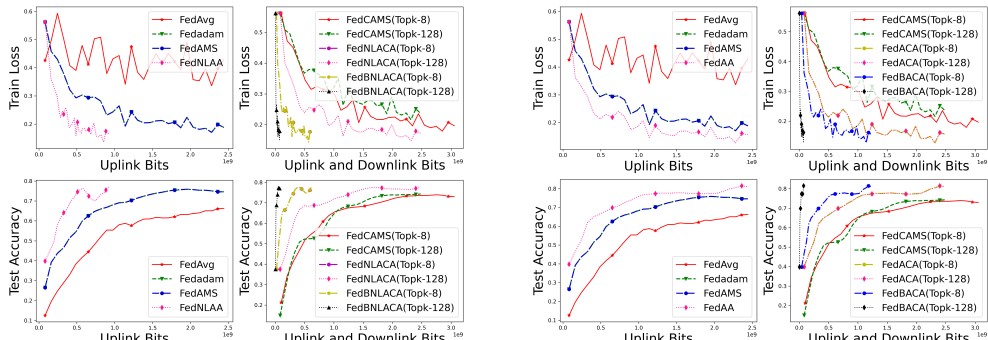

Figure 3: NLA strategy, on Fashion-MNIST via CNN model, and the non-I.I.D. client sampling.

Figure 4: AA strategy, on Fashion-MNIST via CNN model, and the non-I.I.D. client sampling.

algorithms, thus disguising the reduction of communication cost; (ii) FedACA and FedBACA can also converge quickly and with higher accuracy than the other three algorithms, especially FedBACA.

Figures 3-4 represent the relationship between the accuracy of prediction and communication Bits when the model is CNN and the non-I.I.D. client data is sampled from Fashion-MNIST dataset..From Figure 3, we can find that (i) the proposed algorithm FedNLAA not only communicates fewer Bits than the other three state-of-the-art algorithms, but also has higher accuracy; (ii) FedAvg performs the worst; (iii) our proposed algorithms (FedNLACA and FedBNLACA) require only few communication Bits to achieve good accuracy, especially the bidirectional compression algorithm FedBNLACA requires even fewer Bits. These also show that our proposed algorithms are communication efficient. From Figure 4, we can find that (i) the proposed FedAA algorithm can converge more quickly than the other three algorithms, thus disguising the reduction of communication cost; (ii) our algorithms (FedACA and FedBACA) can also converge quickly and with higher accuracy than the other three algorithms, especially FedBACA.

# 6 CONCLUSION

We propose two novel strategies: NLA and AA in the framework of federated learning. They are simple to operate and effective in reducing the communication cost. The NLA strategy achieves communication cost reduction by reducing the amount of information passed and AA strategy reduces the communication cost by accelerating computation. By combining our proposed strategies with compression techniques, we design FedNLAA and FedAA algorithms, which not only achieve communication cost reduction in one-way, but also extend them to bidirectional algorithms (FedBNLACA and FedBACA), which achieve communication cost reduction in bidirectional as well.

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

# Appendix

## A  FEDNLACA AND FEDACA ALGORITHMS

### A.1  PROCEDURE OF FEDNLACA AND FEDACA

In order to reduce communication costs, we propose **Fed**erated **N**ew **L**azy **A**ggregation **C**ompression **AMSG**rad (FedNLACA) and **Fed**erated **A**ccelerated **C**ompression **AMSG**rad (FedACA). These two algorithms combine two of our proposed strategies(strategy 1 (FedNLACA), strategy 2 (FedACA) ) and compression techniques. The detailed procedure is given in Algorithm 3.

### A.2  CONVERGENCE ANALYSIS FOR FEDNLACA AND FEDACA

In the case of full participation:

**Theorem A.1.** *Under Assumption 3.1-3.3, if the local learning rate $\eta_l$ satisfies:* $\eta \leq \min\left\{ \frac{1}{8KL}, \frac{\epsilon}{KC_{\beta,q}[3\eta L + 2C_2\eta L + 2\sqrt{2(1-\beta_2)}G]} \right\}$, *where* $C_{\beta,q} = \sqrt{4\beta_2(1+q^2)^3(1-q^2)^{-2}K^2G^2+\epsilon}$, *then the iterates of FedBNLACA in Algorithm 2 under partial participation scheme satisfy*

$$\min \mathbb{E}[\|\nabla f(\theta_t)\|^2] \leq 4\sqrt{4\beta_2 \frac{(1+q^2)^3}{(1-q^2)^2}\eta_l^2 K^2 G^2 + \epsilon}\left[\frac{f_0 - f_*}{\eta\eta_l KT} + \frac{\Xi}{T} + \Omega\right],,$$

*where* $\Xi = \frac{C_1 G^2 d}{\sqrt{\epsilon}} + \frac{2C_1^2\eta\eta KLG^2 d}{\epsilon}, \Omega = \frac{5\eta^2 KL^2}{\sqrt{2\epsilon}}(\sigma_l^2 + 6K\sigma_g^2) + [(3 + 2C_2)\eta L + 2\sqrt{2(1-\beta_2)}G]\frac{\eta_l}{2m\eta\epsilon}\sigma_l^2$, $C_1 = \frac{\beta_1}{1-\beta_1} + \sqrt{\frac{12q^2}{(1-q^2)^2} + \frac{(1-q^2)^2 C^2}{\alpha^2 m^2 q^2}}$ *and* $C_2 = \frac{\beta_1^2}{(1-\beta_1)^2} + \frac{4(q+\gamma+\frac{\lambda C}{\alpha m})^2}{(1-q^2)^2}$.

**Theorem A.2.** *Under Assumption 3.1-3.3, if the local learning rate $\eta_l$ satisfies:* $\eta \leq \min\left\{ \frac{1}{8KL}, \frac{\epsilon}{KC_{\beta,q}[3\eta L + 2C_2\eta L + 2\sqrt{2(1-\beta_2)}G]} \right\}$, *where* $C_{\beta,q} = \sqrt{4\beta_2(1+q^2)^3(1-q^2)^{-2}K^2G^2+\epsilon}$, *then the iterates of FedBACA in Algorithm 2 under partial participation scheme satisfy*

$$\min \mathbb{E}[\|\nabla f(\theta_t)\|^2] \leq 4\sqrt{4\beta_2 \frac{(1+q^2)^3}{(1-q^2)^2}\eta_l^2 K^2 G^2 + \epsilon}\left[\frac{f_0 - f_*}{\eta\eta_l KT} + \frac{\Xi}{T} + \Omega\right],,$$

*where* $\Xi = \frac{C_1 G^2 d}{\sqrt{\epsilon}} + \frac{2C_1^2\eta\eta KLG^2 d}{\epsilon}, \Omega = \frac{5\eta^2 KL^2}{\sqrt{2\epsilon}}(\sigma_l^2 + 6K\sigma_g^2) + [(3 + 2C_2)\eta L + 2\sqrt{2(1-\beta_2)}G]\frac{\eta_l}{2m\eta\epsilon}\sigma_l^2$, $C_1 = \frac{\beta_1}{1-\beta_1} + \sqrt{\frac{12q^2}{(1-q^2)^2} + \frac{(1-q^2)^2}{q^2}}$ *and* $C_2 = \frac{\beta_1^2}{(1-\beta_1)^2} + \frac{4(q+\gamma+\lambda)^2}{(1-q^2)^2}$.

*Remark* A.1. When the parameters $C = D, \frac{C}{\alpha m} = 1$, the result of Theorem A.1 becomes the result of Theorem A.2. The upper bound for $\min_{t\in[T]}\mathbb{E}[\|\nabla f(\theta_t)\|^2]$ contains three terms: The first two terms decrease as $T$ increases, and this term tends to zero as t tends to infinity. The last term relates to the local stochastic variance $\sigma_l$ and global variance $\sigma_g$. In the i.i.d setting, where the global variance is zero and each worker has the same data distribution, i.e., $\sigma_g = 0$, the variance term $\Omega$ will be smaller.

**Corollary A.1.** *Suppose we choose local learning rate $\eta_l = \Theta(\frac{1}{\sqrt{TK}})$ and the global learning rate $\eta = \Theta(\sqrt{Km})$, when $T$ is sufficient large, i.e.,$T \geq Km$,the convergence rate for FedNLACA and FedACA in Algorithm I under full participation scheme satisfies*

$$\min_{t\in[T]} \mathbb{E}[\|\nabla f(\theta_t)\|^2] = \mathcal{O}\left(\frac{1}{\sqrt{TKm}}\right).$$

*Remark* A.2. Corollary A.1 suggests that with sufficient large $T$, FedCAMS achieves the desired $\mathcal{O}(\frac{1}{\sqrt{TKm}})$ convergence rate which matches the result for its uncompressed counterpart FedNLAA and FedAA. In addition, when $T = \mathcal{O}(Km)$, $\min_{t\in[T]} \mathbb{E}[\|\nabla f(\theta_t)\|^2] = \mathcal{O}\left(\frac{1}{T}\right)$. This suggests that FedNLACA and FedACA can indeed achieve better communication efficiency without sacrificing much on the accuracy.

---

**Algorithm 3** FedNLACA and FedACA

---

**Input:** initial value $\theta_1$, local step size $\eta_l$, global step size $\eta$, constant $\beta_1,\beta_2,\epsilon$, for each client $i \in S_t$, $\Delta_0^i = 0$, compressor $C(\cdot)$

1: $\mathbf{m}_0 \leftarrow 0, \mathbf{v}_0 \leftarrow 0, \mathbf{e}_1^i = 0$
2: **for** $t = 1$ **to** $T$ **do**
3:      Randomly select a subset of clients $S_t$ and the server transmits $\theta_t$ to the subset of clients $S_t$
4:      $\theta_{t,0}^i = \theta_t$
5:      **for** each client $i \in S_t$ in parallel **do**
6:          **for** for $k = 0, ..., K-1$ **do**
7:              Compute local stochastic gradient: $\mathbf{g}_{t,k}^i = \nabla F_i(\theta_{t,k}^i; \xi_{t,k}^i)$
8:              $\theta_{t,k+1}^i = \theta_{t,k}^i - \eta_l \mathbf{g}_{t,k}^i$
9:          **end for**
10:         $\Delta_t^i = \theta_{t,K}^i - \theta_t$
11:         Compress $\widehat{\Delta}_t^i = C(\Delta_t^i + \mathbf{e}_t^i), C(q_t^i) = \widehat{\Delta}_t^i - \widehat{\Delta}_{t-1}^i$,
12:         Judge: If $C(q_t^i)$ satisfies NLA (Example 2) or AA (Example 5),
13:         then outputs the result $\widehat{\widehat{\Delta}}_t^i$ of the judgement and passes it to the server and update $\mathbf{e}_{t+1}^i = \Delta_t^i + \mathbf{e}_t^i - \widehat{\widehat{\Delta}}_t^i$
14:      **end for**
15:      **for** each client $j \notin S_t$ in parallel do **do**
16:         client $j$ maintains the stale compression error $\mathbf{e}_{t+1}^j = \mathbf{e}_t^j$
17:      **end for**
18:      Server aggregates local update: $\widehat{\Delta}_t = \frac{1}{|S_t|} \sum_{i \in \mathcal{S}_t} \widehat{\widehat{\Delta}}_t^i$
19:      Server updates $\mathbf{x}_{t+1}$ using $\hat{\Delta}_t$ in the same way as in Algorithm 1 (Line 14-16)
20: **end for**

---

In the case of partial participation

**Theorem A.3.** *Under Assumption 3.1-3.3, if the local learning rate $\eta_l$ satisfies:* $\eta_l \leq \min\left\{\frac{1}{8KL}, \frac{\epsilon}{KC_{\beta,q}[3\eta L + 2C_2\eta_l L + 2\sqrt{2(1-\beta_2)}G]}\right\}$, *where* $C_{\beta,q} = \sqrt{4\beta_2(1+q^2)^3(1-q^2)^{-2}K^2G^2 + \epsilon}$, *then the iterates of FedBNLACA in Algorithm 2 under partial participation scheme satisfy*

$$\min \mathbb{E}[\|\nabla f(\theta_t)\|^2] \leq 8\sqrt{4\beta_2 \frac{(1+q^2)^3}{(1-q^2)^2}\eta_l^2 K^2 G^2 + \epsilon} \left[\frac{f_0 - f_*}{\eta\eta_l KT} + \frac{\Xi}{T} + \Omega\right]$$

*, where* $\Xi = \frac{C_1 G^3 d}{\sqrt{\epsilon}} + \frac{2C_1^2 \eta\eta_l KLG^2 d}{\epsilon}, \Omega = \frac{C_1 \eta\eta_l KLG^2}{\epsilon} + \frac{5\eta^2 KL^2}{\sqrt{2\epsilon}}(\sigma_l^2 + 6K\sigma_g^2) + [\eta L + \sqrt{2(1-\beta_2)}G]\frac{\eta_l}{\eta n\epsilon}\sigma_l^2 + [\eta L + \sqrt{2(1-\beta_2)}G]\frac{\eta_l(m-n)}{n(m-1)\epsilon}[15K^2 L^2 \eta_l^2(\sigma_l^2 + 6K\sigma_g^2) + 3K\sigma_g^2]$ *and* $C_1 = \frac{\beta_1}{1-\beta_1} + \frac{m}{n}\sqrt{\frac{12q^2}{(1-q^2)^2} + \frac{(1-q^2)^2 C^2}{\alpha^2 n^2 q^2}}$.

**Theorem A.4.** *Under Assumption 3.1-3.3, if the local learning rate $\eta_l$ satisfies:* $\eta_l \leq \min\left\{\frac{1}{8KL}, \frac{\epsilon}{KC_{\beta,q}[3\eta L + 2C_2\eta_l L + 2\sqrt{2(1-\beta_2)}G]}\right\}$, *where* $C_{\beta,q} = \sqrt{4\beta_2(1+q^2)^3(1-q^2)^{-2}K^2G^2 + \epsilon}$, *then the iterates of FedBACA in Algorithm 2 under partial participation scheme satisfy*

$$\min \mathbb{E}[\|\nabla f(\theta_t)\|^2] \leq 8\sqrt{4\beta_2 \frac{(1+q^2)^3}{(1-q^2)^2}\eta_l^2 K^2 G^2 + \epsilon} \left[\frac{f_0 - f_*}{\eta\eta_l KT} + \frac{\Xi}{T} + \Omega\right],$$

*where* $\Xi = \frac{C_1 G^3 d}{\sqrt{\epsilon}} + \frac{2C_1^2 \eta\eta_l KLG^2 d}{\epsilon}, \Omega = \frac{C_1 \eta\eta_l KLG^2}{\epsilon} + \frac{5\eta^2 KL^2}{\sqrt{2\epsilon}}(\sigma_l^2 + 6K\sigma_g^2) + [\eta L + \sqrt{2(1-\beta_2)}G]\frac{\eta_l}{\eta n\epsilon}\sigma_l^2 + [\eta L + \sqrt{2(1-\beta_2)}G]\frac{\eta_l(m-n)}{n(m-1)\epsilon}[15K^2 L^2 \eta_l^2(\sigma_l^2 + 6K\sigma_g^2) + 3K\sigma_g^2]$ *and* $C_1 = \frac{\beta_1}{1-\beta_1} + \frac{m}{n}\sqrt{\frac{12q^2}{(1-q^2)^2} + \frac{(1-q^2)^2}{q^2}}$.

*Remark* A.3. When the parameters $C = D$, $\frac{C}{\alpha n} = 1$, the result of Theorem A.3 becomes the result of Theorem A.4. The upper bound for $\min_{t \in [T]} \mathbb{E}[\|\nabla f(\theta_t)\|^2]$ contains three terms: The first two terms decrease as $T$ increases, and this term tends to zero as t tends to infinity. The last term relates to the local stochastic variance $\sigma_l$ and global variance $\sigma_g$. In the i.i.d setting, where the global variance is zero and each worker has the same data distribution, i.e., $\sigma_g = 0$, the variance term $\Omega$ will be smaller.

**Corollary A.2.** *Suppose we choose local learning rate* $\eta_l = \Theta(\frac{1}{\sqrt{T}K})$ *and the global learning rate* $\eta = \Theta(\sqrt{Kn})$ *, the convergence rate for FedNLACA,FedACA in Algorithm 1 under partial participation scheme without replacement sampling is*

$$\min_{t \in [T]} \mathbb{E}\big[\|\nabla f(\theta_t)\|^2\big] = \mathcal{O}\bigg(\frac{\sqrt{K}}{\sqrt{T}n}\bigg).$$

*Remark* A.4. Note that Corollary A.2 suggests that Theorem A.3,A.4 directly relates to the global variance $\sigma_q^2$. Such convergence rate is consistent with the partial participation result of FedAvg in the non i.i.d case in Yang et al. (2021a). It is shown that the global variance has more influence on the convergence behaviour in partial participation cases. This is especially true for highly non i.i.d. cases where $\sigma_g$ is large.The effect of the number of local updates, $K$, is complex. In partial participation settings, larger values of $K$ result in slower convergence, while full participation suggests the opposite. A similar slowdown was also seen in Wang et al. (2022a).

# B ALL EXPERIMENTS

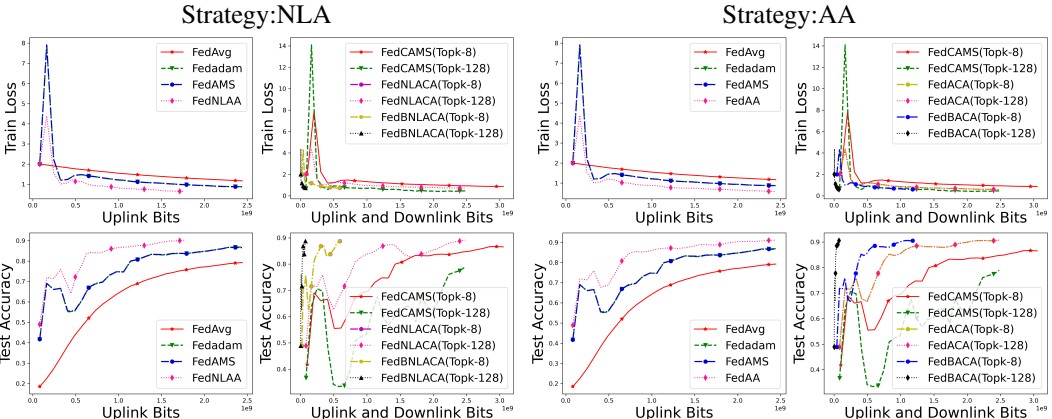

Figure 5: The above figure represents the relationship between the accuracy of the prediction and the communication Bits when the model used is MLP, the data is MNIST dataset, the dataset obeys independent identical distribution and the data is partially used. The left side represents the method using NLA (New Lazy Aggregation) strategy and the right side represents the method using AA (Accelerated Aggregation) strategy.

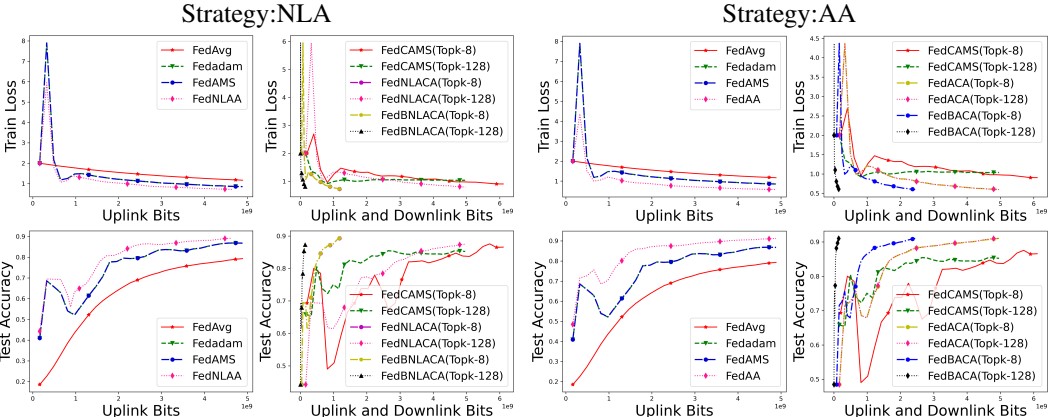

Figure 6: The above figure represents the relationship between the accuracy of the prediction and the communication Bits when the model used is MLP, the data is MNIST dataset, the dataset obeys independent identical distribution and the all data is used. The left side represents the method using NLA (New Lazy Aggregation) strategy and the right side represents the method using AA (Accelerated Aggregation) strategy.

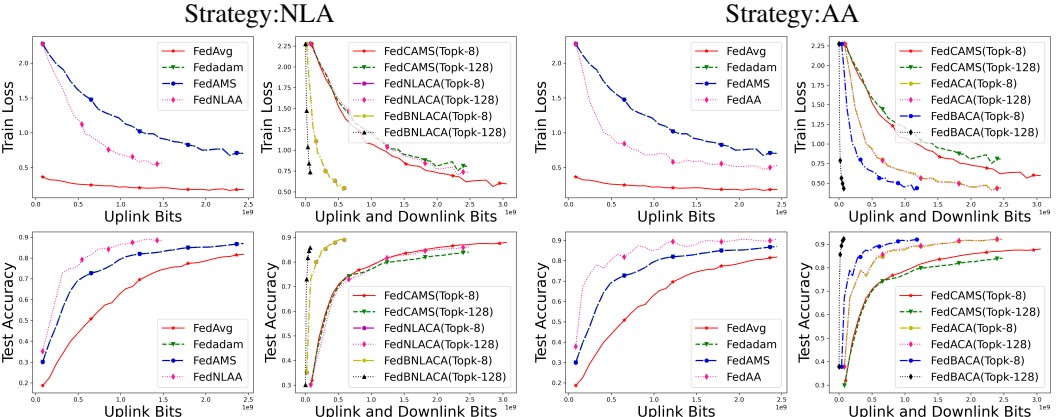

Figure 7: The above figure represents the relationship between the accuracy of the prediction and the communication Bits when the model used is MLP, the data is MNIST dataset, the dataset does not obeys independent identical distribution and the data is partially used. The left side represents the method using NLA (New Lazy Aggregation) strategy and the right side represents the method using AA (Accelerated Aggregation) strategy.

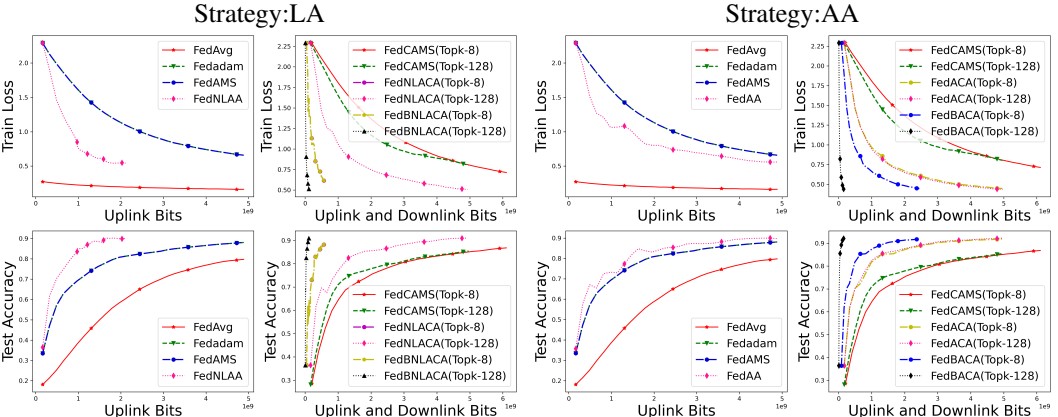

Figure 8: The above figure represents the relationship between the accuracy of the prediction and the communication Bits when the model used is MLP, the data is MNIST dataset, the dataset does not obeys independent identical distribution and all the data is used. The left side represents the method using NLA (New Lazy Aggregation) strategy and the right side represents the method using AA (Accelerated Aggregation) strategy.

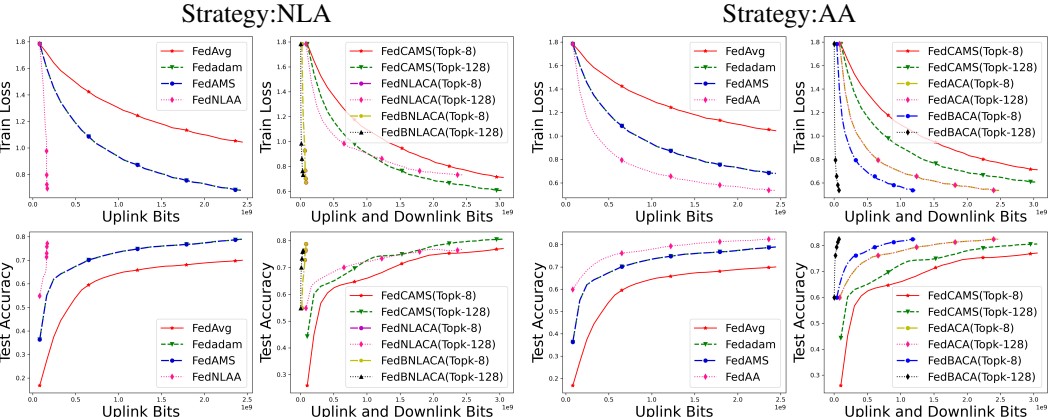

Figure 9: The above figure represents the relationship between the accuracy of the prediction and the communication Bits when the model used is MLP, the data is Fashion-MNIST dataset, the dataset obeys independent identical distribution and the data is partially used. The left side represents the method using NLA (New Lazy Aggregation) strategy and the right side represents the method using AA (Accelerated Aggregation) strategy.

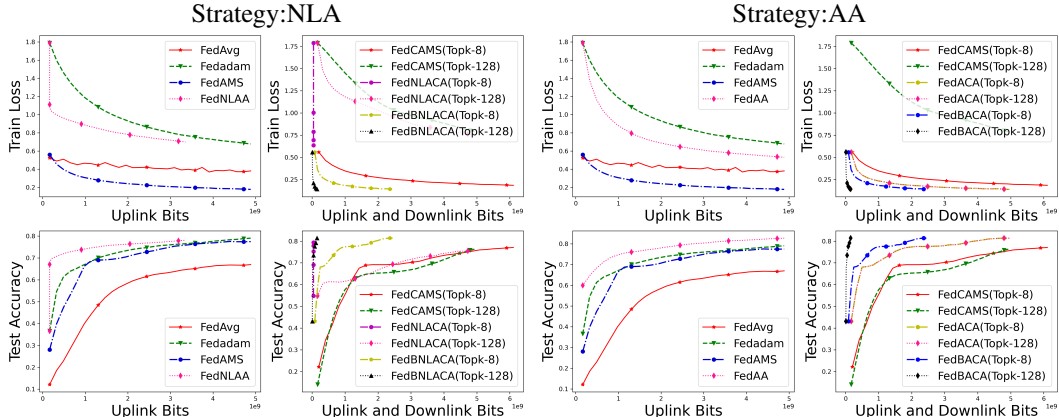

Figure 10: The above figure represents the relationship between the accuracy of the prediction and the communication Bits when the model used is MLP, the data is Fashion-MNIST dataset, the dataset obeys independent identical distribution and the all data is used. The left side represents the method using NLA (New Lazy Aggregation) strategy and the right side represents the method using AA (Accelerated Aggregation) strategy.

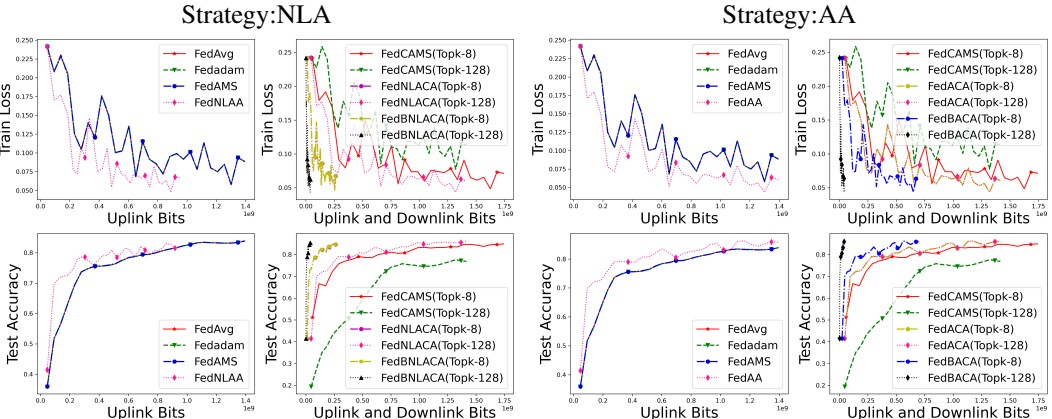

Figure 11: The above figure represents the relationship between the accuracy of the prediction and the communication Bits when the model used is MLP, the data is Fashion-MNIST dataset, the dataset does not obeys independent identical distribution and the data is partially used. The left side represents the method using NLA (New Lazy Aggregation) strategy and the right side represents the method using AA (Accelerated Aggregation) strategy.

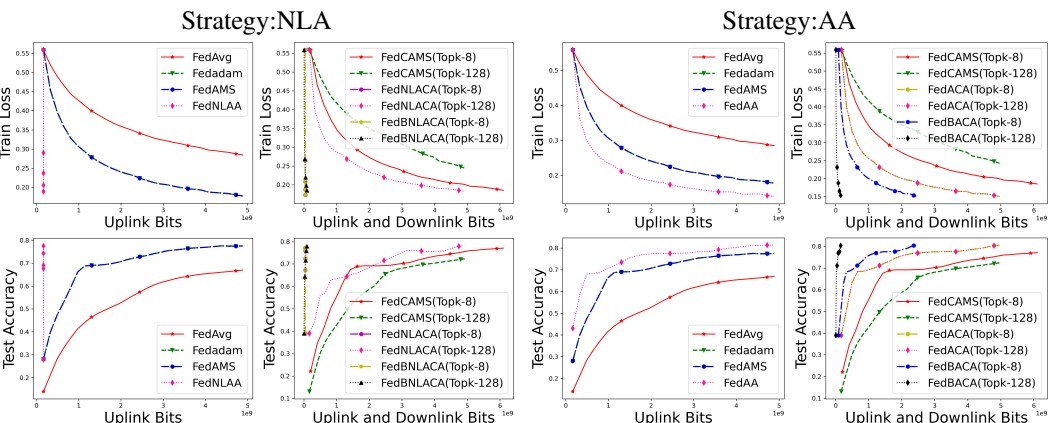

Figure 12: The above figure represents the relationship between the accuracy of the prediction and the communication Bits when the model used is MLP, the data is Fashion-MNIST dataset, the dataset does not obeys independent identical distribution and all the data is used. The left side represents the method using NLA (New Lazy Aggregation) strategy and the right side represents the method using AA (Accelerated Aggregation) strategy.

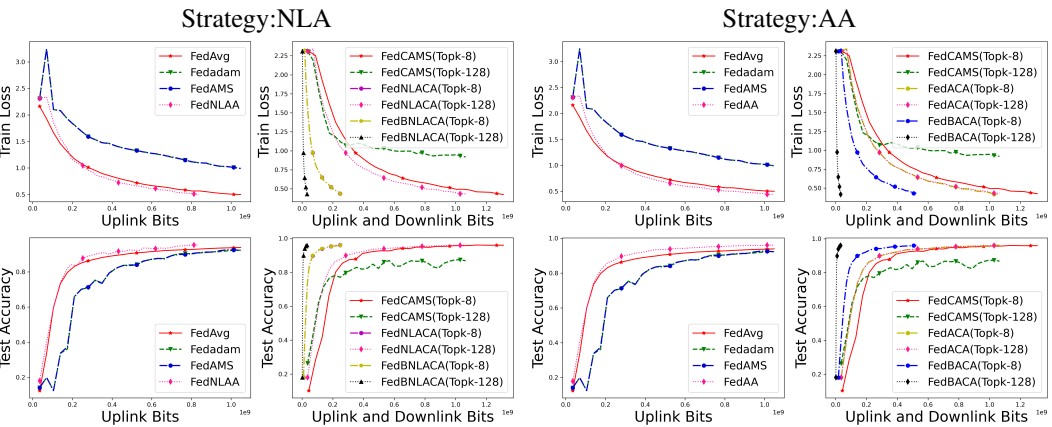

Figure 13: The above figure represents the relationship between the accuracy of the prediction and the communication Bits when the model used is CNN, the data is MNIST dataset, the dataset obeys independent identical distribution and the data is partially used. The left side represents the method using NLA (New Lazy Aggregation) strategy and the right side represents the method using AA (Accelerated Aggregation) strategy.

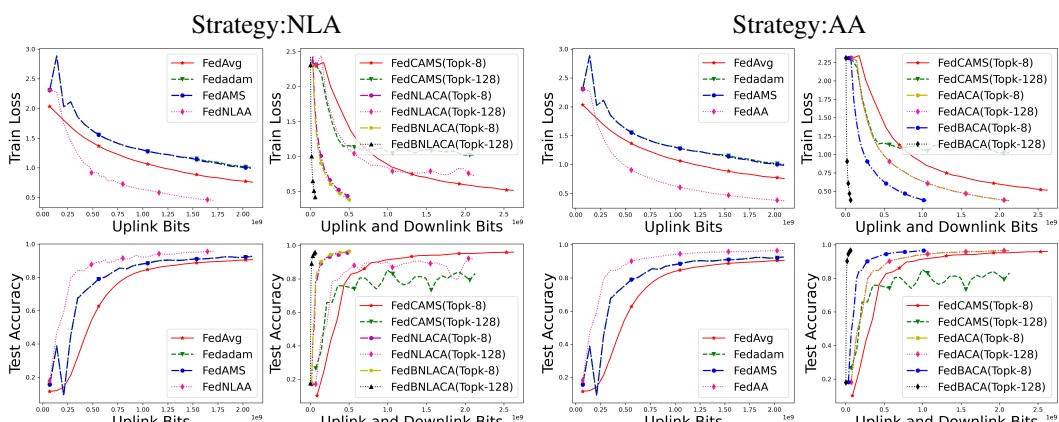

Figure 14: The above figure represents the relationship between the accuracy of the prediction and the communication Bits when the model used is CNN, the data is MNIST dataset, the dataset obeys independent identical distribution and the all data is used. The left side represents the method using NLA (New Lazy Aggregation) strategy and the right side represents the method using AA (Accelerated Aggregation) strategy.

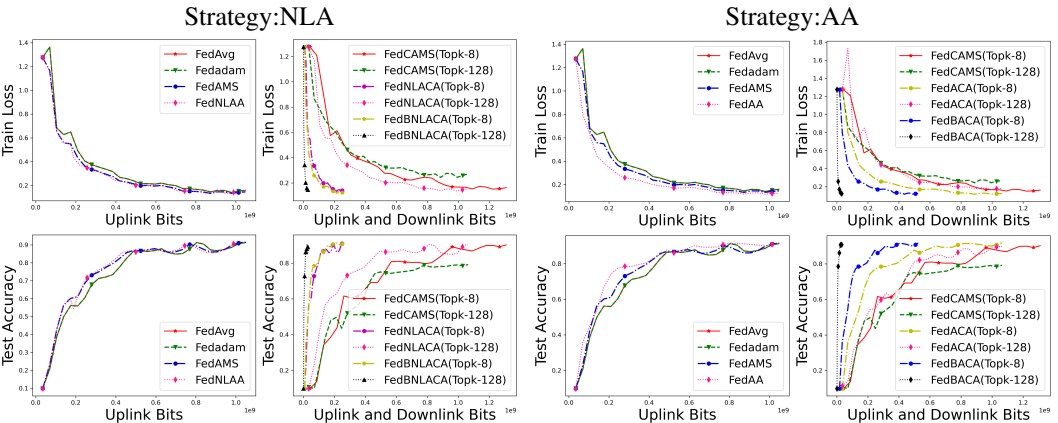

Figure 15: The above figure represents the relationship between the accuracy of the prediction and the communication Bits when the model used is CNN, the data is MNIST dataset, the dataset does not obeys independent identical distribution and the data is partially used. The left side represents the method using NLA (New Lazy Aggregation) strategy and the right side represents the method using AA (Accelerated Aggregation) strategy.

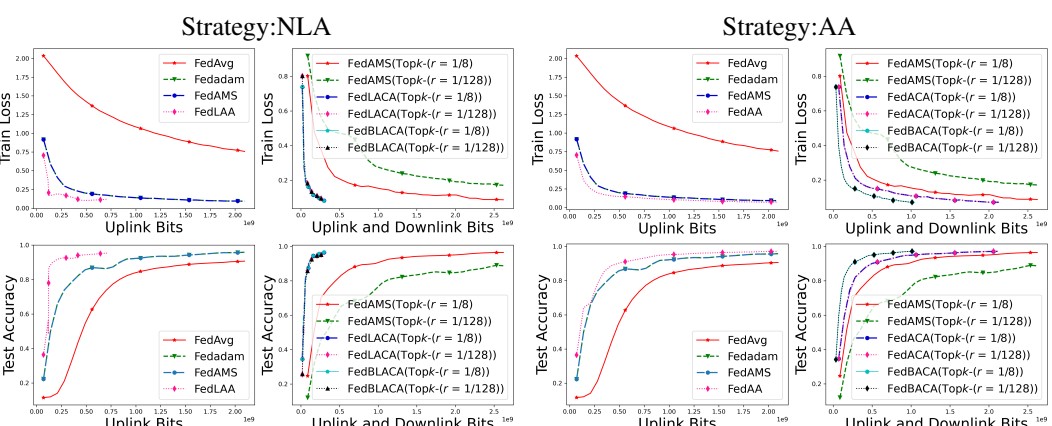

Figure 16: The above figure represents the relationship between the accuracy of the prediction and the communication Bits when the model used is CNN, the data is MNIST dataset, the dataset does not obeys independent identical distribution and all the data is used. The left side represents the method using NLA (New Lazy Aggregation) strategy and the right side represents the method using AA (Accelerated Aggregation) strategy.

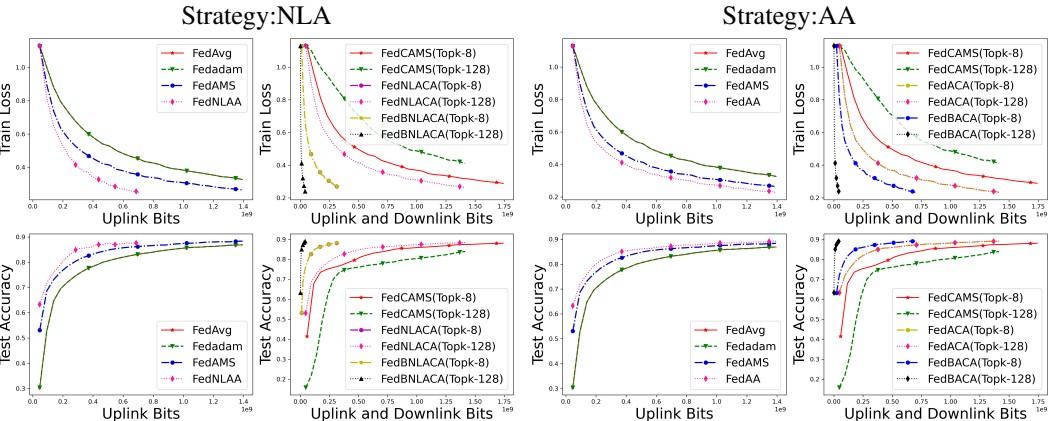

Figure 17: The above figure represents the relationship between the accuracy of the prediction and the communication Bits when the model used is CNN, the data is Fashion-MNIST dataset, the dataset obeys independent identical distribution and the data is partially used. The left side represents the method using NLA (New Lazy Aggregation) strategy and the right side represents the method using AA (Accelerated Aggregation) strategy.

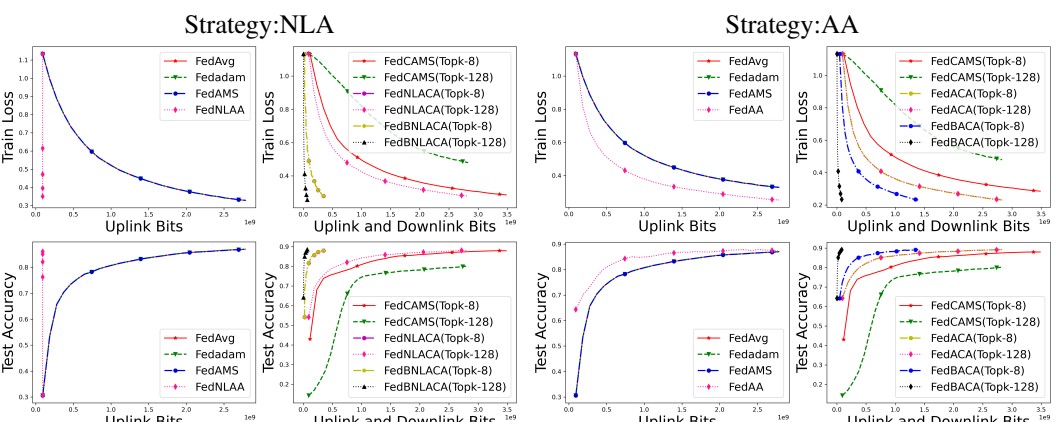

Figure 18: The above figure represents the relationship between the accuracy of the prediction and the communication Bits when the model used is CNN, the data is Fashion-MNIST dataset, the dataset obeys independent identical distribution and the all data is used. The left side represents the method using NLA (New Lazy Aggregation) strategy and the right side represents the method using AA (Accelerated Aggregation) strategy.

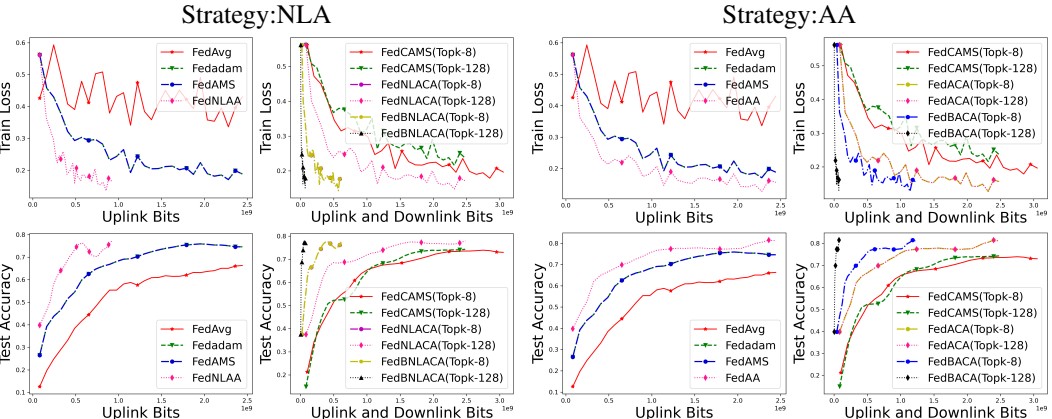

Figure 19: The above figure represents the relationship between the accuracy of the prediction and the communication Bits when the model used is CNN, the data is Fashion-MNIST dataset, the dataset does not obeys independent identical distribution and the data is partially used. The left side represents the method using NLA (New Lazy Aggregation) strategy and the right side represents the method using AA (Accelerated Aggregation) strategy.

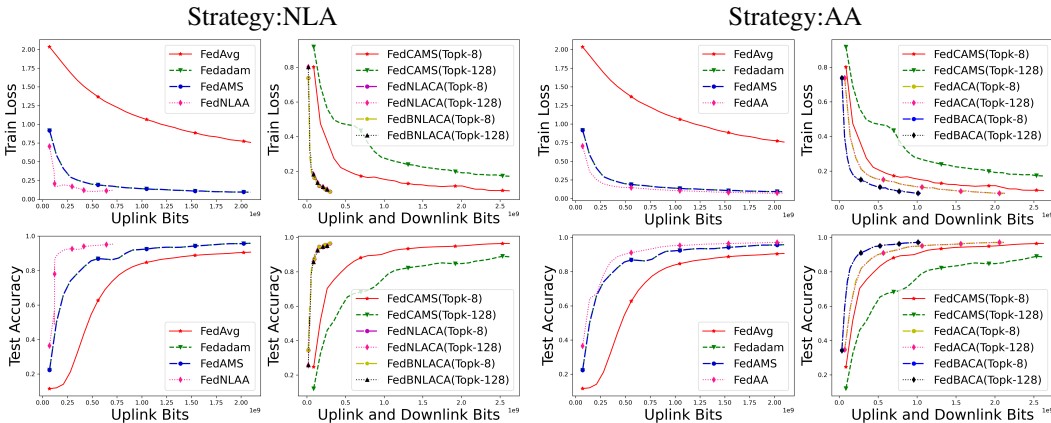

Figure 20: The above figure represents the relationship between the accuracy of the prediction and the communication Bits when the model used is CNN, the data is Fashion-MNIST dataset, the dataset does not obeys independent identical distribution and all the data is used. The left side represents the method using NLA (New Lazy Aggregation) strategy and the right side represents the method using AA (Accelerated Aggregation) strategy.

## C PROOF IN SECTION 4.1

### C.1 PROOF OF THEOREM 4.1

Similar to previous work in the field of adaptive methods Chen et al. (2018c); Wang et al. (2022a), we introduce a Lyapunov sequence $\mathbf{z}_t$: assume $\theta_0 = \theta_1$, for each $t \geq 1$, there is the following equation

$$\mathbf{z}_t = \theta_t + \frac{\beta_1}{1-\beta_1}(\theta_t - \theta_{t-1}) = \frac{1}{1-\beta_1}\theta_t - \frac{\beta_1}{1-\beta_1}\theta_{t-1}. \tag{C.1}$$

For the difference of sequence $z$, there is the following equation

$$\begin{aligned}
\mathbf{z}_{t+1} - \mathbf{z}_t &= \frac{1}{1-\beta_1}(\theta_{t+1} - \theta_t) - \frac{\beta_1}{1-\beta_1}(\theta_t - \theta_{t-1}) \\
&= \frac{1}{1-\beta_1}(\eta\widehat{\mathbf{V}}_t^{-1/2}\mathbf{m}_t) - \frac{\beta_1}{1-\beta_1}\eta\widehat{\mathbf{V}}_{t-1}^{-1/2}\mathbf{m}_{t-1} \\
&= \frac{1}{1-\beta_1}\eta\hat{\mathbf{V}}_t^{-1/2}\left[\beta_1\mathbf{m}_{t-1} + (1-\beta_1)\tilde{\Delta}_t\right] - \frac{\beta_1}{1-\beta_1}\eta\hat{\mathbf{V}}_{t-1}^{-1/2}\mathbf{m}_{t-1} \\
&= \eta\widehat{\mathbf{V}}_t^{-1/2}\Delta_t - \eta\frac{\beta_1}{1-\beta_1}\left(\widehat{\mathbf{V}}_{t-1}^{-1/2} - \widehat{\mathbf{V}}_t^{-1/2}\right)\mathbf{m}_{t-1}.
\end{aligned}$$

Since $f$ is $L$-smooth, taking conditional expectation at time $t$, we get

$$\mathbb{E}[f(\mathbf{z}_{t+1})] - f(\mathbf{z}_t)$$

$$\leq \mathbb{E}[\langle \nabla f(\mathbf{z}_t), \mathbf{z}_{t+1} - \mathbf{z}_t \rangle] + \frac{L}{2}\mathbb{E}[\|\mathbf{z}_{t+1} - \mathbf{z}_t\|^2]$$

$$\leq \mathbb{E}\left[\left\langle \nabla f(\mathbf{z}_t), \eta\widehat{\mathbf{V}}_t^{-1/2}\tilde{\Delta}_t \right\rangle\right] - \mathbb{E}\left[\left\langle \nabla f(\mathbf{z}_t), \eta\frac{\beta_1}{1-\beta_1}\left(\widehat{\mathbf{V}}_{t-1}^{-1/2} - \widehat{\mathbf{V}}_t^{-1/2}\right)\mathbf{m}_{t-1}\right\rangle\right]$$

$$+ \frac{\eta^2 L}{2}\mathbb{E}\left[\left\|\widehat{\mathbf{V}}_t^{-1/2}\tilde{\Delta}_t - \frac{\beta_1}{1-\beta_1}\left(\widehat{\mathbf{V}}_{t-1}^{-1/2} - \widehat{\mathbf{V}}_t^{-1/2}\right)\mathbf{m}_{t-1}\right\|^2\right]$$

$$= \underbrace{\mathbb{E}\left[\left\langle \nabla f(\theta_t), \eta\widehat{\mathbf{V}}_t^{-1/2}\tilde{\Delta}_t \right\rangle\right]}_{I_1} \underbrace{- \eta\mathbb{E}\left[\left\langle \nabla f(\mathbf{z}_t), \frac{\beta_1}{1-\beta_1}\left(\widehat{\mathbf{V}}_{t-1}^{-1/2} - \widehat{\mathbf{V}}_t^{-1/2}\right)\mathbf{m}_{t-1}\right\rangle\right]}_{I_2}$$

$$+ \underbrace{\frac{\eta^2 L}{2}\mathbb{E}\left[\left\|\widehat{\mathbf{V}}_t^{-1/2}\tilde{\Delta}_t - \frac{\beta_1}{1-\beta_1}\left(\widehat{\mathbf{V}}_{t-1}^{-1/2} - \widehat{\mathbf{V}}_t^{-1/2}\right)\mathbf{m}_{t-1}\right\|^2\right]}_{I_3} + \underbrace{\mathbb{E}\left[\left\langle \nabla f(\mathbf{z}_t) - \nabla f(\theta_t), \eta\widehat{\mathbf{V}}_t^{-1/2}\tilde{\Delta}_t \right\rangle\right]}_{I_4}.$$

$$\text{(C.2)}$$

Recall the notation $\widehat{\mathbf{V}}_t = \text{diag}(\widehat{\mathbf{v}}_t) = \text{diag}(\max(\widehat{\mathbf{v}}_{t-1}, \mathbf{v}_t, \epsilon))$.

**Bounding** $I_1$ :

$$I_1 = \mathbb{E}\left[\left\langle \nabla f(\theta_t), \eta\frac{\tilde{\Delta}_t}{\sqrt{\widehat{\mathbf{v}}_t}} \right\rangle\right]$$

$$\leq \eta\mathbb{E}\left[\left\langle \nabla f(\theta_t), \frac{\sqrt{2}\cdot\tilde{\Delta}_t}{\sqrt{\mathbf{v}_t + \epsilon}} \right\rangle\right]$$

$$= \sqrt{2}\eta\mathbb{E}\left[\left\langle \nabla f(\theta_t), \frac{\tilde{\Delta}_t}{\sqrt{\beta_2\mathbf{v}_{t-1} + \epsilon}} \right\rangle\right] + \sqrt{2}\eta\mathbb{E}\left[\left\langle \nabla f(\theta_t), \frac{\tilde{\Delta}_t}{\sqrt{\mathbf{v}_t + \epsilon}} - \frac{\tilde{\Delta}_t}{\sqrt{\beta_2\mathbf{v}_{t-1} + \epsilon}} \right\rangle\right],$$

$$\text{(C.3)}$$

where the first inequality follows by the fact that $\widehat{\mathbf{v}}_t \geq \frac{\mathbf{v}_t + \epsilon}{2}$. For the second term in C.2, then

$$\sqrt{2}\eta\mathbb{E}\left[\left\langle \nabla f(\theta_t), \frac{\tilde{\Delta}_t}{\sqrt{\mathbf{v}_t + \epsilon}} - \frac{\tilde{\Delta}_t}{\sqrt{\beta_2\mathbf{v}_{t-1} + \epsilon}} \right\rangle\right]$$

$$\leq \sqrt{2}\eta\mathbb{E}\|\nabla f(\theta_t)\|\mathbb{E}\left[\left\|\frac{1}{\sqrt{\mathbf{v}_t + \epsilon}} - \frac{1}{\sqrt{\beta_2\mathbf{v}_{t-1} + \epsilon}}\right\| \cdot \|\tilde{\Delta}_t\|\right]$$

$$\leq \frac{\eta\sqrt{2(1-\beta_2)}G}{\epsilon}\mathbb{E}[\|\tilde{\Delta}_t\|^2], \qquad\qquad\qquad \text{(C.4)}$$

where the second inequality follows from Lemma F.1 and F.5, and we will further apply the bound for $\mathbb{E}[\|\tilde{\Delta}_t\|^2]$ following

$$\sqrt{2}\eta\mathbb{E}\left[\left\langle \nabla f(\theta_t), \frac{\tilde{\Delta}_t}{\sqrt{\beta_2\mathbf{v}_{t-1}+\epsilon}} \right\rangle\right]$$

$$= \sqrt{2}\eta\mathbb{E}\left[\left\langle \frac{\nabla f(\theta_t)}{\sqrt{\beta_2\mathbf{v}_{t-1}+\epsilon}}, \tilde{\Delta}_t + \eta K\nabla f(\theta_t) - \eta_l K\nabla f(\theta_t) \right\rangle\right]$$

$$= -\sqrt{2}\eta\eta_t K\mathbb{E}\left[\left\| \frac{\nabla f(\theta_t)}{\sqrt[4]{\beta_2\mathbf{v}_{t-1}+\epsilon}} \right\|^2\right] + \sqrt{2}\eta\mathbb{E}\left[\left\langle \frac{\nabla f(\theta_t)}{\sqrt{\beta_2\mathbf{v}_{t-1}+\epsilon}}, \tilde{\Delta}_t + \eta_l K\nabla f(\theta_t) \right\rangle\right]$$

$$= -\sqrt{2}\eta\eta_h K\mathbb{E}\left[\left\| \frac{\nabla f(\theta_t)}{\sqrt[4]{\beta_2\mathbf{v}_{t-1}+\epsilon}} \right\|^2\right] + \sqrt{2}\eta\left\langle \frac{\nabla f(\theta_t)}{\sqrt{\beta_2\mathbf{v}_{t-1}+\epsilon}}, \mathbb{E}\left[-\frac{1}{m}\sum_{i=1}^{m}\sum_{k=0}^{K-1}\eta\mathbf{g}_{t,k}^i + \eta_l K\nabla f(\theta_t) - \frac{1}{M_t}\sum_{i\in M_t} q_t^i\right] \right\rangle$$

$$= -\sqrt{2}\eta\eta_n K\mathbb{E}\left[\left\| \frac{\nabla f(\theta_t)}{\sqrt[4]{\beta_2\mathbf{v}_{t-1}+\epsilon}} \right\|^2\right] + \sqrt{2}\eta\left\langle \frac{\nabla f(\theta_t)}{\sqrt{\beta_2\mathbf{v}_{t-1}+\epsilon}}, \mathbb{E}\left[-\frac{\eta_l}{m}\sum_{i=1}^{m}\sum_{k=0}^{K-1}\mathbf{g}_{t,k}^i + \frac{\eta_l K}{m}\sum_{i=1}^{m}\nabla F_i(\theta_t) - \frac{1}{M_t}\sum_{i\in M_t} q_t^i\right] \right\rangle,$$

$$\text{(C.5)}$$

where the third equality follows the local update rule. For the last term in C.5, we get

$$\sqrt{2}\eta\left\langle \frac{\nabla f(\theta_t)}{\sqrt{\beta_2\mathbf{v}_{t-1}+\epsilon}}, \mathbb{E}\left[-\frac{\eta_l}{m}\sum_{i=1}^{m}\sum_{k=0}^{K-1}\mathbf{g}_{t,k}^i + \frac{\eta_l K}{m}\sum_{i=1}^{m}\nabla F_i(\theta_t)\right]\right\rangle + \sqrt{2}\eta\mathbb{E}\left\langle \frac{\nabla f(\theta_t)}{\sqrt{\beta_2\mathbf{v}_{t-1}+\epsilon}}, -\frac{1}{M_c}\sum_{i\in M_c} q_t^i\right\rangle$$

$$= \sqrt{2}\eta\left\langle \frac{\sqrt{\eta_l K}}{\sqrt[4]{\beta_2\mathbf{v}_{t-1}+\epsilon}}\nabla f(\theta_t), -\frac{\sqrt{\eta_l K}}{Km}\frac{1}{\sqrt[4]{\beta_2\mathbf{v}_{t-1}+\epsilon}}\mathbb{E}\left[\sum_{i=1}^{m}\sum_{k=0}^{K-1}(\nabla F_i(\theta_{t,k}^i) - \nabla F_i(\theta_t))\right]\right\rangle$$

$$+ \sqrt{2}\eta\mathbb{E}\left\langle \frac{\nabla f(\theta_t)}{\sqrt{\beta_2\mathbf{v}_{t-1}+\epsilon}}, -\frac{1}{M_t}\sum_{i\in M_t} q_t^i\right\rangle$$

$$= \frac{\sqrt{2}\eta\eta_l K}{2}\mathbb{E}\left[\left\| \frac{\nabla f(\theta_t)}{\sqrt[4]{\beta_2\mathbf{v}_{t-1}+\epsilon}} \right\|^2\right] + \frac{\sqrt{2}\eta\eta_l}{2Km^2}\mathbb{E}\left[\left\| \frac{1}{\sqrt[4]{\beta_2\mathbf{v}_{t-1}+\epsilon}}\sum_{i=1}^{m}\sum_{k=0}^{K-1}(\nabla F_i(\theta_{t,k}^i) - \nabla F_i(\theta_t)) \right\|^2\right]$$

$$- \frac{\sqrt{2}\eta\eta_l}{2Km^2}\mathbb{E}\left[\left\| \frac{1}{\sqrt[4]{\beta_2\mathbf{v}_{t-1}+\epsilon}}(\sum_{i=1}^{m}\sum_{k=0}^{K-1}\nabla F_i(\theta_{t,k}^i)) \right\|^2\right] + \sqrt{2}\eta\mathbb{E}\left\langle \frac{\nabla f(\theta_t)}{\sqrt{\beta_2\mathbf{v}_{t-1}+\epsilon}}, -\frac{1}{M_t}\sum_{i\in M_t} q_t^i\right\rangle$$

$$\leq \frac{\sqrt{2}\eta\eta_t K}{2}\mathbb{E}\left[\left\| \frac{\nabla f(\theta_l)}{\sqrt[4]{\beta_2\mathbf{v}_{t-1}+\epsilon}} \right\|^2\right] + \frac{\sqrt{2}\eta\eta_t}{2m}\sum_{i=1}^{m}\sum_{k=0}^{K-1}\mathbb{E}\left[\left\| \frac{\nabla F_i(\theta_{t,k}^i) - \nabla F_i(\theta_t)}{\sqrt[4]{\beta_2\mathbf{v}_{t-1}+\epsilon}} \right\|^2\right]$$

$$- \frac{\sqrt{2}\eta\eta_l}{2Km^2}\mathbb{E}\left[\left\| \frac{1}{\sqrt[4]{\beta_2\mathbf{v}_{t-1}+\epsilon}}(\sum_{i=1}^{m}\sum_{k=0}^{K-1}\nabla F_i(\theta_{t,k}^i)) \right\|^2\right] + \sqrt{2}\eta\mathbb{E}\left\langle \frac{\nabla f(\theta_t)}{\sqrt{\beta_2\mathbf{v}_{t-1}+\epsilon}}, -\frac{1}{M_t}\sum_{i\in M_t} q_t^i\right\rangle$$

$$\leq \frac{\sqrt{2}\eta\eta_t K}{2}\mathbb{E}\left[\left\| \frac{\nabla f(\theta_t)}{\sqrt[4]{\beta_2\mathbf{v}_{t-1}+\epsilon}} \right\|^2\right] + \frac{\sqrt{2}\eta\eta_t}{2m}\sum_{i=1}^{m}\sum_{k=0}^{K-1}\mathbb{E}\left[\left\| \frac{\theta_{t,k}^i - \theta_t}{\sqrt[4]{\beta_2\mathbf{v}_{t-1}+\epsilon}} \right\|^2\right]$$

$$- \frac{\sqrt{2}\eta\eta_l}{2Km^2}\mathbb{E}\left[\left\| \frac{1}{\sqrt[4]{\beta_2\mathbf{v}_{t-1}+\epsilon}}(\sum_{i=1}^{m}\sum_{k=0}^{K-1}\nabla F_i(\theta_{t,k}^i)) \right\|^2\right] + \sqrt{2}\eta\mathbb{E}\left\langle \frac{\nabla f(\theta_t)}{\sqrt{\beta_2\mathbf{v}_{t-1}+\epsilon}}, -\frac{1}{M_t}\sum_{i\in M_t} q_t^i\right\rangle,$$

$$\text{(C.6)}$$

where the second equation follows from $\langle x, y\rangle = \frac{1}{2}[\|x\|^2 + \|y\|^2 - \|x-y\|^2]$, the first inequality holds by applying Cauchy-Schwarz inequality, the second inequality follows from $\langle x, y\rangle \leq \frac{1}{2}\|x\|^2 + \|y\|^2$, the third inequality follows from Assumption 3.1.

Hence by applying Lemma F.14 with the local learning rate condition: $\eta_l \leq \frac{1}{8KL}$ ,then

$$
\sqrt{2} \cdot \eta \left\langle \frac{\nabla f(\theta_t)}{\sqrt{\beta_2 \mathbf{v}_{t-1} + \epsilon}}, \mathbb{E}\left[ -\frac{\eta_l}{m} \sum_{i=1}^{m} \sum_{k=0}^{K-1} \mathbf{g}_{t,k}^i + \frac{\eta_l K}{m} \sum_{i=1}^{m} \nabla F_i(\theta_t) \right] \right\rangle
$$

$$
\leq \frac{3\sqrt{2}\eta\eta_l K}{4} \mathbb{E}\left[ \left\| \frac{\nabla f(\theta_t)}{\sqrt[4]{\beta_2 \mathbf{v}_{t-1} + \epsilon}} \right\|^2 \right] + \frac{5\eta\eta_l^3 K^2 L^2}{\sqrt{2\epsilon}}(\sigma_l^2 + 6K\sigma_g^2)
$$

$$
- \frac{\sqrt{2}\eta\eta_l}{2Km^2} \mathbb{E}\left[ \left\| \frac{1}{\sqrt[4]{\beta_2 \mathbf{v}_{t-1} + \epsilon}} (\sum_{i=1}^{m} \sum_{k=0}^{K-1} \nabla F_i(\theta_{t,k}^i)) \right\|^2 \right] + \sqrt{2}\eta\mathbb{E}\left\langle \frac{\nabla f(\theta_t)}{\sqrt{\beta_2 \mathbf{v}_{t-1} + \epsilon}}, -\frac{1}{M_t} \sum_{i\in M_t} q_t^i \right\rangle.
$$

$$\tag{C.7}$$

Then merging pieces together,

$$
I_1 \leq -\frac{\sqrt{2}\eta\eta_l K}{4} \mathbb{E}\left[ \left\| \frac{\nabla f(\theta_t)}{\sqrt[4]{\beta_2 \mathbf{v}_{t-1} + \epsilon}} \right\|^2 \right] + \frac{5\eta\eta_l^3 K^2 L^2}{\sqrt{2\epsilon}}(\sigma_l^2 + 6K\sigma_g^2)
$$

$$
- \frac{\sqrt{2}\eta\eta_l}{2Km^2} \mathbb{E}\left[ \left\| \frac{1}{\sqrt[4]{\beta_2 \mathbf{v}_{t-1} + \epsilon}} \sum_{i=1}^{m} \sum_{k=0}^{K-1} \nabla F_i(\theta_{t,k}^i) \right\|^2 \right] + \frac{\eta\sqrt{2(1-\beta_2)}G}{\epsilon}\mathbb{E}[\|\tilde{\Delta}_t\|^2]
$$

$$
\leq -\frac{\eta\eta_l K - 2\sqrt{2}\eta}{4} \mathbb{E}\left[ \left\| \frac{\nabla f(\theta_t)}{\sqrt[4]{\beta_2 \mathbf{v}_{t-1} + \epsilon}} \right\|^2 \right] + \frac{5\eta\eta_l^3 K^2 L^2}{\sqrt{2\epsilon}}(\sigma_l^2 + 6K\sigma_g^2)
$$

$$
- \frac{\sqrt{2}\eta\eta_l}{2Km^2} \mathbb{E}\left[ \left\| \frac{1}{\sqrt[4]{\beta_2 \mathbf{v}_{t-1} + \epsilon}} \sum_{i=1}^{m} \sum_{k=0}^{K-1} \nabla F_i(\theta_{t,k}^i) \right\|^2 \right] + \sqrt{2}\eta\mathbb{E}\left\langle \frac{\nabla f(\theta_t)}{\sqrt{\beta_2 \mathbf{v}_{t-1} + \epsilon}}, -\frac{1}{M_t} \sum_{i\in M_t} q_t^i \right\rangle.
$$

$$\tag{C.8}$$

**Bounding $I_2$** : The bound for $I_2$ mainly follows by the update rule and definition of virtual sequence $\mathbf{z}_t$,

$$
I_2 = -\eta\mathbb{E}\left[ \left\langle \nabla f(\mathbf{z}_t), \frac{\beta_1}{1-\beta_1}\left( \widehat{\mathbf{V}}_{t-1}^{-1/2} - \widehat{\mathbf{V}}_t^{-1/2} \right) \mathbf{m}_{t-1} \right\rangle \right]
$$

$$
= -\eta\mathbb{E}\left[ \left\langle \nabla f(\mathbf{z}_t) - \nabla f(\theta_t) + \nabla f(\theta_t), \frac{\beta_1}{1-\beta_1}\left( \widehat{\mathbf{V}}_{t-1}^{-1/2} - \widehat{\mathbf{V}}_t^{-1/2} \right) \mathbf{m}_{t-1} \right\rangle \right]
$$

$$
\leq \eta\mathbb{E}\left[ \|\nabla f(\theta_t)\| \left\| \frac{\beta_1}{1-\beta_1}\left( \widehat{\mathbf{V}}_{t-1}^{-1/2} - \widehat{\mathbf{V}}_t^{-1/2} \right) \mathbf{m}_{t-1} \right\| \right]
$$

$$
+ \eta^2 L\mathbb{E}\left[ \left\| \frac{\beta_1}{1-\beta_1} \widehat{\mathbf{V}}_{t-1}^{-1/2} \mathbf{m}_{t-1} \right\| \left\| \frac{\beta_1}{1-\beta_1}\left( \hat{\mathbf{V}}_{t-1}^{-1/2} - \hat{\mathbf{V}}_t^{-1/2} \right) \mathbf{m}_{t-1} \right\| \right]
$$

$$
\leq \eta\frac{\beta_1}{1-\beta_1}\eta KG^2\mathbb{E}\left[ \left\| \widehat{\mathbf{V}}_{t-1}^{-1/2} - \widehat{\mathbf{V}}_t^{-1/2} \right\|_1 \right] + \eta^2 \frac{\beta_1^2}{(1-\beta_1)^2}L\eta_l^2 K^2 G^2 \epsilon^{-1/2}\mathbb{E}\left[ \left\| \widehat{\mathbf{V}}_{t-1}^{-1/2} - \widehat{\mathbf{V}}_t^{-1/2} \right\|_1 \right],
$$

$$\tag{C.9}$$

where the last inequality holds by applying Lemma F.5 and the fact of $\widehat{\mathbf{v}}_{t-1} \geq \epsilon$.

**Bounding $I_3$**: It can be bounded as follows:

$$I_3 = \frac{\eta^2 L}{2} \mathbb{E}\left[\left\|\widehat{\mathbf{V}}_t^{-1/2}\tilde{\Delta}_t + \frac{\beta_1}{1-\beta_1}\left(\widehat{\mathbf{V}}_{t-1}^{-1/2} - \widehat{\mathbf{V}}_t^{-1/2}\right)\mathbf{m}_{t-1}\right\|^2\right]$$

$$\leq \eta^2 L\mathbb{E}\left[\left\|\widehat{\mathbf{V}}_t^{-1/2}\tilde{\Delta}_t\right\|^2\right] + \eta^2 L\mathbb{E}\left[\left\|\frac{\beta_1}{1-\beta_1}\left(\widehat{\mathbf{V}}_{t-1}^{-1/2} - \widehat{\mathbf{V}}_t^{-1/2}\right)\mathbf{m}_{t-1}\right\|^2\right]$$

$$\leq \eta^2 L\mathbb{E}\left[\left\|\widehat{\mathbf{V}}_t^{-1/2}\tilde{\Delta}_t\right\|^2\right] + \eta^2 L\frac{\beta_1^2}{(1-\beta_1)^2}\eta^2 K^2 G^2\mathbb{E}\left[\left\|\widehat{\mathbf{V}}_{t-1}^{-1/2} - \widehat{\mathbf{V}}_t^{-1/2}\right\|^2\right], \qquad \text{(C.10)}$$

where the first inequality follows by Cauchy-Schwarz inequality, and the second one follows by Lemma F.5.

**Bounding $I_4$ :**

$$I_4 = \mathbb{E}\left[\left\langle \nabla f(\mathbf{z}_t) - \nabla f(\mathbf{x}_t), \eta\widehat{\mathbf{V}}_t^{-1/2}\Delta_t\right\rangle\right]$$

$$\leq \mathbb{E}\left[\|\nabla f(\mathbf{z}_t) - \nabla f(\mathbf{x}_t)\|\|\eta\widehat{\mathbf{V}}_t^{-1/2}\Delta_t\|\right]$$

$$\leq L\mathbb{E}\left[\|\mathbf{z}_t - \mathbf{x}_t\|\|\eta\widehat{\mathbf{V}}_t^{-1/2}\Delta_t\|\right]$$

$$\leq \frac{\eta^2 L}{2}\mathbb{E}\left[\left\|\widehat{\mathbf{V}}_t^{-1/2}\Delta_t\right\|^2\right] + \frac{\eta^2 L}{2}\mathbb{E}\left[\left\|\frac{\beta_1}{1-\beta_1}\widehat{\mathbf{V}}_{t-1}^{-1/2}\mathbf{m}_{t-1}\right\|^2\right],$$

where the first inequality holds by the fact of $\langle \mathbf{a}, \mathbf{b}\rangle \leq \|\mathbf{a}\|\|\mathbf{b}\|$, the second one follows from Assumption 3.1 and the third one holds by the definition of virtual sequence $\mathbf{z}_t$ and the fact of $\|\mathbf{a}\|\|\mathbf{b}\| \leq \frac{1}{2}\|\mathbf{a}\|^2 + \frac{1}{2}\|\mathbf{b}\|^2$. Then summing $I_4$ over $t = 1, \cdots, T$, then

$$\sum_{t=1}^T I_4 \leq \frac{\eta^2 L}{2\epsilon}\sum_{t=1}^T \mathbb{E}[\|\tilde{\Delta}_t\|^2] + \frac{\eta^2 L}{2\epsilon}\sum_{t=1}^T \mathbb{E}\left[\left\|\frac{\beta_1}{1-\beta_1}\mathbf{m}_t\right\|^2\right]$$

$$\leq \frac{\eta^2 L}{2\epsilon}\sum_{t=1}^T \mathbb{E}[\|\tilde{\Delta}_t\|^2] + \frac{\eta^2 L}{2\epsilon}\frac{\beta_1^2}{(1-\beta_1)^2}\sum_{t=1}^T \mathbb{E}[\|\mathbf{m}_t\|^2]. \qquad \text{(C.11)}$$

By Lemma F.10,

$$\sum_{t=1}^T \mathbb{E}[\|\mathbf{m}_t\|^2] \leq \frac{TK\eta_l^2}{m}\sigma_l^2 + \frac{2\eta_l^2}{m^2}\sum_{t=1}^T \mathbb{E}\left[\left\|\sum_{i=1}^m\sum_{k=0}^{K-1}\nabla F_i(\theta_{t,k}^i)\right\|^2\right] + \frac{2}{m^2}\sum_{t=1}^T \mathbb{E}\left\|\frac{1}{M_t}\sum_{i\in M_t}q_t^i\right\|^2$$

The summation of $I_4$ term is bounded by

$$\sum_{t=1}^T I_4 \leq \frac{\eta^2 L}{2\epsilon}\sum_{t=1}^T \mathbb{E}[\|\tilde{\Delta}_t\|^2] + \frac{2\beta_1^2}{(1-\beta_1)^2}\frac{\eta^2 L}{2\epsilon}\frac{\eta_l^2}{m^2}\sum_{t=1}^T \mathbb{E}\left\|\sum_{i=1}^m\sum_{k=0}^{K-1}\nabla F_i(\theta_{t,k}^i)\right\|^2$$

$$+ \frac{\beta_1^2}{(1-\beta_1)^2}\frac{\eta^2 L}{2\epsilon}\frac{TK\eta_l^2}{m}\sigma_l^2 + \frac{2\beta_1^2}{(1-\beta_1)^2}\frac{\eta^2 L}{2\epsilon}\frac{1}{m^2}\sum_{t=1}^T \mathbb{E}\left\|\frac{1}{M_t}\sum_{i\in M_t}q_t^i\right\|^2. \qquad \text{(C.12)}$$

Merging pieces together: Substituting (C.8), (C.9) and (C.10) into (C.2), summing over from $t = 1$ to $T$ and then adding (C.11), then

$$\mathbb{E}[f(\mathbf{z}_{T+1})] - f(\mathbf{z}_1) = \sum_{t=1}^{T}[I_1 + I_2 + I_3 + I_4]$$

$$\leq -\frac{\eta\eta_l K - 2\sqrt{2}\eta}{4}\sum_{t=1}^{T}\mathbb{E}\left[\left\|\frac{\nabla f(\theta_t)}{\sqrt[4]{\beta_2 \mathbf{v}_{t-1} + \epsilon}}\right\|^2\right] + \frac{5\eta\eta_l^3 K^2 L^2 T}{\sqrt{2\epsilon}}(\sigma_l^2 + 6K\sigma_g^2) + \frac{\sqrt{2(1-\beta_2)}\eta G}{\epsilon}\sum_{t=1}^{T}\mathbb{E}[\|\tilde{\Delta}_l\|^2]$$

$$- \frac{\eta\eta_l}{2Km^2}\sum_{t=1}^{T}\mathbb{E}\left[\left\|\frac{1}{\sqrt[4]{\beta_2 \mathbf{v}_{t-1} + \epsilon}}\sum_{i=1}^{m}\sum_{k=0}^{K-1}\nabla F_i(\theta_{t,k}^i))\right\|^2\right]$$

$$+ \frac{\beta_1}{1-\beta_1}\eta\eta_l KG^2\sum_{t=1}^{T}\mathbb{E}\left[\left\|\widehat{\mathbf{V}}_{t-1}^{-1/2} - \widehat{\mathbf{V}}_{t}^{-1/2}\right\|_1\right] + \frac{\beta_1^2}{(1-\beta_1)^2}\frac{\eta^2\eta_l^2 K^2 G^2}{\sqrt{\epsilon}}\sum_{t=1}^{T}\mathbb{E}\left[\left\|\widehat{\mathbf{V}}_{t-1}^{-1/2} - \widehat{\mathbf{V}}_{t}^{-1/2}\right\|_1\right]$$

$$+ \frac{\beta_1^2}{(1-\beta_1)^2}\eta^2\eta_n^2 K^2 LG^2\sum_{t=1}^{T}\mathbb{E}\left[\left\|\widehat{\mathbf{V}}_{t-1}^{-1/2} - \widehat{\mathbf{V}}_{t}^{-1/2}\right\|^2\right] + \eta^2 L\sum_{t=1}^{T}\mathbb{E}\left[\left\|\widehat{\mathbf{V}}_{t}^{-1/2}\tilde{\Delta}_t\right\|^2\right]$$

$$+ \frac{\eta^2 L}{2\epsilon}\sum_{t=1}^{T}\mathbb{E}[\|\tilde{\Delta}_t\|^2] + \frac{\eta^2 L}{2\epsilon}\frac{\beta_1^2}{(1-\beta_1)^2}\sum_{t=1}^{T}\mathbb{E}[\|\mathbf{m}_t\|^2]. \tag{C.13}$$

By applying Lemma F.6 is into all terms containing the second moment estimate of model difference $\tilde{\Delta}_t$ in (C.13),and the fact that$(\sqrt{\beta_2 K^2 G^2 + \epsilon})^{-1}\|\theta\| \leq (\sqrt{\beta_2\eta_l^2 K^2 G^2 + \epsilon})^{-1}\|\theta\| \leq \|\frac{\theta}{\sqrt{\beta_2 \mathbf{v}_t + \epsilon}}\| \leq \epsilon^{-1/2}\|\theta\|$ , then

$$\mathbb{E}[f(\mathbf{z}_{T+1})] - f(\mathbf{z}_1)$$

$$\leq -\frac{\eta\eta_t K - 2\sqrt{2}\eta}{4}\sum_{t=1}^{T}\mathbb{E}\left[\left\|\frac{\nabla f(\theta_t)}{\sqrt[4]{\beta_2 \mathbf{v}_{t-1} + \epsilon}}\right\|^2\right] + \frac{5\eta\eta_t^3 K^2 L^2 T}{\sqrt{2\epsilon}}(\sigma_l^2 + 6K\sigma_g^2) + \frac{\beta_1}{1-\beta_1}\frac{\eta\eta KG^2 d}{\sqrt{\epsilon}}$$

$$+ \frac{\beta_1^2}{(1-\beta_1)^2}\frac{2\eta^2\eta_l^2 K^2 LG^2 d}{\epsilon} - \frac{2\eta\eta_l}{2Km^2}\sum_{t=1}^{T}\mathbb{E}\left[\left\|\frac{1}{\sqrt[4]{\beta_2 \mathbf{v}_{t-1} + \epsilon}}\sum_{i=1}^{m}\sum_{k=0}^{K-1}\nabla F_i(\theta_{t,k}^i))\right\|^2\right]$$

$$+ \left(\eta^2 L + \frac{\eta^2 L}{2} + \sqrt{2(1-\beta_2)}\eta G\right)\left[\frac{KT\eta_l^2}{m\epsilon}\sigma_l^2 + \frac{2\eta_l^2}{m^2\epsilon}\sum_{t=1}^{T}\mathbb{E}\left[\left\|\sum_{i=1}^{m}\sum_{k=0}^{K-1}\nabla F_i(\theta_{t,k}^i)\right\|^2\right]\right]$$

$$+ \frac{2\beta_1^2}{(1-\beta_1)^2}\frac{\eta^2 L}{2\epsilon}\frac{\eta_l^2}{m^2}\sum_{t=1}^{T}\mathbb{E}\left[\left\|\sum_{i=1}^{m}\sum_{k=0}^{K-1}\nabla F_i(\theta_{i,k}^i)\right\|^2\right] + \frac{\beta_1^2}{(1-\beta_1)^2}\frac{\eta^2 L}{2\epsilon}\frac{TK\eta_l^2}{m}\sigma_l^2$$

$$+ \left(\eta^2 L + \frac{\eta^2 L}{2} + \sqrt{2(1-\beta_2)} + \eta G\frac{\beta_1^2}{(1-\beta_1)^2}\frac{\eta^2 L}{2}\right)\frac{2}{m^2\epsilon}\sum_{t=1}^{T}\mathbb{E}\left[\left\|\frac{1}{M_t}\sum_{i\in M_t}q_t^i\right\|^2\right]$$

$$+ \sqrt{2}\eta\sum_{t=1}^{T}\mathbb{E}\left\langle\frac{\nabla f(\theta_t)}{\sqrt{\beta_2 \mathbf{v}_{t-1} + \epsilon}}, -\frac{1}{M_t}\sum_{i\in M_t}q_t^i\right\rangle$$

$$\leq -\frac{\eta\eta_l K - 2\sqrt{2}\eta}{4\sqrt{\beta_2\eta_l^2 K^2 G^2 + \epsilon}}\sum_{t=1}^{T}\mathbb{E}[\|\nabla f(\theta_t)\|^2] + \frac{5\eta\eta_l^3 K^2 L^2 T}{\sqrt{2\epsilon}}(\sigma_l^2 + 6K\sigma_q^2)$$

$$+ \frac{\beta_1}{1-\beta_1}\frac{\eta\eta KG^2 d}{\sqrt{\epsilon}} + \frac{\beta_1^2}{(1-\beta_1)^2}\frac{2\eta^2\eta_t^2 K^2 LG^2 d}{\epsilon}$$

$$+ \left( \eta^2 L + \frac{\eta^2 L}{2} + \sqrt{2(1-\beta_2)} \eta G + \frac{\beta_1^2}{(1-\beta_1)^2} \frac{\eta^2 L}{2} \right) \frac{KT\eta_l^2}{m\epsilon} \sigma_l^2 - \sum_{t=1}^{T} \mathbb{E} \left[ \left\| \sum_{i=1}^{m} \sum_{k=0}^{K-1} \nabla F_i(\theta_{t,k}^i) \right\|^2 \right]$$

$$\cdot \left[ \frac{\eta \eta_l}{2\sqrt{\beta_2 K^2 G^2 + \epsilon} Km^2} - \left( \eta^2 L + \frac{\eta^2 L}{2} + \eta\sqrt{2(1-\beta_2)} G + \frac{\beta_1^2}{(1-\beta_1)^2} \frac{\eta^2 L}{2} \right) \frac{2\eta_l^2}{m^2\epsilon} \right]$$

$$+ \left( \eta^2 L + \frac{\eta^2 L}{2} + \sqrt{2(1-\beta_2)} \eta G + \frac{\beta_1^2}{(1-\beta_1)^2} \frac{\eta^2 L}{2} \right) \frac{T2K^2C^2\eta_l^2 G^2}{\alpha^2 m^2 \epsilon} + \sqrt{2}\eta T \mathbb{E} \left\| \frac{\nabla f(\theta_t)}{\sqrt{\beta_2 \mathbf{v}_{t-1} + \epsilon}} \right\| \cdot \mathbb{E} \left\| - \frac{1}{M_t} \sum_{i \in M_t} q_t^i \right\|$$

$$\leq -\frac{\eta \eta_l K - 2\sqrt{2}\eta}{4\sqrt{\beta_2 \eta_l^2 K^2 G^2 + \epsilon}} \sum_{t=1}^{T} \mathbb{E}[\|\nabla f(\theta_t)\|^2] + \frac{5\eta \eta_l^3 K^2 L^2 T}{\sqrt{2\epsilon}} (\sigma_l^2 + 6K\sigma_g^2)$$

$$+ \frac{\beta_1}{1-\beta_1} \frac{\eta \eta_l K G^2 d}{\sqrt{\epsilon}} + \frac{\beta_1^2}{(1-\beta_1)^2} \frac{2\eta^2 \eta_l^2 K^2 L G^2 d}{\epsilon} + \frac{\sqrt{2}\eta \eta_l K^2 G^2}{2Km^2 \sqrt{\beta_2 \eta_l^2 K^2 G^2 + \epsilon}}$$

$$+ \left( \eta^2 L + \frac{\eta^2 L}{2} + \eta\sqrt{2(1-\beta_2)} G + \frac{\beta_1^2}{(1-\beta_1)^2} \frac{\eta^2 L}{2} \right) \left( \frac{2KT\eta_l^2}{m\epsilon} \sigma_l^2 \right) + \frac{\sqrt{2}\eta \eta_l K G C T}{\alpha m \epsilon}$$

$$+ \left( \eta^2 L + \frac{\eta^2 L}{2} + \sqrt{2(1-\beta_2)} \eta G + \frac{\beta_1^2}{(1-\beta_1)^2} \frac{\eta^2 L}{2} \right) \frac{2TK^2 C^2 \eta_l^2 G^2}{\alpha^2 m^2 \epsilon}.$$

The last inequality holds due to additional constraint of local learning rate $\eta_l$ with the inequality $\frac{\eta \eta_l}{2\sqrt{\beta_2 K^2 G^2 + \epsilon} Km^2} - (\frac{3\eta^2 L}{2} + 1) \eta\sqrt{2(1-\beta_2)} G + \frac{\beta_1^2}{(1-\beta_1)^2} \frac{\eta^2 L}{2}) \frac{2\eta_l^2}{m^2\epsilon} \geq 0$, thus obtain the constraint $\eta_l \leq \frac{\epsilon}{K\sqrt{\beta_2 K^2 G^2 + \epsilon}[(3+C_1^2)\eta L + 2\sqrt{2(1-\beta_2)}G]}$. Hence

$$\frac{\eta \eta_l K}{4\sqrt{\beta_2 \eta_l^2 K^2 G^2 + \epsilon} \cdot T} \sum_{t=1}^{T} \mathbb{E}[\|\nabla f(\theta_t)\|^2]$$

$$\leq \frac{f(\mathbf{z}_0) - \mathbb{E}[f(\mathbf{z}_T)]}{T} + \frac{5\eta \eta_l^3 K^2 L^2}{\sqrt{2\epsilon}} (\sigma_l^2 + 6K\sigma_g^2) + [\frac{\sqrt{2}\eta}{2\epsilon} + (3+C_1^2)\eta^2 L + 2\sqrt{2(1-\beta_2)}\eta G] (\frac{2K\eta_l^2}{m\epsilon} \sigma_l^2 + \frac{2K^2 C^2 \eta_l^2 G^2}{\alpha^2 m^2 \epsilon})$$

$$+ \frac{C_1 \eta \eta_l K G^2 d}{T\sqrt{\epsilon}} + \frac{2C_1^2 \eta^2 \eta_l^2 K^2 L G^2 d}{T\epsilon} + \frac{\sqrt{2}\eta \eta_l K G C}{\alpha m \epsilon}.$$

Therefore

$$\min \mathbb{E}[\|\nabla f(\theta_t)\|^2] \leq 4\sqrt{\beta_2 \eta_l^2 K^2 G^2 + \epsilon} \cdot \left[ \frac{f_0 - f_*}{(\eta \eta_l K - 2\sqrt{2}\eta)T} + \frac{\Xi}{T} + \Omega \right],$$

where

$$\Xi = \frac{C_1 G^2 d}{\sqrt{\epsilon}} + \frac{2C_1^2 \eta \eta_l K L G^2 d}{\epsilon}$$

, and

$$\Omega = \frac{5\eta_l^2 K^2 L^2}{\sqrt{2\epsilon}} (\sigma_l^2 + 6K\sigma_g^2) + (3+C_1^2)\eta^2 L + 2\sqrt{2(1-\beta_2)}\eta G] (\frac{2\eta_l}{m\eta\epsilon} \sigma_l^2 + \frac{2KC^2 \eta_l G^2}{\alpha^2 \eta m^2 \epsilon}) + \frac{\sqrt{2}GC}{\alpha m \epsilon}$$

,that $C_1 = \frac{\beta_1}{1-\beta_1}$.

The proof of Theorem 4.2 is similar to the above proof procedure and the detailed proof will not be given here.

## C.2 PROOF OF COROLLARY 4.1

If pick $\eta = \Theta(\frac{1}{\sqrt{TK}})$, $\eta = \Theta(\sqrt{Km})$ and $T = \mathcal{O}(Km)$ ,then $\min_{t\in[T]} \mathbb{E}[\|\nabla f(\theta_t)\|^2] = \mathcal{O}(\frac{1}{T})$.

**Assumption C.1.** (Compression Dissimilarity). For the biased compressor, there exists a constant $\xi$ such that, for each iteration $t \geq 0$, that

$$\left\| \mathcal{C}\left( \frac{1}{m} \sum_{i=1}^{m} [\Delta_t^i + \mathbf{e}_t^i] \right) - \frac{1}{m} \sum_{i=1}^{m} \mathcal{C}(\Delta_t^i + \mathbf{e}_t^i) \right\|$$

$$\leq \gamma \left\| \frac{1}{m} \sum_{i=1}^{m} \Delta_t^i \right\| \tag{C.14}$$

and

$$\left\| \frac{1}{M_t} \sum_{i \in M_t} \mathcal{C}(q_t^i) \right\| \leq \lambda \left\| \frac{1}{m} \sum_{i=1}^{m} \Delta_t^i \right\|. \tag{C.15}$$

Here $M_t$ denotes the set of all clients satisfying (3) or (4) at round $t$, and $\mathcal{C}(q_t^i)$ is given in Algorithms 2 and 3. The assumption of bounded gradient is usually adopted in adaptive gradient methods Alistarh et al. (2018).

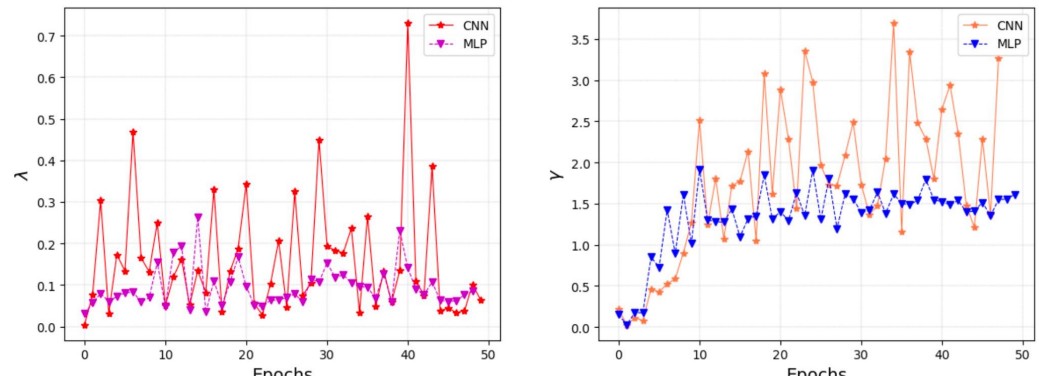

Figure 21: Empirical justification for Assumption C.1. on CNN and MLP based on the Fashion-MINIST datset.

## C.3  PROOF OF THEOREM 4.3

Notations and equations: For partial participation, i.e. $|S_t| = n, \forall t \in [T]$. The global model difference is the average of local model difference from the subset $S_t$, i.e., $\tilde{\Delta}_t = \frac{1}{n} \sum_{i \in \mathcal{S}_t} \tilde{\Delta}_i^t$. Denote $\bar{\Delta}_t = \frac{1}{m} \sum_{i=1}^{m} \tilde{\Delta}_t^i$, and for convenience, we follow the previous notation of $\widehat{\mathbf{V}}_t = \text{diag}(\widehat{\mathbf{v}}_t + \epsilon)$. Next we show that the global model difference $\tilde{\Delta}_t$ is an unbiased estimator of $\tilde{\Delta}_t$;

$$\mathbb{E}_{\mathcal{S}_t}[\tilde{\Delta}_t] = \frac{1}{n} \mathbb{E}_{\mathcal{S}_t}[\sum_{i=1}^{n} \tilde{\Delta}_t^{w_i}] = \mathbb{E}_{\mathcal{S}_t}[\tilde{\Delta}_t^{w_1}] = \frac{1}{m} \sum_{i=1}^{m} \tilde{\Delta}_t^i = \bar{\Delta}_t.$$

Define the virtual sequence $\mathbf{z}_t$ same as previous: assume $\theta_0 = \theta_1$, for each $t \geq 1$, then

$$\mathbf{z}_t = \theta_t + \frac{\beta_1}{1 - \beta_1}(\theta_t - \theta_{t-1}) = \frac{1}{1 - \beta_1}\theta_t - \frac{\beta_1}{1 - \beta_1}\theta_{t-1},$$

$$\mathbf{z}_{t+1} - \mathbf{z}_t = \eta \widehat{\mathbf{V}}_t^{-1/2} \tilde{\Delta}_t - \eta \frac{\beta_1}{1 - \beta_1}\left( \widehat{\mathbf{V}}_{t-1}^{-1/2} - \widehat{\mathbf{V}}_t^{-1/2} \right) \mathbf{m}_{t-1}.$$

By Assumption 3.1, we get

$$\mathbb{E}[f(\mathbf{z}_{t+1})] - f(\mathbf{z}_t)$$

$$\leq \mathbb{E}\left[\left\langle \nabla f(\mathbf{z}_t), \eta \widehat{\mathbf{V}}_t^{-1/2}\tilde{\Delta}_t \right\rangle\right] - \mathbb{E}\left[\left\langle \nabla f(\mathbf{z}_t), \eta \frac{\beta_1}{1-\beta_1}\left(\widehat{\mathbf{V}}_{t-1}^{-1/2} - \widehat{\mathbf{V}}_t^{-1/2}\right)\mathbf{m}_{t-1}\right\rangle\right]$$

$$+ \frac{\eta^2 L}{2}\mathbb{E}\left[\left\|\widehat{\mathbf{V}}_t^{-1/2}\tilde{\Delta}_t - \frac{\beta_1}{1-\beta_1}\left(\widehat{\mathbf{V}}_{t-1}^{-1/2} - \widehat{\mathbf{V}}_t^{-1/2}\right)\mathbf{m}_{t-1}\right\|^2\right]$$

$$= \underbrace{\mathbb{E}\left[\left\langle \nabla f(\theta_t), \eta \widehat{\mathbf{V}}_t^{-1/2}\tilde{\Delta}_t \right\rangle\right]}_{I_1'} \underbrace{- \eta\mathbb{E}\left[\left\langle \nabla f(\mathbf{z}_t), \frac{\beta_1}{1-\beta_1}\left(\widehat{\mathbf{V}}_{t-1}^{-1/2} - \widehat{\mathbf{V}}_t^{-1/2}\right)\mathbf{m}_{t-1}\right\rangle\right]}_{I_2'}$$

$$+ \underbrace{\frac{\eta^2 L}{2}\mathbb{E}\left[\left\|\widehat{\mathbf{V}}_t^{-1/2}\tilde{\Delta}_t - \frac{\beta_1}{1-\beta_1}\left(\widehat{\mathbf{V}}_{t-1}^{-1/2} - \widehat{\mathbf{V}}_t^{-1/2}\right)\mathbf{m}_{t-1}\right\|^2\right]}_{I_3'} + \underbrace{\mathbb{E}\left[\left\langle \nabla f(\mathbf{z}_t) - \nabla f(\theta_t), \eta \widehat{\mathbf{V}}_t^{-1/2}\tilde{\Delta}_t \right\rangle\right]}_{I_4'}.$$

Since $\tilde{\Delta}_t$ is an unbiased estimator of $\bar{\Delta}_t$, the main difference of convergence analysis for partial participation cases is bounding $\mathbb{E}[\|\tilde{\Delta}_t\|^2]$.

The bound for $I_2'$ is exactly the same as the bound for $I_2$. For the corresponding three terms, $I_1'$, $I_3'$ and $I_4'$ which include the second-order momentum estimate of $\tilde{\Delta}_t$. For $I_1'$, then

$$I_1' = \mathbb{E}\left[\left\langle \nabla f(\theta_t), \eta \frac{\tilde{\Delta}_t}{\sqrt{\hat{\mathbf{v}}_t}} \right\rangle\right]$$

$$\leq \eta\mathbb{E}\left[\left\langle \nabla f(\theta_t), \frac{\sqrt{2}\cdot\tilde{\Delta}_t}{\sqrt{\mathbf{v}_t} + \epsilon} \right\rangle\right]$$

$$= \eta\mathbb{E}\left[\left\langle \nabla f(\theta_t), \frac{\sqrt{2}\cdot\tilde{\Delta}_t}{\sqrt{\beta_2 \mathbf{v}_{t-1}} + \epsilon} \right\rangle\right] + \eta\mathbb{E}\left[\left\langle \nabla f(\theta_t), \frac{\sqrt{2}\cdot\tilde{\Delta}_t}{\sqrt{\mathbf{v}_t} + \epsilon} - \frac{\sqrt{2}\cdot\tilde{\Delta}_t}{\sqrt{\beta_2 \mathbf{v}_{t-1}} + \epsilon} \right\rangle\right]. \quad \text{(C.16)}$$

The first term in (C.16) does not change in partial participation scheme. The second term is changed due to the variance of $\tilde{\Delta}_t$ changes. For the second term of $I_1'$,

$$\sqrt{2}\eta\mathbb{E}\left[\left\langle \nabla f(\theta_t), \frac{\tilde{\Delta}_t}{\sqrt{\mathbf{v}_t} + \epsilon} - \frac{\tilde{\Delta}_t}{\sqrt{\beta_2 \mathbf{v}_{t-1}} + \epsilon} \right\rangle\right] \leq \frac{\sqrt{2(1-\beta_2)}\eta G}{\epsilon}\mathbb{E}[\|\tilde{\Delta}_t\|^2]. \quad \text{(C.17)}$$

For $I_3'$,

$$\sum_{t=1}^{T} I_3' \leq \frac{\eta^2 L}{\epsilon}\sum_{t=1}^{T}\mathbb{E}[\|\tilde{\Delta}_t\|^2] + \eta^2 L\frac{\beta_1^2}{(1-\beta_1)^2}\eta_l^2 G^2\sum_{t=1}^{T}\mathbb{E}\left[\left\|\left(\widehat{\mathbf{V}}_{t-1}^{-1/2} - \widehat{\mathbf{V}}_t^{-1/2}\right)\right\|^2\right], \quad \text{(C.18)}$$

and for $I_4'$ similar to (C.11), We get

$$\sum_{t=1}^{T} I_4' \leq \frac{\eta^2 L}{2\epsilon}\sum_{t=1}^{T}\mathbb{E}[\|\tilde{\Delta}_t\|^2] + \frac{\eta^2 L}{2\epsilon}\frac{\beta_1^2}{(1-\beta_1)^2}\sum_{t=1}^{T}\mathbb{E}[\|\mathbf{m}_t\|^2]. \quad \text{(C.19)}$$

From Lemma F.10, then

$$\sum_{t=1}^{T}\mathbb{E}[\|\mathbf{m}_t\|^2] \leq \frac{KT\eta_l^2}{n}\sigma_l^2 + \frac{2\eta_l^2}{n^2}\sum_{t=1}^{T}\mathbb{E}\left[\left\|\sum_{i\in\mathcal{S}_t}\sum_{k=0}^{K-1}\nabla F_i(\theta_{t,k}^i)\right\|^2\right] + \frac{2}{n^2}\sum_{t=1}^{T}\mathbb{E}\left\|\frac{1}{M_t}\sum_{i\in M_t}q_t^i\right\|^2. \quad \text{(C.20)}$$

Then substituting (C.20) into (C.19), then

$$
\begin{aligned}
\sum_{t=1}^{T} I_4' &\leq \frac{\eta^2 L}{2\epsilon} \sum_{t=1}^{T} \mathbb{E}[\|\tilde{\Delta}_t\|^2] + \frac{2\beta_1^2}{(1-\beta_1)^2} \frac{\eta^2 \eta_l^2 L}{2n^2\epsilon} \sum_{t=1}^{T} \mathbb{E}\left[\left\|\sum_{i\in\mathcal{S}_t}\sum_{k=0}^{K-1} \nabla F_i(\theta_{t,k}^i)\right\|^2\right] \\
&\quad + \frac{\beta_1^2}{(1-\beta_1)^2} \frac{\eta^2 \eta_l^2 K T L}{2n\epsilon} \sigma_l^2 + \frac{1}{n^2} \mathbb{E}\left\|\frac{2}{M_t}\sum_{i\in M_t} q_t^i\right\|^2 \\
&\leq \frac{\eta^2 L}{2\epsilon} \sum_{t=1}^{T} \mathbb{E}[\|\tilde{\Delta}_t\|^2] + \frac{2\beta_1^2}{(1-\beta_1)^2} \frac{\eta^2 \eta_l^2 L}{2n^2\epsilon} \sum_{t=1}^{T} \mathbb{E}\left[\left\|\sum_{i=1}^{m}\sum_{k=0}^{K-1} \mathbb{P}\{i\in\mathcal{S}_t\}\nabla F_i(\theta_{t,k}^i)\right\|^2\right] \\
&\quad + \frac{\beta_1^2}{(1-\beta_1)^2} \frac{\eta^2 \eta_l^2 K T L}{2n\epsilon} \sigma_l^2 + \frac{2}{n^2} \sum_{t=1}^{T} \mathbb{E}\left\|\frac{1}{M_t}\sum_{i\in M_t} q_t^i\right\|^2,
\end{aligned}
\tag{C.21}
$$

where we will further apply the bound for $\mathbb{E}[\|\tilde{\Delta}_t\|^2]$ following by Lemma F.8. The second term in (C.21) can be bounded from (F.11)). Therefore, summing up (C.17), (C.18) and (C.9), summing over from $t=1$ to $T$, then adding (C.19),

$$
\begin{aligned}
\mathbb{E}[f(\mathbf{z}_{T+1})] - f(\mathbf{z}_1) &= \sum_{t=1}^{T} [I_1' + I_2 + I_3' + I_4'] \\
&\leq -\frac{\eta\eta_l K}{4} \sum_{t=1}^{T} \mathbb{E}\left[\left\|\frac{\nabla f(\theta_t)}{\sqrt[4]{\beta_2 \mathbf{v}_{t-1} + \epsilon}}\right\|^2\right] + \frac{5\eta\eta_l^3 K^2 L^2 T}{\sqrt{2\epsilon}}(\sigma_l^2 + 6K\sigma_g^2) + \frac{\sqrt{2(1-\beta_2)}\eta G}{\epsilon} \sum_{t=1}^{T} \mathbb{E}[\|\tilde{\Delta}_t\|^2] \\
&\quad - \frac{\eta m}{2Km^2} \sum_{t=1}^{T} \mathbb{E}\left[\left\|\frac{1}{\sqrt[4]{\beta_2 \mathbf{v}_{t-1} + \epsilon}}\sum_{i=1}^{m}\sum_{k=0}^{K-1} \nabla F_i(\theta_{t,k}^i)\right\|^2\right] \\
&\quad + \frac{\beta_1}{1-\beta_1}\eta\eta KG^2 \sum_{t=1}^{T} \mathbb{E}\left[\left\|\hat{\mathbf{V}}_{t-1}^{-1/2} - \hat{\mathbf{V}}_t^{-1/2}\right\|_1\right] + \frac{\beta_1^2}{(1-\beta_1)^2}\frac{\eta^2\eta_l^2 K^2 G^2}{\sqrt{\epsilon}} \sum_{t=1}^{T} \mathbb{E}\left[\left\|\hat{\mathbf{V}}_{t-1}^{-1/2} - \hat{\mathbf{V}}_t^{-1/2}\right\|_1\right] \\
&\quad + \frac{\beta_1^2}{(1-\beta_1)^2}\eta^2\eta_l^2 K^2 LG^2 \sum_{t=1}^{T} \mathbb{E}\left[\left\|\hat{\mathbf{V}}_{t-1}^{-1/2} - \hat{\mathbf{V}}_t^{-1/2}\right\|^2\right] + \eta^2 L \sum_{t=1}^{T} \mathbb{E}\left[\left\|\hat{\mathbf{V}}_t^{-1/2}\tilde{\Delta}_t\right\|^2\right] \\
&\quad + \frac{\eta^2 L}{2\epsilon} \sum_{t=1}^{T} \mathbb{E}[\|\tilde{\Delta}_t\|^2] + \frac{\eta^2 L}{2\epsilon}\frac{\beta_1^2}{(1-\beta_1)^2} \sum_{t=1}^{T} \mathbb{E}[\|\mathbf{m}_t\|^2] + \sqrt{2}\eta\mathbb{E}\left\langle \frac{\nabla f(\theta_t)}{\sqrt{\beta_2 \mathbf{v}_{t-1} + \epsilon}}, -\frac{1}{M_t}\sum_{i\in M_t} q_t^i\right\rangle.
\end{aligned}
\tag{C.22}
$$

By applying Lemma F.6 into all terms containing the second moment estimate of model difference $\tilde{\Delta}_t$ in (C.22), using the fact that $(\sqrt{\beta_2 K^2 G^2 + \epsilon})^{-1}\|\theta\| \leq (\sqrt{\beta_2 \eta_l^2 K^2 G^2 + \epsilon})^{-1}\|\theta\| \leq \|\frac{\theta}{\sqrt{\beta_2\gamma_2+\epsilon}}\| \leq \epsilon^{-1/2}\|\theta\|$, and applying Lemma F.10, we get

$$\mathbb{E}[f(\mathbf{z}_{T+1})] - f(\mathbf{z}_1)$$

$$\leq -\frac{\eta\eta_l K}{4\sqrt{\beta_2\eta_l^2 K^2 G^2 + \epsilon}} \sum_{t=1}^{T} \mathbb{E}[\|\nabla f(\theta_t)\|^2] + \frac{5\eta\eta_l^3 K^2 L^2 T}{\sqrt{2\epsilon}}(\sigma_l^2 + 6K\sigma_g^2)$$

$$+ \frac{\beta_1}{1-\beta_1}\frac{\eta\eta_l K G^2 d}{\sqrt{\epsilon}} + \frac{\beta_1^2}{(1-\beta_1)^2}\frac{2\eta^2\eta_l^2 K^2 L G^2 d}{\epsilon}$$

$$+ \left(\frac{3\eta^2 L}{2} + \frac{\beta_1^2}{2(1-\beta_1)^2}\eta^2 L + \sqrt{2(1-\beta_2)}\eta G\right)\frac{KT\eta_l^2}{n\epsilon}\sigma_l^2 - \sum_{t=1}^{T}\mathbb{E}\left[\left\|\sum_{i=1}^{m}\sum_{k=0}^{K-1}\nabla F_i(\theta_{t,k}^i)\right\|^2\right]$$

$$\cdot\left[\frac{\eta\eta_l}{2\sqrt{\beta_2 K^2 G^2 + \epsilon}Km^2} - \left(\frac{3\eta^2 L}{2} + \frac{\beta_1^2}{2(1-\beta_1)^2}\eta^2 L + \sqrt{2(1-\beta_2)}\eta G\right)\frac{2\eta_l^2(n-1)}{mn(m-1)\epsilon}\right]$$

$$+ \left(\frac{3\eta^2 L}{2} + \frac{\beta_1^2}{2(1-\beta_1)^2}\eta^2 L + \sqrt{2(1-\beta_2)}\eta G\right)\frac{\eta_l^2(m-n)}{mn(m-1)\epsilon}\left[15mK^3 L^3\eta_l^2(\sigma_l^2 + 6K\sigma_g^2)T\right]$$

$$+ (90mK^4 L^2\eta_l^2 + 3mK^2)\sum_{t=1}^{T}\mathbb{E}[\|\nabla f(\theta_t)\|^2] + 3mK^2 T\sigma_g^2\Big]$$

$$+ \left(\frac{3\eta^2 L}{2} + \frac{\beta_1^2}{2(1-\beta_1)^2}\eta^2 L + \sqrt{2(1-\beta_2)}\eta G\right)\frac{2}{n^2\epsilon}\sum_{t=1}^{T}\mathbb{E}\left\|\frac{1}{M_t}\sum_{i\in M_t}q_t^i\right\|^2 + \frac{\sqrt{2}\eta\eta_l KGCT}{\alpha n\epsilon},$$

then

$$\mathbb{E}[f(\mathbf{z}_{T+1})] - f(\mathbf{z}_1)$$

$$\leq -\frac{\eta\eta_l K}{4\sqrt{\beta_2\eta_l^2 K^2 G^2 + \epsilon}} \sum_{t=1}^{T} \mathbb{E}[\|\nabla f(\theta_t)\|^2] + \frac{5\eta\eta_l^3 K^2 L^2 T}{\sqrt{2\epsilon}}(\sigma_l^2 + 6K\sigma_g^2) + \frac{\beta_1}{1-\beta_1}\frac{\eta\eta_l K G^2 d}{\sqrt{\epsilon}}$$

$$+ \frac{\beta_1^2}{(1-\beta_1)^2}\frac{2\eta^2\eta_l^2 K^2 L G^2 d}{\epsilon} + \left(\frac{3\eta^2 L}{2} + \frac{\beta_1^2}{2(1-\beta_1)^2}\eta^2 L + \sqrt{2(1-\beta_2)}\eta G\right)\left(\frac{KT\eta_l^2}{n\epsilon}\sigma_l^2 + \frac{2T\eta_l^2 C^2 K^2 G^2}{\alpha^2 n^2\epsilon}\right)$$

$$+ \left(\frac{3\eta^2 L}{2} + \frac{\beta_1^2}{2(1-\beta_1)^2}\eta^2 L + \sqrt{2(1-\beta_2)}\eta G\right)\frac{\eta_l^2(m-n)}{mn(m-1)\epsilon}\left[15mK^3 L^3\eta_l^2(\sigma_l^2 + 6K\sigma_g^2)T\right]$$

$$+ (90mK^4 L^2\eta_l^2 + 3mK^2)\sum_{t=1}^{T}\mathbb{E}[\|\nabla f(\theta_t)\|^2] + 3mK^2 T\sigma_g^2\Big] + \frac{\sqrt{2}\eta\eta_l KGCT}{\alpha n\epsilon}.$$

By adopting additional constraint of local learning rate $\eta_l$ with the inequality $\left(\frac{3\eta^2 L}{2} + \frac{\beta_1}{2(1-\beta_1)^2}\eta^2 L + \sqrt{2(1-\beta_2)}\eta G\right)\frac{2\eta_l^2(n-1)}{mn(m-1)\epsilon} - \frac{\eta\eta_l}{2\sqrt{\beta_2 K^2 G^2 + \epsilon}Km^2} \leq 0$, thus we obtain the constraint $\eta_l \leq \frac{n(m-1)}{m(n-1)}\frac{\epsilon}{\sqrt{\beta_2 K^2 G^2 + \epsilon}K(3\eta L + C_1^2\eta_l L + 2\sqrt{2(1-\beta_2)}G)}$, and we further need $\eta_l$ satisfies $\frac{\eta_l K}{4\sqrt{\beta_2\eta_l^2 K^2 G^2 + \epsilon}} - \left(\frac{3\eta^2 L}{2} + \frac{\beta_1^2}{2(1-\beta_1)^2}\eta^2 L + \frac{\eta_l}{4\sqrt{\beta_2\eta_l^2 K^2 G^2 + \epsilon}} + \sqrt{2(1-\beta_2)}\eta G\right)\frac{\eta_l^2(m-n)}{mn(m-1)\epsilon}(90mK^4 L^2\eta_l^2 + 3mK^2) \geq \frac{\eta\eta_l K}{8\sqrt{\beta_2\eta_l^2 K^2 G^2 + \epsilon}}$. Hence we have the following condition on local learning rate

$$\eta \leq \frac{n(m-1)\epsilon}{48m(n-1)}\left[K\sqrt{\beta_2 K^2 G^2 + \epsilon}\cdot\left(\frac{3\eta L}{2} + \frac{\beta_1^2}{2(1-\beta_1)^2}\eta L + \sqrt{2(1-\beta_2)}G\right)\right]^{-1},$$

then

$$\frac{\eta\eta_l K}{8\sqrt{\beta_2\eta_l^2 K^2 G^2 + \epsilon}\cdot T}\sum_{i=1}^{T}\mathbb{E}[\|\nabla f(\theta_t)\|^2]$$

$$\leq \frac{f(\mathbf{z}_0) - \mathbb{E}[f(\mathbf{z}_T)]}{T} + \frac{C_1\eta\eta_l KG^2 d}{T\sqrt{\epsilon}} + \frac{2C_1^2\eta^2\eta_l^2 K^2 LG^2 d}{T\epsilon}$$

$$+ \frac{5\eta\eta_l^3 K^2 L^2}{\sqrt{2\epsilon}}(\sigma_l^2 + 6K\sigma_g^2) + \left(\frac{3\eta^2 L}{2} + \frac{\beta_1^2}{2(1-\beta_1)^2}\eta^2 L + \sqrt{2(1-\beta_2)}\eta G\right)\left(\frac{K\eta_l^2}{n\epsilon}\sigma_l^2 + \frac{2\eta_l^2 C^2 K^2 G^2}{\alpha^2 n^2\epsilon}\right)$$

$$+ \left(\frac{3\eta^2 L}{2} + \frac{\beta_1^2}{2(1-\beta_1)^2}\eta^2 L + \sqrt{2(1-\beta_2)}\eta G\right)\frac{\eta_l^2(m-n)}{mn(m-1)\epsilon}[15mK^3 L^2\eta_l^2(\sigma_l^2 + 6K\sigma_g^2) + 3mK^2\sigma_g^2] + \frac{\sqrt{2}\eta\eta_l KGC}{\alpha n\epsilon}$$

Therefore

$$\min \mathbb{E}[\|\nabla f(\theta_t)\|^2] \leq 8\sqrt{\beta_2\eta_l^2 K^2 G^2 + \epsilon}\left[\frac{f_0 - f_*}{\eta\eta_l KT} + \frac{\Xi}{T} + \Omega\right],$$

where $\Xi = \frac{C_1 G^2 d}{\sqrt{\epsilon}} + \frac{2G_1^2\eta\eta_l KLG^2 d}{\epsilon}$, $\Omega = \frac{5\eta^2 KL^2}{\sqrt{2\epsilon}}(\sigma_l^2 + 6K\sigma_q^2) + [(3 + C_1^2)\eta L + 2\sqrt{2(1-\beta_2)}G](\frac{\eta_l}{n\eta\epsilon}\sigma_l^2 + \frac{2\eta_l K^2 C^2 G^2}{\alpha^2\eta n^2\epsilon}) + [(3+C_1^2)\eta L + 2\sqrt{2(1-\beta_2)}G]\frac{\eta_l(m-n)}{2n(m-1)\epsilon}[15K^2 L^2\eta_l^2(\sigma_l^2 + 6K\sigma_g^2) + 3K\sigma_g^2]\frac{1}{\eta\eta_l K} + \frac{\sqrt{2}GC}{\alpha n\epsilon}$ and $C_1 = \frac{\beta_1}{1-\beta_1}$.

The proof of Theorem 4.4 is similar to the above proof procedure and the detailed proof will not be given here.

## C.4 PROOF OF COROLLARY 4.2

If choose $\eta_l = \Theta(\frac{1}{\sqrt{TK}})$ and $\eta = \Theta(\sqrt{Kn})$, we get $\min_{t\in[T]}\mathbb{E}[\|\nabla f(\theta_t)\|^2] = \mathcal{O}(\frac{\sqrt{K}}{\sqrt{Tn}})$.

# D PROOF OF THEOREMS IN SECTION A

## D.1 PROOF OF THEOREM A.1

Notations and equations: From the update rule of Algorithm 3, then $\mathbf{e}_1 = 0, \mathbf{e}_t = \frac{1}{m}\sum_{i=1}^{m}\mathbf{e}_t^i$ and $\mathbf{m}_t = (1-\beta_1)\sum_{i=1}^{t}\beta_1^{t-i}\widehat{\widehat{\Delta}}_i$. Denote a global uncompressed difference $\Delta_t = \frac{1}{m}\sum_{i=1}^{m}\Delta_t^i$. Denote a virtual momentum sequence: $\mathbf{m}_t' = \beta_1\mathbf{m}_{t-1}' + (1-\beta_1)\Delta_t$, hence we have $\mathbf{m}_t' = (1-\beta_1)\sum_{i=1}^{t}\beta_1^{t-i}\Delta_i$. By the aforementioned definition and notation, then

$$\widehat{\Delta}_t - \Delta_t = \frac{1}{m}\sum_{i=1}^{m}(\widehat{\widehat{\Delta}}_t^i - \Delta_t^i) = \frac{1}{m}\sum_{i=1}^{m}(\widehat{\Delta}_t^i - \Delta_t^i) - \frac{1}{M_t}\sum_{i\in M_t}\mathcal{C}(q_t^i)$$

$$= \frac{1}{m}\sum_{i=1}^{m}(\mathbf{e}_t^i - \mathbf{e}_{t+1}^i) - \frac{1}{M_t}\sum_{i\in M_t}\mathcal{C}(q_t^i) = \mathbf{e}_t - \mathbf{e}_{t+1} - \frac{1}{M_t}\sum_{i\in M_t}\mathcal{C}(q_t^i). \qquad (D.1)$$

Denote the weighted averaging error sequence $\Gamma_t = (1-\beta_1)\sum_{\tau=1}^{t}\beta_1^{t-\tau}\mathbf{e}_r$, with the input $\mathbf{e}_1 = 0$, we obtain the relation between $\Gamma_t$ and $\mathbf{m}_t$ as follows

$$\mathbf{m}_t - \mathbf{m}_t' = (1-\beta_1)\sum_{\tau=1}^{t}\beta_1^{t-\tau}(\widehat{\Delta}_\tau - \Delta_\tau) = (1-\beta_1)\sum_{\tau=1}^{t}\beta_1^{t-\tau}(\mathbf{e}_\tau - \mathbf{e}_{\tau+1}) - (1-\beta_1)\sum_{\tau=1}^{t}\beta_1^{t-\tau}\frac{1}{M_t}\sum_{i\in M_t}\mathcal{C}(q_t^i)$$

$$= \Gamma_t - \Gamma_{t+1} - (1-\beta_1)\sum_{\tau=1}^{t}\beta_1^{t-\tau}\frac{1}{M_t}\sum_{i\in M_t}\mathcal{C}(q_t^i), \qquad (D.2)$$

where the last step holds due to $\Gamma_{t+1} = (1 - \beta_1) \sum_{\tau=1}^{t+1} \beta_1^{t-\tau} \mathbf{e}_{\tau+1} = (1 - \beta_1) \sum_{\tau=1}^{t} \beta_1^{t-\tau} \mathbf{e}_{\tau+1} + \beta_1^t \mathbf{e}_1$.

Similar to previous works studied adaptive methods (Chen et al. (2018c); Wang et al. (2022a)), we introduce a Lyapunov sequence $\mathbf{z}_t$:assume $\theta_0 = \theta_1$, for each $t \geq 1$,

$$\mathbf{z}_t = \theta_t + \frac{\beta_1}{1 - \beta_1}(\theta_t - \theta_{t-1}) = \frac{1}{1 - \beta_1}\theta_t - \frac{\beta_1}{1 - \beta_1}\theta_{t-1}.$$

Therefore, by the update rule of $\theta_t$,

$$\mathbf{y}_{t+1} = \theta_{t+1} + \eta \frac{\beta_1}{1 - \beta_1} \hat{\mathbf{V}}_t^{-1/2} \mathbf{m}_t - (1 - \beta_1) \sum_{\tau=1}^{t} \beta_1^{t-\tau} \frac{1}{M_t} \sum_{i \in M_t} \mathcal{C}(q_t^i)]$$

$$= \theta_{t+1} + \eta \frac{\beta_1}{1 - \beta_1} \hat{\mathbf{V}}_t^{-1/2} [\mathbf{m}_t' + \Gamma_t - \Gamma_{t+1}] - (1 - \beta_1) \sum_{\tau=1}^{t} \beta_1^{t-\tau} \frac{1}{M_t} \sum_{i \in M_t} \mathcal{C}(q_t^i)$$

$$= \theta_{t+1} + \eta \frac{\beta_1}{1 - \beta_1} \hat{\mathbf{V}}_t^{-1/2} \mathbf{m}_t' + \eta \frac{\beta_1}{1 - \beta_1} \hat{\mathbf{V}}_t^{-1/2} \left[ \frac{\Gamma_{t+1} - (1 - \beta_1)\mathbf{e}_{t+1}}{\beta_1} - \Gamma_{t+1} \right] - (1 - \beta_1) \sum_{\tau=1}^{t} \beta_1^{t-\tau} \frac{1}{M_t} \sum_{i \in M_t} \mathcal{C}(q_t^i)$$

$$= \theta_{t+1} + \eta \frac{\beta_1}{1 - \beta_1} \hat{\mathbf{V}}_t^{-1/2} \mathbf{m}_t' + \eta \hat{\mathbf{V}}_t^{-1/2} \Gamma_{t+1} - \eta \hat{\mathbf{V}}_t^{-1/2} \mathbf{e}_{t+1} - (1 - \beta_1) \sum_{\tau=1}^{t} \beta_1^{t-\tau} \frac{1}{M_t} \sum_{i \in M_t} \mathcal{C}(q_t^i). \tag{D.3}$$

The third equation holds due to the fact that $\Gamma_{t+1} = \beta_1 \Gamma_t + (1 - \beta_1)\mathbf{e}_{t+1}$. We then introduce a new sequence based on the previous Lyapunov sequence $\mathbf{y}_t$ as follows

$$\mathbf{z}_{t+1} = \mathbf{y}_{t+1} + (1-\beta_1) \sum_{\tau=1}^{t} \beta_1^{t-\tau} \frac{1}{M_t} \sum_{i \in M_t} \mathcal{C}(q_t^i) + \eta \hat{\mathbf{V}}_t^{-1/2} \mathbf{e}_{t+1} = \theta_{t+1} + \eta \frac{\beta_1}{1 - \beta_1} \hat{\mathbf{V}}_t^{-1/2} \mathbf{m}_t' + \eta \hat{\mathbf{V}}_t^{-1/2} \Gamma_{t+1}. \tag{D.4}$$

The sequence difference $\mathbf{z}_{t+1} - \mathbf{z}_t$ can be represented by

$$\mathbf{z}_{t+1} - \mathbf{z}_t = \theta_{t+1} - \theta_t + \eta \frac{\beta_1}{1 - \beta_1} \hat{\mathbf{V}}_t^{-1/2} \mathbf{m}_t' - \eta \frac{\beta_1}{1 - \beta_1} \hat{\mathbf{V}}_{t-1}^{-1/2} \mathbf{m}_{t-1}' + \eta \hat{\mathbf{V}}_t^{-1/2} \Gamma_{t+1} - \eta \hat{\mathbf{V}}_{t-1}^{-1/2} \Gamma_t$$

$$= \eta \hat{\mathbf{V}}_t^{-1/2} \mathbf{m}_t + \eta \hat{\mathbf{V}}_t^{-1/2} \Gamma_{t+1} + \eta \frac{\beta_1}{1 - \beta_t} \hat{\mathbf{V}}_t^{-1/2} \mathbf{m}_t' - \eta \frac{\beta_1}{1 - \beta_t} \hat{\mathbf{V}}_{t-1}^{-1/2} \mathbf{m}_{t-1}' - \eta \hat{\mathbf{V}}_{t-1}^{-1/2} \Gamma_t,$$

where the second equation follows the update rule of $\theta_{t+1}$. Following (D.2), then combining likely terms and applying the definition of $\mathbf{m}_t'$, then

$$\mathbf{z}_{t+1} - \mathbf{z}_t = \eta \hat{\mathbf{V}}_t^{-1/2} \mathbf{m}_t' + \eta \hat{\mathbf{V}}_t^{-1/2} \Gamma_t + \eta \frac{\beta_1}{1 - \beta_1} \hat{\mathbf{V}}_t^{-1/2} \mathbf{m}_t' - \eta \frac{\beta_1}{1 - \beta_1} \hat{\mathbf{V}}_{t-1}^{-1/2} \mathbf{m}_{t-1}' - \eta \hat{\mathbf{V}}_{t-1}^{-1/2} \Gamma_t$$

$$= \eta \frac{1}{1 - \beta_1} \hat{\mathbf{V}}_t^{-1/2} \mathbf{m}_t' - \eta \frac{\beta_1}{1 - \beta_1} \hat{\mathbf{V}}_{t-1}^{-1/2} \mathbf{m}_{t-1}' + \eta \hat{\mathbf{V}}_t^{-1/2} \Gamma_t - \eta \hat{\mathbf{V}}_{t-1}^{-1/2} \Gamma,$$

$$= \eta \frac{1}{1 - \beta_1} \hat{\mathbf{V}}_t^{-1/2} [\beta_1 \mathbf{m}_{t-1}' + (1 - \beta_1)\Delta_t] - \eta \frac{\beta_1}{1 - \beta_1} \hat{\mathbf{V}}_{t-1}^{-1/2} \mathbf{m}_{t-1}' + \eta \hat{\mathbf{V}}_t^{-1/2} \Gamma_t - \eta \hat{\mathbf{V}}_{t-1}^{-1/2} \Gamma_t$$

$$= \eta \hat{\mathbf{V}}_t^{-1/2} \Delta_t - \eta \frac{\beta_1}{1 - \beta_1} \left( \hat{\mathbf{V}}_{t-1}^{-1/2} - \hat{\mathbf{V}}_t^{-1/2} \right) \mathbf{m}_{t-1}' - \eta \left( \hat{\mathbf{V}}_{t-1}^{-1/2} - \hat{\mathbf{V}}_t^{-1/2} \right) \Gamma_t. \tag{D.5}$$

Therefore, we obtain a helpful Lyapunov sequence for our proof of FedCAMS. The proof of Fed-CAMS in full participation settings has a similar outline with the proof of FedAMS. By Assumption 3.1, then

$$\mathrm{E}[f(\mathbf{z}_{t+1})] - f(\mathbf{z}_t)$$

$$\leq \mathbb{E}[\langle \nabla f(\mathbf{z}_t), \mathbf{z}_{t+1} - \mathbf{z}_t \rangle] + \frac{L}{2}\mathbb{E}[\|\mathbf{z}_{t+1} - \mathbf{z}_t\|^2]$$

$$\leq \mathbb{E}\left[\left\langle \nabla f(\mathbf{z}_t), \eta \widehat{\mathbf{V}}_t^{-1/2}\Delta_t \right\rangle\right]$$

$$- \mathbb{E}\left[\left\langle \nabla f(\mathbf{z}_t), \eta \frac{\beta_1}{1-\beta_1}\left(\widehat{\mathbf{V}}_{t-1}^{-1/2} - \widehat{\mathbf{V}}_t^{-1/2}\right)\mathbf{m}'_{t-1} + \eta\left(\widehat{\mathbf{V}}_{t-1}^{-1/2} - \widehat{\mathbf{V}}_t^{-1/2}\right)\boldsymbol{\Gamma}_t \right\rangle\right]$$

$$+ \frac{\eta^2 L}{2}\mathbb{E}\left[\left\|\widehat{\mathbf{V}}_t^{-1/2}\Delta_t - \frac{\beta_1}{1-\beta_1}\left(\widehat{\mathbf{V}}_{t-1}^{-1/2} - \widehat{\mathbf{V}}_t^{-1/2}\right)\mathbf{m}'_{t-1} - \left(\widehat{\mathbf{V}}_{t-1}^{-1/2} - \widehat{\mathbf{V}}_t^{-1/2}\right)\boldsymbol{\Gamma}_t\right\|^2\right]$$

$$= \underbrace{\mathbb{E}\left[\left\langle \nabla f(\theta_t), \eta\widehat{\mathbf{V}}_t^{-1/2}\Delta_t \right\rangle\right]}_{T_1} \underbrace{-\eta\mathbb{E}\left[\left\langle \nabla f(\mathbf{z}_t), \frac{\beta_1}{1-\beta_1}\left(\widehat{\mathbf{V}}_{t-1}^{-1/2} - \widehat{\mathbf{V}}_t^{-1/2}\right)\mathbf{m}'_{t-1} + \left(\widehat{\mathbf{V}}_{t-1}^{-1/2} - \widehat{\mathbf{V}}_t^{-1/2}\right)\boldsymbol{\Gamma}_t \right\rangle\right]}_{T_2}$$

$$+ \underbrace{\frac{\eta^2 L}{2}\mathbb{E}\left[\left\|\widehat{\mathbf{V}}_t^{-1/2}\Delta_t - \frac{\beta_1}{1-\beta_1}\left(\widehat{\mathbf{V}}_{t-1}^{-1/2} - \widehat{\mathbf{V}}_t^{-1/2}\right)\mathbf{m}'_{t-1} - \left(\widehat{\mathbf{V}}_{t-1}^{-1/2} - \widehat{\mathbf{V}}_t^{-1/2}\right)\boldsymbol{\Gamma}_t\right\|^2\right]}_{T_3}$$

$$+ \underbrace{\mathbb{E}\left[\left\langle \nabla f(\mathbf{z}_t) - \nabla f(\theta_t), \eta\widehat{\mathbf{V}}_t^{-1/2}\Delta_t \right\rangle\right]}_{T_4}. \tag{D.6}$$

here we recall the notation $\widehat{\mathbf{V}}_t = \mathrm{diag}(\widehat{\mathbf{v}}_t) = \mathrm{diag}(\max(\widehat{\mathbf{v}}_{t-1}, \mathbf{v}_t, \epsilon))$.

**Bounding $T_1$:**

$$T1 = \mathbb{E}\left[\left\langle \nabla f(\theta_t), \eta\frac{\Delta_t}{\sqrt{\widehat{\mathbf{v}}_t}} \right\rangle\right]$$

$$\leq \eta\mathbb{E}\left[\left\langle \nabla f(\theta_t), \frac{\sqrt{2}\cdot\Delta_t}{\sqrt{\mathbf{v}_t + \epsilon}} \right\rangle\right]$$

$$= \sqrt{2}\eta\mathbb{E}\left[\left\langle \nabla f(\theta_t), \frac{\Delta_t}{\sqrt{\beta_2\mathbf{v}_{t-1} + \epsilon}} \right\rangle\right] + \sqrt{2}\eta\mathbb{E}\left[\left\langle \nabla f(\theta_t), \frac{\Delta_t}{\sqrt{\mathbf{v}_t + \epsilon}} - \frac{\Delta_t}{\sqrt{\beta_2\mathbf{v}_{t-1} + \epsilon}} \right\rangle\right], \tag{D.7}$$

where the first inequality follows by the fact that $\hat{v}_t \geq \frac{v_t + \epsilon}{2}$. For the second term in (D.7), then

$$\sqrt{2}\cdot\eta\mathbb{E}\left[\left\langle \nabla f(\theta_t), \frac{\Delta_t}{\sqrt{\mathbf{v}_t + \epsilon}} - \frac{\Delta_t}{\sqrt{\beta_2\mathbf{v}_{t-1} + \epsilon}} \right\rangle\right]$$

$$\leq \sqrt{2}\cdot\eta\cdot\mathbb{E}\|\nabla f(\theta_t)\|\mathbb{E}\left[\left\|\frac{1}{\sqrt{\mathbf{v}_t + \epsilon}} - \frac{1}{\sqrt{\beta_2\mathbf{v}_{t-1} + \epsilon}}\right\| \cdot \|\Delta_t\|\right]$$

$$\leq \frac{\eta\sqrt{2(1-\beta_2)}G}{\epsilon}\mathbb{E}[\|\Delta_t\|^2],$$

where the second inequality follows from Lemma F.1 and F.5, and we will further apply the bound for $E[\|\Delta_t\|^2]$ by applying Lemma F.7. For the first term in (D.6), then

$$\sqrt{2} \cdot \eta \mathbb{E}\left[\left\langle \nabla f(\theta_t), \frac{\Delta_t}{\sqrt{\beta_2 \mathbf{v}_{t-1} + \epsilon}} \right\rangle\right]$$

$$= \sqrt{2} \cdot \eta \mathbb{E}\left[\left\langle \frac{\nabla f(\theta_t)}{\sqrt{\beta_2 \mathbf{v}_{t-1} + \epsilon}}, \Delta_t + \eta K \nabla f(\theta_t) - \eta_t K \nabla f(\theta_t) \right\rangle\right]$$

$$= -\sqrt{2}\eta\eta_l K \mathbb{E}\left[\left\|\frac{\nabla f(\theta_t)}{\sqrt[4]{\beta_2 \mathbf{v}_{t-1} + \epsilon}}\right\|^2\right] + \sqrt{2}\eta \mathbb{E}\left[\left\langle \frac{\nabla f(\theta_t)}{\sqrt{\beta_2 \mathbf{v}_{t-1} + \epsilon}}, \Delta_t + \eta_l K \nabla f(\theta_t) \right\rangle\right]$$

$$= -\sqrt{2}\eta\eta_t K \mathbb{E}\left[\left\|\frac{\nabla f(\theta_t)}{\sqrt[4]{\beta_2 \mathbf{v}_{t-1} + \epsilon}}\right\|^2\right] + \sqrt{2}\eta \left\langle \frac{\nabla f(\theta_t)}{\sqrt{\beta_2 \mathbf{v}_{t-1} + \epsilon}}, \mathbb{E}\left[-\frac{\eta_l}{m}\sum_{i=1}^{m}\sum_{k=0}^{K-1}\mathbf{g}_{t,k}^i + \eta_l K \nabla f(\theta_t)\right]\right\rangle$$

$$= -\sqrt{2}\eta\eta_l K \mathbb{E}\left[\left\|\frac{\nabla f(\theta_t)}{\sqrt[4]{\beta_2 \mathbf{v}_{t-1} + \epsilon}}\right\|^2\right] + \sqrt{2}\eta \left\langle \frac{\nabla f(\theta_t)}{\sqrt{\beta_2 \mathbf{v}_{t-1} + \epsilon}}, \mathbb{E}\left[-\frac{\eta_l}{m}\sum_{i=1}^{m}\sum_{k=0}^{K-1}\mathbf{g}_{t,k}^i + \frac{\eta_l K}{m}\sum_{i=1}^{m}\nabla F_i(\theta_t)\right]\right\rangle$$

$$\text{(D.8)}$$

For the last term in (D.8),

$$\sqrt{2}\eta \left\langle \frac{\nabla f(\theta_t)}{\sqrt{\beta_2 \mathbf{v}_{t-1} + \epsilon}}, \mathbb{E}\left[-\frac{\eta_l}{m}\sum_{i=1}^{m}\sum_{k=0}^{K-1}\mathbf{g}_{t,k}^i + \frac{\eta_l K}{m}\sum_{i=1}^{m}\nabla F_i(\theta_t)\right]\right\rangle$$

$$= \sqrt{2}\eta \left\langle \frac{\sqrt{\eta K}}{\sqrt[4]{\beta_2 \mathbf{v}_{t-1} + \epsilon}}\nabla f(\theta_t), -\frac{\sqrt{\eta_n K}}{Km}\frac{1}{\sqrt[4]{\beta_2 \mathbf{v}_{t-1} + \epsilon}}\mathbb{E}\left[\sum_{i=1}^{m}\sum_{k=0}^{K-1}(\nabla F_i(\theta_{t,k}^i) - \nabla F_i(\theta_t))\right]\right\rangle$$

$$= \frac{\sqrt{2}\eta\eta_l K}{2}\left\|\frac{\nabla f(\theta_t)}{\sqrt[4]{\beta_2 \mathbf{v}_{t-1} + \epsilon}}\right\|^2 + \frac{\sqrt{2}\eta\eta_l}{2Km^2}\mathbb{E}\left\|\frac{1}{\sqrt[4]{\beta_2 \mathbf{v}_{t-1} + \epsilon}}\sum_{i=1}^{m}\sum_{k=0}^{K-1}(\nabla F_i(\theta_{t,k}^i) - \nabla F_i(\theta_t))\right\|^2$$

$$- \frac{\sqrt{2}\eta\eta_l}{2Km^2}\mathbb{E}\left[\left\|\frac{1}{\sqrt[4]{\beta_2 \mathbf{v}_{t-1} + \epsilon}}\sum_{i=1}^{m}\sum_{k=0}^{K-1}\nabla F_i(\theta_{t,k}^i)\right\|^2\right]$$

$$\leq \frac{\sqrt{2}\eta\eta_l K}{2}\left\|\frac{\nabla f(\theta_t)}{\sqrt[4]{\beta_2 \mathbf{v}_{t-1} + \epsilon}}\right\|^2 + \frac{\sqrt{2}\eta\eta_l}{2m}\cdot\sum_{i=1}^{m}\sum_{k=0}^{K-1}\mathbb{E}\left[\left\|\frac{\nabla F_i(\theta_{t,k}^i) - \nabla F_i(\theta_t)}{\sqrt[4]{\beta_2 \mathbf{v}_{t-1} + \epsilon}}\right\|^2\right]$$

$$- \frac{\sqrt{2}\eta\eta_l}{2Km^2}\mathbb{E}\left[\left\|\frac{1}{\sqrt[4]{\beta_2 \mathbf{v}_{t-1} + \epsilon}}\sum_{i=1}^{m}\sum_{k=0}^{K-1}\nabla F_i(\theta_{t,k}^i)\right\|^2\right],$$

where the second equation follows from $\langle \mathbf{x}, \mathbf{y} \rangle = \frac{1}{2}[\|\mathbf{x}\|^2 + \|\mathbf{y}\|^2 - \|\mathbf{x} - \mathbf{y}\|^2]$, and the inequality holds by applying Cauchy-Schwarz inequality. Then by Assumption 3.1,

$$\sqrt{2}\eta \left\langle \frac{\nabla f(\theta_t)}{\sqrt{\beta_2 \mathbf{v}_{t-1} + \epsilon}}, \mathbb{E}\left[-\frac{\eta_t}{m}\sum_{i=1}^{m}\sum_{k=0}^{K-1}\mathbf{g}_{t,k}^i + \frac{\eta_l K}{m}\sum_{i=1}^{m}\nabla F_i(\theta_t)\right]\right\rangle$$

$$\leq \frac{\sqrt{2}\eta\eta_l K}{2}\left\|\frac{\nabla f(\theta_t)}{\sqrt[4]{\beta_2 \mathbf{v}_{t-1} + \epsilon}}\right\|^2 + \frac{\sqrt{2}\eta\eta_l L^2}{2m}\sum_{i=1}^{m}\sum_{k=0}^{K-1}\mathbb{E}\left[\left\|\frac{\theta_{t,k}^i - \theta_t}{\sqrt[4]{\beta_2 \mathbf{v}_{t-1} + \epsilon}}\right\|^2\right]$$

$$- \frac{\sqrt{2}\eta\eta_l}{2Km^2}\mathbb{E}\left[\left\|\frac{1}{\sqrt[4]{\beta_2 \mathbf{v}_{t-1} + \epsilon}}\sum_{i=1}^{m}\sum_{k=0}^{K-1}\nabla F_i(\theta_{t,k}^i)\right\|^2\right]$$

$$\leq \frac{3\sqrt{2}\eta\eta_l K}{4}\left\|\frac{\nabla f(\theta_t)}{\sqrt[4]{\beta_2 \mathbf{v}_{t-1} + \epsilon}}\right\|^2 + \frac{5\eta\eta_l^3 K^2 L^2}{\sqrt{2\epsilon}}(\sigma_l^2 + 6K\sigma_g^2) \qquad\qquad - \frac{\sqrt{2}\eta\eta_l}{2Km^2}\mathbb{E}\Big[\Big\|\frac{1}{\sqrt[4]{\beta_2 \mathbf{v}_{t-1} + \epsilon}}\sum_{i=1}^{m}\sum_{k=0}^{K-1}\nabla F_i(\theta_t^i$$

where the last inequality holds by applying Lemma F.14 and the constraint of local learning rate $\eta_n \leq \frac{1}{8KL}$.Then

$$
\begin{aligned}
T_1 \leq & -\frac{\sqrt{2} \cdot \eta\eta_l K}{4} \mathbb{E}\left[\left\|\frac{\nabla f(\theta_t)}{\sqrt[4]{\beta_2 \mathbf{v}_{t-1} + \epsilon}}\right\|^2\right] + \frac{5\eta\eta_l^3 K^2 L^2}{\sqrt{2}\epsilon}(\sigma_l^2 + 6K\sigma_g^2) \\
& - \frac{\sqrt{2} \cdot \eta\eta_l}{2Km^2}\mathbb{E}\left[\left\|\frac{1}{\sqrt[4]{\beta_2\mathbf{v}_{t-1}+\epsilon}}\sum_{i=1}^{m}\sum_{k=0}^{K-1}\nabla F_i(\theta_{t,k}^i)\right\|^2\right] + \frac{\eta\sqrt{2(1-\beta_2)}G}{\epsilon}\mathbb{E}[\|\Delta_t\|^2] \\
\leq & -\frac{\eta\eta_l K}{4}\mathbb{E}\left[\left\|\frac{\nabla f(\theta_t)}{\sqrt[4]{\beta_2\mathbf{v}_{t-1}+\epsilon}}\right\|^2\right] + \frac{5\eta\eta_l^3 K^2 L^2}{\sqrt{2}\epsilon}(\sigma_l^2 + 6K\sigma_g^2) \\
& - \frac{\eta\eta_l}{2Km^2}\mathbb{E}\left[\left\|\frac{1}{\sqrt[4]{\beta_2\mathbf{v}_{t-1}+\epsilon}}\sum_{i=1}^{m}\sum_{k=0}^{K-1}\nabla F_i(\theta_{t,k}^i)\right\|^2\right] + \frac{\eta\sqrt{2(1-\beta_2)}G}{\epsilon}\mathbb{E}[\|\Delta_t\|^2]. \quad \text{(D.9)}
\end{aligned}
$$

**Bounding $T_2$:** The bound for $T_2$ mainly follows by the update rule and definition of virtual sequence $\mathbf{z}_t$.

$$
\begin{aligned}
T_2 = & -\eta\mathbb{E}\left[\left\langle \nabla f(\mathbf{z}_t), \frac{\beta_1}{1-\beta_1}\left(\widehat{\mathbf{V}}_{t-1}^{-1/2} - \widehat{\mathbf{V}}_t^{-1/2}\right)\mathbf{m}_{t-1}' + \left(\widehat{\mathbf{V}}_{t-1}^{-1/2} - \widehat{\mathbf{V}}_t^{-1/2}\right)\Gamma_t \right\rangle\right] \\
= & \eta\mathbb{E}\left[\left\langle -\nabla f(\theta_t) + \nabla f(\theta_t) - \nabla f(\mathbf{z}_t), \left(\widehat{\mathbf{V}}_{t-1}^{-1/2} - \widehat{\mathbf{V}}_t^{-1/2}\right)\left(\frac{\beta_1}{1-\beta_1}\mathbf{m}_{t-1}' + \Gamma_t\right)\right\rangle\right] \\
\leq & \eta\mathbb{E}\left[\|\nabla f(\theta_t)\|\left\|\left(\widehat{\mathbf{V}}_{t-1}^{-1/2} - \widehat{\mathbf{V}}_t^{-1/2}\right)\left(\frac{\beta_1}{1-\beta_1}\mathbf{m}_{t-1}' + \Gamma_t\right)\right\|\right] \\
& + \eta^2 L\mathbb{E}\left[\left\|\widehat{\mathbf{V}}_{t-1}^{-1/2}\left(\frac{\beta_1}{1-\beta_1}\mathbf{m}_{t-1}' + \Gamma_t\right)\right\|\left\|\left(\widehat{\mathbf{V}}_{t-1}^{-1/2} - \widehat{\mathbf{V}}_t^{-1/2}\right)\left(\frac{\beta_1}{1-\beta_1}\mathbf{m}_{t-1}' + \Gamma_t\right)\right\|\right] \\
\leq & \eta C_1\eta_l KG^2\mathbb{E}\left[\left\|\widehat{\mathbf{V}}_{t-1}^{-1/2} - \widehat{\mathbf{V}}_t^{-1/2}\right\|_1\right] + \eta^2 C_1^2 L\eta_l^2 K^2 G^2\epsilon^{-1/2}\mathbb{E}\left[\left\|\widehat{\mathbf{V}}_{t-1}^{-1/2} - \widehat{\mathbf{V}}_t^{-1/2}\right\|_1\right],
\end{aligned}
$$
$$\text{(D.10)}$$

where the last inequality holds by Lemma F.5, here $C_1 = \frac{\beta_1}{1-\beta_1} + \sqrt{\frac{12q^2}{(1-q^2)^2} + \frac{(1-q^2)^2 C^2}{\alpha^2 m^2 q^2}}$.

**Bounding $T_3$:**

$$
\begin{aligned}
T_3 = & \frac{\eta^2 L}{2}\mathbb{E}\left[\left\|\hat{\mathbf{V}}_t^{-1/2}\Delta_t + \frac{\beta_1}{1-\beta_1}\left(\hat{\mathbf{V}}_{t-1}^{-1/2} - \hat{\mathbf{V}}_t^{-1/2}\right)\mathbf{m}_{t-1}' + \left(\hat{\mathbf{V}}_{t-1}^{-1/2} - \hat{\mathbf{V}}_t^{-1/2}\right)\Gamma_t\right\|^2\right] \\
\leq & \eta^2 L\mathbb{E}\left[\left\|\widehat{\mathbf{V}}_t^{-1/2}\Delta_t\right\|^2\right] + \eta^2 L\mathbb{E}\left[\left\|\frac{\beta_1}{1-\beta_1}\left(\widehat{\mathbf{V}}_{t-1}^{-1/2} - \widehat{\mathbf{V}}_t^{-1/2}\right)\mathbf{m}_{t-1}' + \left(\widehat{\mathbf{V}}_{t-1}^{-1/2} - \widehat{\mathbf{V}}_t^{-1/2}\right)\boldsymbol{\Gamma}_t\right\|^2\right] \\
\leq & \eta^2 L\mathbb{E}\left[\left\|\widehat{\mathbf{V}}_t^{-1/2}\Delta_t\right\|^2\right] + \eta^2 LC_1^2\eta_l^2 K^2 G^2\mathbb{E}\left[\left\|\widehat{\mathbf{V}}_{t-1}^{-1/2} - \widehat{\mathbf{V}}_t^{-1/2}\right\|^2\right], \quad \text{(D.11)}
\end{aligned}
$$

where the first inequality follows by Cauchy-Schwarz inequality, and the second one follows by Lemma F.5, here $C_1 = \frac{\beta_1}{1-\beta_1} + \sqrt{\frac{12q^2}{(1-q^2)^2} + \frac{(1-q^2)^2 C^2}{\alpha^2 m^2 q^2}}$.

**Bounding $T_4$:**

$$T_4 = \mathbb{E}\left[\left\langle \nabla f(\mathbf{z}_t) - \nabla f(\theta_t), \eta \widehat{\mathbf{V}}_t^{-1/2}\Delta_t \right\rangle\right]$$

$$\leq \mathbb{E}\left[\|\nabla f(\mathbf{z}_t) - \nabla f(\theta_t)\|\|\eta\widehat{\mathbf{V}}_t^{-1/2}\Delta_t\|\right]$$

$$\leq L\mathbb{E}\left[\|\mathbf{z}_t - \theta_t\|\|\eta\widehat{\mathbf{V}}_t^{-1/2}\Delta_t\|\right]$$

$$\leq \frac{\eta^2 L}{2}\mathbb{E}\left[\left\|\widehat{\mathbf{V}}_t^{-1/2}\Delta_t\right\|^2\right] + \frac{\eta^2 L}{2}\mathbb{E}\left[\left\|\frac{\beta_1}{1-\beta_1}\widehat{\mathbf{V}}_{t-1}^{-1/2}\mathbf{m}'_{t-1} + \widehat{\mathbf{V}}_{t-1}^{-1/2}\mathbf{\Gamma}_t\right\|^2\right],$$

where the first inequality holds by the fact of $\langle \mathbf{a}, \mathbf{b}\rangle \leq \|\mathbf{a}\|\|\mathbf{b}\|$, the second one follows from Assumption 3.1 and the third one holds by the definition of virtual sequence $\mathbf{z}_t$ and the fact of $\|\mathbf{a}\|\|\mathbf{b}\| \leq \frac{1}{2}\|\mathbf{a}\|^2 + \frac{1}{2}\|\mathbf{b}\|^2$. Then summing $T_4$ over $t = 1, \cdots, T$,

$$\sum_{t=1}^{T} T_4 \leq \frac{\eta^2 L}{2}\sum_{t=1}^{T}\mathbb{E}\left[\left\|\widehat{\mathbf{V}}_t^{-1/2}\Delta_t\right\|^2\right] + \frac{\eta^2 L}{2\epsilon}\sum_{t=1}^{T}\mathbb{E}\left[\left\|\frac{\beta_1}{1-\beta_1}\mathbf{m}'_{t-1} + \mathbf{\Gamma}_t\right\|^2\right]$$

$$\leq \frac{\eta^2 L}{2\epsilon}\sum_{t=1}^{T}\mathbb{E}[\|\Delta_t\|^2] + \frac{\eta^2 L}{\epsilon}\left[\frac{\beta_1^2}{(1-\beta_1)^2}\sum_{t=1}^{T}\mathbb{E}\|\mathbf{m}'_{t-1}\|^2 + \sum_{t=1}^{T}\mathbb{E}\|\mathbf{\Gamma}_t\|^2\right]. \qquad (D.12)$$

By Lemma F.11,

$$\sum_{t=1}^{T}\mathbb{E}[\|\mathbf{m}'_{t-1}\|^2] \leq \frac{TK\eta_l^2}{m}\sigma_l^2 + \frac{\eta_l^2}{m^2}\sum_{t=1}^{T}\mathbb{E}\left[\left\|\sum_{i=1}^{m}\sum_{k=0}^{K-1}\nabla F_i(\theta_{t,k}^i)\right\|^2\right],$$

and

$$\sum_{t=1}^{T}\mathbb{E}[\|\mathbf{\Gamma}_t\|^2] \leq \frac{4T(q+\gamma)^2}{(1-q^2)^2}\frac{K\eta_l^2}{m}\sigma_l^2 + \frac{\eta_l^2}{m^2}\frac{4(q+\gamma)^2}{(1-q^2)^2}\sum_{t=1}^{T}\mathbb{E}\left[\left\|\sum_{i=1}^{m}\sum_{k=0}^{K-1}\nabla F_i(\theta_{t,k}^i)\right\|^2\right].$$

Therefore, the $T_4$ term is bounded by

$$\sum_{t=1}^{T} T_4 \leq \frac{\eta^2 L}{2\epsilon}\sum_{t=1}^{T}\mathbb{E}[\|\Delta_t\|^2] + \frac{C_2\eta^2 L}{\epsilon}\frac{\eta_l^2}{m^2}\sum_{t=1}^{T}\mathbb{E}\left[\left\|\sum_{i=1}^{m}\sum_{k=0}^{K-1}\nabla F_i(\theta_{t,k}^i)\right\|^2\right] + \frac{C_2\eta^2 L}{\epsilon}\frac{TK\eta_l^2}{m}\sigma_l^2,$$

$$(D.13)$$

where $C_2 = \frac{4(q+\gamma+\frac{\lambda C}{\alpha m})^2}{(1-q^2)^2} + \frac{\beta_1^2}{(1-\beta_1)^2}$.

Merging pieces together: Substituting (D.9), (D.10) and (D.11) into (D.6), summing over from $t = 1$ to $T$ and then adding (D.13),

$$\mathbb{E}[f(\mathbf{z}_{T+1})] - f(\mathbf{z}_1) = \sum_{t=1}^{T}[T_1 + T_2 + T_3 + T_4]$$

$$\leq -\frac{\eta\eta_l K}{4}\sum_{t=1}^{T}\mathbb{E}\left[\left\|\frac{\nabla f(\theta_t)}{\sqrt[4]{\beta_2\mathbf{v}_{t-1}+\epsilon}}\right\|^2\right] + \frac{5\eta\eta_l^3 K^2 L^2 T}{\sqrt{2}\epsilon}(\sigma_l^2 + 6K\sigma_g^2) + \frac{\sqrt{2(1-\beta_2)}\eta G}{\epsilon}\sum_{t=1}^{T}\mathbb{E}[\|\Delta_t\|^2]$$

$$-\frac{\eta m}{2Km^2}\sum_{t=1}^{T}\mathbb{E}\left[\left\|\frac{1}{\sqrt[4]{\beta_2\mathbf{v}_{t-1}+\epsilon}}\sum_{i=1}^{m}\sum_{k=0}^{K-1}\nabla F_i(\theta_{t,k}^i))\right\|^2\right] + C_1\eta\eta_t KG^2\sum_{t=1}^{T}\mathbb{E}\left[\left\|\widehat{\mathbf{V}}_{t-1}^{-1/2}-\widehat{\mathbf{V}}_t^{-1/2}\right\|^2\right]$$

$$+\frac{C_1^2\eta^2\eta_l^2 K^2 G^2}{\sqrt{\epsilon}}\sum_{t=1}^{T}\mathbb{E}\left[\left\|\widehat{\mathbf{V}}_{t-1}^{-1/2}-\widehat{\mathbf{V}}_t^{-1/2}\right\|_1\right] + C_1^2\eta^2\eta_l^2 K^2 LG^2\sum_{t=1}^{T}\mathbb{E}\left[\left\|\widehat{\mathbf{V}}_{t-1}^{-1/2}-\widehat{\mathbf{V}}_t^{-1/2}\right\|^2\right]$$

$$+\eta^2 L\sum_{t=1}^{T}\mathbb{E}\left[\left\|\widehat{\mathbf{V}}_t^{-1/2}\Delta_t\right\|^2\right] + \frac{\eta^2 L}{2}\sum_{t=1}^{T}\mathbb{E}\left[\left\|\widehat{\mathbf{V}}_t^{-1/2}\Delta_t\right\|^2\right] + \frac{\eta^2 L}{2}\frac{\beta_1^2}{(1-\beta_1)^2}\sum_{t=1}^{T}\mathbb{E}[\|\mathbf{m}_t'\|^2]$$

$$+\frac{\eta^2 L}{2}\sum_{t=1}^{T}\mathbb{E}[\|\mathbf{\Gamma}_t\|^2]. \tag{D.14}$$

Hence by organizing and applying Lemmas F.5, then

$$\mathbb{E}[f(\mathbf{z}_{T+1})] - f(\mathbf{z}_1)$$

$$\leq -\frac{\eta\eta_l K}{4}\sum_{t=1}^{T}\mathbb{E}\left[\left\|\frac{\nabla f(\theta_t)}{\sqrt[4]{\beta_2\mathbf{v}_{t-1}+\epsilon}}\right\|^2\right] + \frac{5\eta\eta_l^3 K^2 L^2 T}{\sqrt{2}\epsilon}(\sigma_l^2 + 6K\sigma_g^2)$$

$$-\frac{\eta\eta_l}{2Km^2}\sum_{t=1}^{T}\mathbb{E}\left[\left\|\frac{1}{\sqrt[4]{\beta_2\mathbf{v}_{t-1}+\epsilon}}\sum_{i=1}^{m}\sum_{k=0}^{K-1}\nabla F_i(\theta_{t,k}^i))\right\|^2\right] + \frac{C_1\eta\eta_l KG^2 d}{\sqrt{\epsilon}} + \frac{2C_1^2\eta^2\eta_l^2 K^2 LG^2 d}{\epsilon}$$

$$+\left(\eta^2 L + \frac{\eta^2 L}{2} + \sqrt{2(1-\beta_2)}\eta G\right)\left[\frac{KT\eta_l^2}{m\epsilon}\sigma_l^2 + \frac{\eta_l^2}{m^2\epsilon}\sum_{t=1}^{T}\mathbb{E}\left[\left\|\sum_{i=1}^{m}\sum_{k=0}^{K-1}\nabla F_i(\theta_{t,k}^i)\right\|^2\right]\right]$$

$$+\frac{\eta^2 L}{\epsilon}\frac{\eta_l^2 C_2}{m^2}\sum_{t=1}^{T}\mathbb{E}\left[\left\|\sum_{i=1}^{m}\sum_{k=0}^{K-1}\nabla F_i(\theta_{t,k}^i)\right\|^2\right] + \frac{\eta^2 L}{\epsilon}\frac{TK\eta_l^2 C_2}{m}\sigma_l^2,$$

by applying Lemma F.6 into all terms containing the second moment estimate of model difference $\Delta_t$ in (D.14), using the fact that $\left(\sqrt{\beta_2\frac{(1+q^2)^3}{(1-q^2)^2}K^2 G^2 + \epsilon}\right)^{-1}\|\theta\| \leq \left(\sqrt{\beta_2\frac{(1+q^2)^3}{(1-q^2)}\eta_l^2 K^2 G^2 + \epsilon}\right)^{-1}\|\theta\| \leq \|\frac{\theta}{\sqrt{\beta_2\mathbf{v}+\epsilon}}\| \leq \epsilon^{-1/2}\|\theta\|$, and applying Lemma F.3 and F.13,

$$\mathbb{E}[f(\mathbf{z}_{T+1})] - f(\mathbf{z}_1)$$

$$\leq -\frac{\eta\eta_l K}{4\sqrt{4\beta_2\frac{(1+q^2)^3}{(1-q^2)^2}\eta_l^2 K^2 G^2 + \epsilon}}\sum_{t=1}^{T}\mathbb{E}[\|\nabla f(\theta_t)\|^2] + \frac{5\eta\eta_l^3 K^2 L^2 T}{\sqrt{2}\epsilon}(\sigma_l^2 + 6K\sigma_g^2)$$

$$+ \frac{C_1 \eta \eta_l K G^2 d}{\sqrt{\epsilon}} + \frac{2 C_1^2 \eta^2 \eta_l^2 K^2 L G^2 d}{\epsilon} + \left( \frac{3 \eta^2 L}{2} + C_2 \eta^2 L + \sqrt{2(1-\beta_2)} \eta G \right) \frac{KT \eta_l^2}{m \epsilon} \sigma_l^2 + \frac{\eta_l \eta T C^2 K^2 G^2}{\alpha^2 m^2 \epsilon}$$

$$- \sum_{t=1}^{T} \mathbb{E} \left[ \left\| \sum_{i=1}^{m} \sum_{k=0}^{K-1} \nabla F_i(\theta_{t,k}^i) \right\|^2 \right] \left[ \frac{\eta \eta_l}{2\sqrt{4 \beta_2 \frac{(1+q^2)^3}{(1-q^2)^2} \eta_l^2 K^2 G^2 + \epsilon} K m^2} - \left( \frac{3 \eta^2 L}{2} + C_2 \eta^2 L + \sqrt{2(1-\beta_2)} \eta G \right) \frac{\eta_l^2}{m^2 \epsilon} \right]$$

$$\leq - \frac{\eta \eta_l K}{4 C_0} \sum_{t=1}^{T} \mathbb{E}[\|\nabla f(\theta_t)\|^2] + \frac{5 \eta \eta_l^3 K^2 L^2 T}{\sqrt{2\epsilon}} (\sigma_l^2 + 6K \sigma_g^2)$$

$$+ \frac{C_1 \eta \eta_l K G^2 d}{\sqrt{\epsilon}} + \frac{2 C_1^2 \eta^2 \eta_l^2 K^2 L G^2 d}{\epsilon} + \left( \frac{3 \eta^2 L}{2} + C_2 \eta^2 L + \sqrt{2(1-\beta_2)} \eta G \right) \frac{KT \eta_l^2}{m \epsilon} \sigma_l^2,$$

where the last inequality holds by $\eta_l \leq \frac{\epsilon}{\sqrt{4 \beta_2 (1+q^2)^3 (1-q^2)^{-2} K^2 G^2 + \epsilon} \cdot K(3 \eta L + 2 C_2 \eta L + 2\sqrt{2(1-\beta_2)} G)}$.
Here

$$\frac{\eta \eta_l K}{4 \sqrt{4 \beta_2 \frac{(1+q^2)^3}{(1-q^2)^2} \eta_l^2 K^2 G^2 + \epsilon} \cdot T} \sum_{t=1}^{T} \mathbb{E}[\|\nabla f(\theta_t)\|^2]$$

$$\leq \frac{f(\mathbf{z}_0) - \mathbb{E}[f(\mathbf{z}_T)]}{T} + \frac{5 \eta \eta_l^3 K^2 L^2}{\sqrt{2\epsilon}} (\sigma_l^2 + 6K \sigma_g^2) + \frac{C_1 \eta \eta_l K G^2 d}{T \sqrt{\epsilon}} + \frac{2 C_1^2 \eta^2 \eta_l^2 K^2 L G^2 d}{T \epsilon}$$

$$+ \left[ 3 \eta^2 L + 2 C_2 \eta^2 L + 2\sqrt{2(1-\beta_2)} \eta G \right] \frac{K \eta_l^2}{2 m \epsilon} \sigma_l^2, \tag{D.15}$$

where $C_1 = \frac{\beta_1}{1-\beta_1} + \sqrt{\frac{12 q^2}{(1-q^2)^2} + \frac{(1-q^2)^2 C^2}{\alpha^2 m^2 q^2}}$ and $C_2 = \frac{\beta_1^2}{(1-\beta_1)^2} + \frac{4(q+\gamma+\frac{\lambda C}{\alpha m})^2}{(1-q^2)^2}$. (D.15 )also implies,

$$\min \mathbb{E}[\|\nabla f(\theta_t)\|^2] \leq 4 \sqrt{4 \beta_2 \frac{(1+q^2)^3}{(1-q^2)^2} \eta_l^2 K^2 G^2 + \epsilon} \left[ \frac{f_0 - f_*}{\eta \eta_l KT} + \frac{\Xi}{T} + \Omega \right],$$

where $\Xi = \frac{C_1 G^2 d}{\sqrt{\epsilon}} + \frac{2 C_1^2 \eta \eta_l K L G^2 d}{\epsilon}, \Omega = \frac{5 \eta^2 K L^2}{\sqrt{2\epsilon}} (\sigma_l^2 + 6K \sigma_g^2) + [(3 + 2 C_2) \eta L + 2\sqrt{2(1-\beta_2)} G] \frac{\eta_l}{2 m \eta \epsilon} \sigma_l^2, C_1 = \frac{\beta_1}{1-\beta_1} + \sqrt{\frac{12 q^2}{(1-q^2)^2} + \frac{(1-q^2)^2 C^2}{\alpha^2 m^2 q^2}}$ and $C_2 = \frac{\beta_1^2}{(1-\beta_1)^2} + \frac{4(q+\gamma+\frac{\lambda C}{\alpha m})^2}{(1-q^2)^2}$.

The proof of Theorem A.2 is similar to the above proof procedure and the detailed proof will not be given here.

## D.2 PROOF OF COROLLARY A.1

Let $\eta_l = \Theta(\frac{1}{\sqrt{T}K})$,$\eta = \Theta(\sqrt{Km})$ and $T = \mathcal{O}(Km)$ ,the convergence rate under full participation scheme is $\mathcal{O}(\frac{1}{T})$.

## D.3 PROOF OF THEOREM A.3

**Proof of Theorem A.3:**

Notations and equations: From the update rule of Algorithm 3, then $\mathbf{e}_1 = 0, \mathbf{e}_t = \frac{1}{m} \sum_{i=1}^{m} \mathbf{e}_t^i$ and $m_t = (1-\beta_1) \sum_{i=1}^{t} \beta_1^{t-i} \widehat{\Delta}_t^i$. Denote a global uncompressed difference $\Delta_t = \frac{1}{|S_t|} \sum_{i \in S_t} \Delta_t^i$. Denote a virtual momentum sequence: $\mathbf{m}_t' = \beta_1 \mathbf{m}_{t-1}' + (1-\beta_1) \Delta_t$, hence $\mathbf{m}_t' = (1-\beta_1) \sum_{i=1}^{t} \beta_1^{t-i} \Delta_i$. Define additional two virtual sequences $\Delta_t' = \frac{1}{n} \sum_{i=1}^{m} \Delta_t^i$ and $\widehat{\Delta}_t' = \frac{1}{n} \sum_{i=1}^{m} \widehat{\Delta}_t^i$. Note that when the client $i$ does not take part in the round of participation at step $t$, we have $\Delta_t^i = \widehat{\Delta}_t^i = 0$, therefore, $\Delta_t' = \Delta_t$ and $\widehat{\Delta}_t' = \widehat{\Delta}_t$.

By the aforementioned definition and notation, define a subset $\mathcal{S}_t = \{w_1^t, w_2^t, ..., w_n^t\}$, we have

$$\widehat{\Delta}_t - \Delta_t = \frac{1}{|\mathcal{S}_t|} \sum_{i \in \mathcal{S}_t} (\widehat{\Delta}_t^i - \Delta_t^i) = \frac{1}{n} \sum_{i=1}^{m} (\widehat{\Delta}_t^i - \Delta_t^i) = \frac{1}{n} \sum_{i=1}^{m} (\mathbf{e}_t^i - \mathbf{e}_{t+1}^i) = \mathbf{e}_t' - \mathbf{e}_{t+1}',$$

where the compression errors have the same structure, $\mathbf{e}_t' = \frac{1}{n} \sum_{i=1}^{m} \mathbf{e}_t^i$. Similar to the previous analysis, define the following sequence:

$$\Gamma_{t+1} := (1 - \beta_1) \sum_{\tau=1}^{t+1} \beta_1^{t+1-\tau} \mathbf{e}_\tau',$$

and keep using the Lyapunov function $\mathbf{z}_t$ from (D.4). For the expectation of model difference $\Delta_t$, then

$$\mathbb{E}_{\mathcal{S}_t}[\Delta_t] = \frac{1}{n} \mathbb{E}_{\mathcal{S}_t} \left[ \sum_{i=1}^{n} \Delta_t^{w_i} \right] = \mathbb{E}_{\mathcal{S}_t}[\Delta_t^{w_1}] = \frac{1}{m} \sum_{i=1}^{m} \Delta_t^i = \bar{\Delta}_t.$$

The proof of FedCAMS in partial participation settings has a similar outline combing the proof of partial participation in FedAMS and full participation in FedCAMS. By Assumption 3.1, then

$$\mathbb{E}[f(\mathbf{z}_{t+1})] - f(\mathbf{z}_t)$$

$$\leq \underbrace{\mathbb{E}\left[\left\langle \nabla f(\theta_t), \eta \widehat{\mathbf{V}}_t^{-1/2} \Delta_t \right\rangle\right]}_{T_1'}$$

$$\underbrace{- \mathbb{E}\left[\left\langle \nabla f(\mathbf{z}_t), \eta \frac{\beta_1}{1-\beta_1} \left(\widehat{\mathbf{V}}_{t-1}^{-1/2} - \widehat{\mathbf{V}}_t^{-1/2}\right) m_{t-1}' + \left(\widehat{\mathbf{V}}_{t-1}^{-1/2} - \widehat{\mathbf{V}}_t^{-1/2}\right) \Gamma_t \right\rangle\right]}_{T_2'}$$

$$+ \underbrace{\frac{\eta^2 L}{2} \mathbb{E}\left[\left\|\widehat{\mathbf{V}}_t^{-1/2} \Delta_t - \frac{\beta_1}{1-\beta_1} \left(\widehat{\mathbf{V}}_{t-1}^{-1/2} - \widehat{\mathbf{V}}_t^{-1/2}\right) m_{t-1}' - \left(\widehat{\mathbf{V}}_{t-1}^{-1/2} - \widehat{\mathbf{V}}_t^{-1/2}\right) \Gamma_t \right\|^2\right]}$$

$$+ \underbrace{\mathbb{E}\left[\left\langle \nabla f(\mathbf{z}_t) - \nabla f(\theta_t), \eta \widehat{\mathbf{V}}_t^{-1/2} \Delta_t \right\rangle\right]}_{T_4'}$$

Note that the bound for $T_2'$ is exactly the same as the bound for $T_2$. For the three corresponding terms, $T_1', T_3'$ and $T_4'$ which include the second-order momentum estimate of $\Delta_t$. For $T_1'$, similar to the full participation settings,

$$T_1' \leq \sqrt{2}\mathbb{E}\left[\left\langle \nabla f(\theta_t), \eta \frac{\Delta_t}{\sqrt{\beta_2 \mathbf{v}_{t-1} + \epsilon}} \right\rangle\right] + \sqrt{2}\eta \mathbb{E}\left[\left\langle \nabla f(\theta_t), \frac{\Delta_t}{\sqrt{\mathbf{v}_t + \epsilon}} - \frac{\Delta_t}{\sqrt{\beta_2 \mathbf{v}_{t-1} + \epsilon}} \right\rangle\right].$$
(D.16)

The first term in (D.16) does not change in partial participation scheme. The second term is changed due to the variance of $\Delta_t$ changes. For the second term of $T_1'$, then

$$\sqrt{2}\eta \mathbb{E}\left[\left\langle \nabla f(\theta_t), \frac{\Delta_t}{\sqrt{\mathbf{v}_t + \epsilon}} - \frac{\Delta_t}{\sqrt{\beta_2 \mathbf{v}_{t-1} + \epsilon}} \right\rangle\right] \leq \frac{\sqrt{2(1-\beta_2)}\eta G}{\epsilon} \mathbb{E}[\|\Delta_t\|^2].$$

For $T_3'$, similar to the proof of $T_3$, we have

$$\sum_{t=1}^{T} T_3' \leq \frac{\eta^2 L}{\epsilon} \sum_{t=1}^{T} \mathbb{E}[\|\Delta_t\|^2] + \eta^2 L C_1^2 \eta_l^2 K^2 G^2 \sum_{t=1}^{T} \mathbb{E}\left[\left\|\widehat{\mathbf{V}}_{t-1}^{-1/2} - \widehat{\mathbf{V}}_t^{-1/2}\right\|^2\right],$$

where $C_1 = \frac{\beta_1}{1-\beta_1} + \frac{m}{n}\sqrt{\frac{12q^2}{(1-q^2)^2} + \frac{(1-q^2)^2 C^2}{\alpha^2 n^2 q^2}}$ in partial participation,

$$
\begin{aligned}
T_4' &= \eta\mathbb{E}\left[\left\langle f(\mathbf{z}_t) - f(\theta_t), \widehat{\mathbf{V}}_t^{-1/2}\Delta_t\right\rangle\right] \\
&\leq \eta\mathbb{E}\left[\|f(\mathbf{z}_t) - f(\theta_t)\|\left\|\widehat{\mathbf{V}}_t^{-1/2}\Delta_t\right\|\right] \\
&\leq \eta^2 L\mathbb{E}\left[\left\|\frac{\beta_1}{1-\beta_1}\widehat{\mathbf{V}}_{t-1}^{-1/2}\mathbf{m}_{t-1}' + \widehat{\mathbf{V}}_{t-1}^{-1/2}\Gamma_t\right\|\left\|\widehat{\mathbf{V}}_t^{-1/2}\Delta_t\right\|\right] \\
&\leq \frac{C_1\eta^2\eta_l^2 K^2 LG^2}{\epsilon}.
\end{aligned}
$$

The summation from $T_1'$ to $T_4'$ over total iteration T is:

$$
\mathbb{E}[f(\mathbf{z}_{T+1})] - f(\mathbf{z}_1) = \sum_{t=1}^{T}[T_1' + T_2' + T_3' + T_4']
$$

$$
\begin{aligned}
&\leq -\frac{\eta\eta_l K}{4}\sum_{t=1}^{T}\mathbb{E}\left[\left\|\frac{\nabla f(\theta_t)}{\sqrt[4]{\beta_2\mathbf{v}_{t-1}+\epsilon}}\right\|^2\right] + \frac{5\eta\eta_l^3 K^2 L^2 T}{\sqrt{2}\epsilon}(\sigma_l^2 + 6K\sigma_g^2) + \frac{\sqrt{2(1-\beta_2)}\eta G}{\epsilon}\sum_{t=1}^{T}\mathbb{E}[\|\Delta_t\|^2] \\
&\quad - \frac{\eta\eta_l}{2Km^2}\sum_{t=1}^{T}\mathbb{E}\left[\left\|\frac{1}{\sqrt[4]{\beta_2\mathbf{v}_{t-1}+\epsilon}}\sum_{i=1}^{m}\sum_{k=0}^{K-1}\nabla F_i(\theta_t))\right\|^2\right] + C_1\eta\eta_l KG^2\sum_{t=1}^{T}\mathbb{E}\left[\left\|\widehat{\mathbf{V}}_{t-1}^{-1/2} - \widehat{\mathbf{V}}_t^{-1/2}\right\|_1\right] \\
&\quad + C_1^2\eta^2\eta_l^2 K^2 LG^2\epsilon^{-1/2}\sum_{t=1}^{T}\mathbb{E}\left[\left\|\widehat{\mathbf{V}}_{t-1}^{-1/2} - \widehat{\mathbf{V}}_t^{-1/2}\right\|_1\right] + C_1^2\eta^2\eta_l^2 K^2 LG^2\sum_{t=1}^{T}\mathbb{E}\left[\left\|\widehat{\mathbf{V}}_{t-1}^{-1/2} - \widehat{\mathbf{V}}_t^{-1/2}\right\|^2\right] \\
&\quad + \frac{\eta^2 L}{\epsilon}\sum_{t=1}^{T}\mathbb{E}[\|\Delta_t\|^2] + \frac{C_1 T\eta^2\eta_l^2 K^2 LG^2}{\epsilon}
\end{aligned}
$$

$$
\begin{aligned}
&\leq -\frac{\eta\eta_l K}{4\sqrt{4\beta_2\frac{(1+q^2)^3}{(1-q^2)^2}\eta_l^2 K^2 G^2 + \epsilon}}\sum_{t=1}^{T}\mathbb{E}[\|\nabla f(\theta_t)\|^2] + \frac{5\eta\eta_l^3 K^2 L^2 T}{\sqrt{2}\epsilon}(\sigma_l^2 + 6K\sigma_g^2) + \frac{C_1\eta\eta_h KG^2 d}{T\sqrt{\epsilon}} \\
&\quad + \frac{2C_1^2\eta^2\eta_l^2 K^2 LG^2 d}{T\epsilon} - \frac{\eta\eta_l}{2\sqrt{4\beta_2\frac{(1+q^2)^3}{(1-q^2)^2}\eta_l^2 K^2 G^2 + \epsilon}Km^2}\sum_{t=1}^{T}\mathbb{E}\left[\left\|\sum_{i=1}^{m}\sum_{k=0}^{K-1}\nabla F_i(\theta_t))\right\|^2\right] \\
&\quad + \left(\frac{\eta^2\eta_l^2 LKT}{n\epsilon} + \frac{\sqrt{2(1-\beta_2)}\eta\eta_l^2 KTG}{n\epsilon}\right)\sigma_l^2 + \frac{C_1 T\eta^2\eta_l^2 K^2 LG^2}{\epsilon} \\
&\quad + \left(\frac{\eta^2\eta_l^2 L}{\epsilon} + \frac{\sqrt{2(1-\beta_2)}\eta\eta_l^2 G}{\epsilon}\right)\frac{m-n}{mn(m-1)}\left[15mK^3 L^3\eta_l^2(\sigma_l^2 + 6K\sigma_g^2)T\right] \\
&\quad + (90mK^4 L^2\eta_l^2 + 3mK^2)\sum_{t=1}^{T}\mathbb{E}[\|\nabla f(\theta_t)\|^2] + 3mK^2 T\sigma_g^2\Big] \\
&\quad + \left(\eta^2\eta_l^2 L + \sqrt{2(1-\beta_2)}\eta\eta_l^2 G\right)\frac{n-1}{mn(m-1)}\sum_{t=1}^{T}\mathbb{E}\left[\left\|\sum_{i=1}^{m}\sum_{k=0}^{K-1}\nabla F_i(\theta_t))\right\|^2\right].
\end{aligned}
$$

The proof outline is similar with previous proof. We take the use of Lemma F.3,F.9,F.13 for corresponding terms. By additional constraints of local learning rate $\eta_n$ with the inequality $[\eta^2 L + \sqrt{2(1-\beta_2)}\eta G]\frac{\eta_l^2(n-1)}{mn(m-1)\epsilon} - \frac{\eta\eta_l}{2Km^2}\left[\sqrt{4\beta_2\frac{(1+q^2)^3}{(1-q^2)^2}\eta_l^2 K^2 G^2 + \epsilon}\right]^{-1} \leq 0$,we obtain the constraint $\eta_l \leq \frac{n(m-1)}{m(n-1)}\frac{\epsilon}{2K\sqrt{4\beta_2(1+q^2)^3(1-q^2)^{-2}K^2 G^2 + \epsilon}[\eta L + \sqrt{2(1-\beta_2)G}]}$,and we further need $\eta_l$ satis-

fies $\frac{\eta\eta_l K}{4\sqrt{4\beta_2(1+q^2)^3(1-q^2)^{-2}\eta_l^2 K^2 G^2+\epsilon}} - (\eta^2 L + \sqrt{2(1-\beta_2)}\eta G)\frac{\eta_l^2(m-n)}{mn(m-1)\epsilon}(90mK^4 L^2\eta_l^2 + 3mK^2) \geq$ $\frac{\eta\eta_l K}{8\sqrt{4\beta_2(1+q^2)^3(1-q^2)^{-2}\eta_l^2 K^2 G^2+\epsilon}}$. Hence for the convergence rate,then

$$\frac{\eta\eta_l K}{8\sqrt{4\beta_2\frac{(1+q^2)^3}{(1-q^2)^2}\eta_l^2 K^2 G^2+\epsilon}\cdot T}\sum_{i=1}^{T}\mathbb{E}[\|\nabla f(\theta_t)\|^2]$$

$$\leq \frac{f(\mathbf{z}_0)-\mathbb{E}[f(\mathbf{z}_T)]}{T} + \frac{5\eta\eta_l^3 K^2 L^2}{\sqrt{2\epsilon}}(\sigma_l^2 + 6K\sigma_g^2) + \left(\eta L + \sqrt{2(1-\beta_2)}G\right)\frac{\eta\eta_l^2 K}{n\epsilon}\sigma_l^2$$

$$+ \frac{C_1\eta\eta_l KG^2 d}{T\sqrt{\epsilon}} + \frac{2C_1^2\eta^2\eta_l^2 K^2 LG^2 d}{T\epsilon} + \frac{C_1\eta^2\eta_l^2 K^2 LG^2}{\epsilon}$$

$$+ \left(\frac{\eta^2\eta_l^2 L}{\epsilon} + \frac{\sqrt{2(1-\beta_2)}\eta\eta_l^2 G}{\epsilon}\right)\frac{m-n}{mn(m-1)}[15mK^3 L^2\eta_l^2(\sigma_l^2 + 6K\sigma_g^2) + 3mK^2\sigma_g^2]$$

.

Therefore

$$\min\mathbb{E}[\|\nabla f(\theta_t)\|^2] \leq 8\sqrt{4\beta_2\frac{(1+q^2)^3}{(1-q^2)^2}\eta_l^2 K^2 G^2+\epsilon}\left[\frac{f_0-f_*}{\eta\eta_l KT} + \frac{\Xi}{T} + \Omega\right]$$

, where $\Xi = \frac{C_1 G^3 d}{\sqrt{\epsilon}} + \frac{2C_1^2\eta\eta_l KLG^2 d}{\epsilon}$, $\Omega = \frac{C_1\eta\eta_l KLG^2}{\epsilon} + \frac{5\eta^2 KL^2}{\sqrt{2\epsilon}}(\sigma_l^2 + 6K\sigma_g^2) + [\eta L + \sqrt{2(1-\beta_2)}G]\frac{\eta_l}{\eta n\epsilon}\sigma_l^2 + [\eta L + \sqrt{2(1-\beta_2)}G]\frac{\eta_l(m-n)}{n(m-1)\epsilon}[15K^2 L^2\eta_l^2(\sigma_l^2 + 6K\sigma_g^2) + 3K\sigma_g^2]$ and $C_1 = \frac{\beta_1}{1-\beta_1} + \frac{m}{n}\sqrt{\frac{12q^2}{(1-q^2)^2} + \frac{(1-q^2)^2 C^2}{\alpha^2 n^2 q^2}}$.

The proof of Theorem A.4 is similar to the above proof procedure and the detailed proof will not be given here.

### D.4 PROOF OF COROLLARY A.2

If choose $\eta_l = \Theta(\frac{1}{\sqrt{TK}})$ and $\eta = \Theta(\sqrt{Kn})$,we get $\min_{t\in[T]}\mathbb{E}[\|\nabla f(\theta_t)\|^2] = \mathcal{O}(\frac{\sqrt{K}}{\sqrt{Tn}})$.

## E PROOF OF THEOREMS IN SECTION 4.3 AND PARTIAL PARTICIPATION SETTING FOR FEDBNLACA,FEDBACA

### E.1 PROOF OF THEOREM 4.5

Notations and equations: From the update rule of Algorithm 2, we get $\mathbf{e}_1 = 0, \mathbf{e}_t = \frac{1}{m}\sum_{i=1}^{m}\mathbf{e}_t^i$ and $\mathbf{m}_t = (1-\beta_1)\sum_{i=1}^{t}\tilde{\beta}_1^{t-i}\widehat{\widehat{\Delta}}_i$. Denote a global uncompressed difference $\Delta_t = \frac{1}{m}\sum_{i=1}^{m}\Delta_t^i$. Denote a virtual momentum sequence: $\mathbf{m}_t' = \beta_1\mathbf{m}_{t-1}' + (1-\beta_1)\Delta_t$, hence we have $\mathbf{m}_t' = (1-\beta_1)\sum_{i=1}^{t}\beta_1^{t-i}\Delta_i$. By the aforementioned definition and notation, then

$$\widehat{\Delta}_t - \Delta_t = \frac{1}{m}\sum_{i=1}^{m}(\widehat{\widehat{\Delta}}_t^i - \Delta_t^i) = \frac{1}{m}\sum_{i=1}^{m}(\widehat{\Delta}_t^i - \Delta_t^i) - \frac{1}{M_t}\sum_{i\in M_t}\mathcal{C}(q_t^i)$$

$$= \frac{1}{m}\sum_{i=1}^{m}(\mathbf{e}_t^i - \mathbf{e}_{t+1}^i) - \frac{1}{M_t}\sum_{i\in M_t}\mathcal{C}(q_t^i) = \mathbf{e}_t - \mathbf{e}_{t+1} - \frac{1}{M_t}\sum_{i\in M_t}\mathcal{C}(q_t^i). \quad (E.1)$$

Denote the weighted averaging error sequence $\Gamma_t = (1-\beta_1)\sum_{\tau=1}^{t}\beta_1^{t-\tau}\mathbf{e}_r$, with the input $\mathbf{e}_1 = 0$, we obtain the relation between $\Gamma_t$ and $\mathbf{m}_t$ as follows

$$\mathbf{m}_t - \mathbf{m}_t' = (1 - \beta_1) \sum_{\tau=1}^{t} \beta_1^{t-\tau} (\widehat{\Delta}_\tau - \Delta_\tau) = (1 - \beta_1) \sum_{\tau=1}^{t} \beta_1^{t-\tau} (\mathbf{e}_\tau - \mathbf{e}_{\tau+1}) - (1 - \beta_1) \sum_{\tau=1}^{t} \beta_1^{t-\tau} \frac{1}{M_t} \sum_{i \in M_t} \mathcal{C}(q_t^i)$$

$$= \Gamma_t - \Gamma_{t+1} - (1 - \beta_1) \sum_{\tau=1}^{t} \beta_1^{t-\tau} \frac{1}{M_t} \sum_{i \in M_t} \mathcal{C}(q_t^i), \tag{E.2}$$

where the last step holds due to $\Gamma_{t+1} = (1 - \beta_1) \sum_{\tau=1}^{t+1} \beta_1^{t-\tau} \mathbf{e}_{\tau+1} = (1 - \beta_1) \sum_{\tau=1}^{t} \beta_1^{t-\tau} \mathbf{e}_{\tau+1} + \beta_1^t \mathbf{e}_1$. Similar to previous works studied adaptive methods, we introduce a Lyapunov sequence $z_t$: assume $\theta_0 = \theta_1$, for each $t \geq 1$,

$$\mathbf{z}_t = \theta_t + \frac{\beta_1}{1 - \beta_1} (\theta_t - \theta_{t-1}) = \frac{1}{1 - \beta_1} \theta_t - \frac{\beta_1}{1 - \beta_1} \theta_{t-1}.$$

Therefore, by the update rule of $\theta_t$,

$$\mathbf{y}_{t+1} = \theta_{t+1} + \eta \frac{\beta_1}{1 - \beta_1} \hat{\mathbf{V}}_t^{-1/2} \mathbf{m}_t - \eta \frac{\beta_1}{1 - \beta_1} \hat{\mathbf{V}}_t^{-1/2} (\widehat{\overline{\theta}}_t - \theta_t) + (1 - \beta_1) \sum_{\tau=1}^{t} \beta_1^{t-\tau} \frac{1}{M_t} \sum_{i \in M_t} \mathcal{C}(q_t^i)$$

$$= \theta_{t+1} + \eta \frac{\beta_1}{1 - \beta_1} \widehat{\mathbf{V}}_t^{-1/2} [\mathbf{m}_t' + \Gamma_t - \Gamma_{t+1}] - (1 - \beta_1) \sum_{\tau=1}^{t} \beta_1^{t-\tau} \frac{1}{M_t} \sum_{i \in M_t} \mathcal{C}(q_t^i) + \eta \frac{\beta_1}{1 - \beta_1} \widehat{\mathbf{V}}_t^{-1/2} (\widehat{\overline{\theta}}_t - \theta_t)$$

$$= \theta_{t+1} + \eta \frac{\beta_1}{1 - \beta_1} \widehat{\mathbf{V}}_t^{-1/2} \mathbf{m}_t' + \eta \frac{\beta_1}{1 - \beta_1} \widehat{\mathbf{V}}_t^{-1/2} \left[ \frac{\Gamma_{t+1} - (1 - \beta_1) \mathbf{e}_{t+1}}{\beta_1} - \Gamma_{t+1} \right]$$

$$- (1 - \beta_1) \sum_{\tau=1}^{t} \beta_1^{t-\tau} \frac{1}{M_t} \sum_{i \in M_t} \mathcal{C}(q_t^i) - \eta \frac{\beta_1}{1 - \beta_1} \hat{\mathbf{V}}_t^{-1/2} (\widehat{\overline{\theta}}_t - \theta_t)$$

$$= \theta_{t+1} + \eta \frac{\beta_1}{1 - \beta_1} \hat{\mathbf{V}}_t^{-1/2} \mathbf{m}_t' + \eta \hat{\mathbf{V}}_t^{-1/2} \Gamma_{t+1} - \eta \hat{\mathbf{V}}_t^{-1/2} \mathbf{e}_{t+1} - (1 - \beta_1) \sum_{\tau=1}^{t} \beta_1^{t-\tau} \frac{1}{M_t} \sum_{i \in M_t} \mathcal{C}(q_t^i)$$

$$+ \eta \frac{\beta_1}{1 - \beta_1} \hat{\mathbf{V}}_t^{-1/2} (\widehat{\overline{\theta}}_t - \theta_t). \tag{E.3}$$

The third equation holds due to the fact that $\Gamma_{t+1} = \beta_1 \Gamma_t + (1 - \beta_1) \mathbf{e}_{t+1}$. We then introduce a new sequence based on the previous Lyapunov sequence $\mathbf{y}_t$ as follows

$$\mathbf{z}_{t+1} = \mathbf{y}_{t+1} + (1 - \beta_1) \sum_{\tau=1}^{t} \beta_1^{t-\tau} \frac{1}{M_t} \sum_{i \in M_t} \mathcal{C}(q_t^i) + \eta \hat{\mathbf{V}}_t^{-1/2} \mathbf{e}_{t+1}$$

$$= \theta_{t+1} + \eta \frac{\beta_1}{1 - \beta_1} \hat{\mathbf{V}}_t^{-1/2} \mathbf{m}_t' + \eta \hat{\mathbf{V}}_t^{-1/2} \Gamma_{t+1} + \eta \frac{\beta_1}{1 - \beta_1} \hat{\mathbf{V}}_t^{-1/2} (\widehat{\overline{\theta}}_t - \theta_t). \tag{E.4}$$

The sequence difference $\mathbf{z}_{t+1} - \mathbf{z}_t$ can be represented by

$$\mathbf{z}_{t+1} - \mathbf{z}_t = \theta_{t+1} - \theta_t + \eta \frac{\beta_1}{1 - \beta_1} \widehat{\mathbf{V}}_t^{-1/2} \mathbf{m}_t' - \eta \frac{\beta_1}{1 - \beta_1} \widehat{\mathbf{V}}_{t-1}^{-1/2} \mathbf{m}_{t-1}'$$

$$+ \eta \widehat{\mathbf{V}}_t^{-1/2} \Gamma_{t+1} - \eta \widehat{\mathbf{V}}_{t-1}^{-1/2} \Gamma_t + \eta \frac{1}{1 - \beta_1} \hat{\mathbf{V}}_t^{-1/2} (\widehat{\overline{\theta}}_t - \theta_t)$$

$$= \eta \hat{\mathbf{V}}_t^{-1/2} \mathbf{m}_t + \eta \hat{\mathbf{V}}_t^{-1/2} \Gamma_{t+1} + \eta \frac{\beta_1}{1 - \beta_t} \hat{\mathbf{V}}_t^{-1/2} \mathbf{m}_t'$$

$$- \eta \frac{\beta_1}{1 - \beta_t} \hat{\mathbf{V}}_{t-1}^{-1/2} \mathbf{m}_{t-1}' - \eta \hat{\mathbf{V}}_{t-1}^{-1/2} \Gamma_t + \eta \frac{1}{1 - \beta_1} \hat{\mathbf{V}}_t^{-1/2} (\widehat{\overline{\theta}}_t - \theta_t),$$

where the second equation follows the update rule of $\theta_{t+1}$. Following (E.2), then combining likely terms and applying the definition of $\mathbf{m}_t'$, we have

$$\mathbf{z}_{t+1} - \mathbf{z}_t = \eta \hat{\mathbf{V}}_t^{-1/2} \mathbf{m}_t' + \eta \hat{\mathbf{V}}_t^{-1/2} \Gamma_t + \eta \frac{\beta_1}{1-\beta_1} \hat{\mathbf{V}}_t^{-1/2} \mathbf{m}_t'$$

$$- \eta \frac{\beta_1}{1-\beta_1} \hat{\mathbf{V}}_{t-1}^{-1/2} \mathbf{m}_{t-1}' - \eta \hat{\mathbf{V}}_{t-1}^{-1/2} \Gamma_t + \eta \frac{1}{1-\beta_1} \hat{\mathbf{V}}_t^{-1/2} (\widehat{\bar{\theta}}_t - \theta_t)$$

$$= \eta \frac{1}{1-\beta_1} \hat{\mathbf{V}}_t^{-1/2} \mathbf{m}_t' - \eta \frac{\beta_1}{1-\beta_1} \hat{\mathbf{V}}_{t-1}^{-1/2} \mathbf{m}_{t-1}' + \eta \hat{\mathbf{V}}_t^{-1/2} \Gamma_t + \eta \hat{\mathbf{V}}_{t-1}^{-1/2} \Gamma + \eta \frac{1}{1-\beta_1} \hat{\mathbf{V}}_t^{-1/2} (\widehat{\bar{\theta}}_t - \theta_t),$$

$$= \eta \frac{1}{1-\beta_1} \hat{\mathbf{V}}_t^{-1/2} [\beta_1 \mathbf{m}_{t-1}' + (1-\beta_1)\Delta_t] - \eta \frac{\beta_1}{1-\beta_1} \hat{\mathbf{V}}_{t-1}^{-1/2} \mathbf{m}_{t-1}'$$

$$+ \eta \hat{\mathbf{V}}_t^{-1/2} \Gamma_t - \eta \hat{\mathbf{V}}_{t-1}^{-1/2} \Gamma_t + \eta \frac{1}{1-\beta_1} \hat{\mathbf{V}}_t^{-1/2} (\widehat{\bar{\theta}}_t - \theta_t)$$

$$= \eta \hat{\mathbf{V}}_t^{-1/2} \Delta_t - \eta \frac{\beta_1}{1-\beta_1} \left( \hat{\mathbf{V}}_{t-1}^{-1/2} - \hat{\mathbf{V}}_t^{-1/2} \right) \mathbf{m}_{t-1}' - \eta \left( \hat{\mathbf{V}}_{t-1}^{-1/2} - \hat{\mathbf{V}}_t^{-1/2} \right) \Gamma_t + \eta \frac{1}{1-\beta_1} \hat{\mathbf{V}}_t^{-1/2} (\widehat{\bar{\theta}}_t - \theta_t).$$

Therefore, we obtain a helpful Lyapunov sequence for our proof of FedCAMS. The proof of Fed-CAMS in full participation settings has a similar outline with the proof of FedAMS. By Assumption 3.1,

$$\mathrm{E}[f(\mathbf{z}_{t+1})] - f(\mathbf{z}_t)$$

$$\leq \mathbb{E}[\langle \nabla f(\mathbf{z}_t), \mathbf{z}_{t+1} - \mathbf{z}_t \rangle] + \frac{L}{2} \mathbb{E}[\|\mathbf{z}_{t+1} - \mathbf{z}_t\|^2]$$

$$\leq \mathbb{E}\left[ \left\langle \nabla f(\mathbf{z}_t), \eta \hat{\mathbf{V}}_t^{-1/2} \Delta_t \right\rangle \right]$$

$$- \mathbb{E}\left[ \left\langle \nabla f(\mathbf{z}_t), \eta \frac{\beta_1}{1-\beta_1} \left( \hat{\mathbf{V}}_{t-1}^{-1/2} - \hat{\mathbf{V}}_t^{-1/2} \right) \mathbf{m}_{t-1}' + \eta \left( \hat{\mathbf{V}}_{t-1}^{-1/2} - \hat{\mathbf{V}}_t^{-1/2} \right) \Gamma_t \right\rangle \right]$$

$$+ \frac{\eta^2 L}{2} \mathbb{E}\left[ \left\| \hat{\mathbf{V}}_t^{-1/2} \Delta_t - \frac{\beta_1}{1-\beta_1} \left( \hat{\mathbf{V}}_{t-1}^{-1/2} - \hat{\mathbf{V}}_t^{-1/2} \right) \mathbf{m}_{t-1}' - \left( \hat{\mathbf{V}}_{t-1}^{-1/2} - \hat{\mathbf{V}}_t^{-1/2} \right) \Gamma_t \right\|^2 \right]$$

$$= \underbrace{\mathbb{E}\left[ \left\langle \nabla f(\theta_t), \eta \hat{\mathbf{V}}_t^{-1/2} \Delta_t \right\rangle \right]}_{T_1} \underbrace{- \eta \mathbb{E}\left[ \left\langle \nabla f(\mathbf{z}_t), \frac{\beta_1}{1-\beta_1} \left( \hat{\mathbf{V}}_{t-1}^{-1/2} - \hat{\mathbf{V}}_t^{-1/2} \right) \mathbf{m}_{t-1}' + \left( \hat{\mathbf{V}}_{t-1}^{-1/2} - \hat{\mathbf{V}}_t^{-1/2} \right) \Gamma_t \right\rangle \right]}_{T_2}$$

$$+ \underbrace{\frac{\eta^2 L}{2} \mathbb{E}\left[ \left\| \hat{\mathbf{V}}_t^{-1/2} \Delta_t - \frac{\beta_1}{1-\beta_1} \left( \hat{\mathbf{V}}_{t-1}^{-1/2} - \hat{\mathbf{V}}_t^{-1/2} \right) \mathbf{m}_{t-1}' - \left( \hat{\mathbf{V}}_{t-1}^{-1/2} - \hat{\mathbf{V}}_t^{-1/2} \right) \Gamma_t \right\|^2 \right]}_{T_3}$$

$$+ \underbrace{\mathbb{E}\left[ \left\langle \nabla f(\mathbf{z}_t) - \nabla f(\theta_t), \eta \hat{\mathbf{V}}_t^{-1/2} \Delta_t \right\rangle \right]}_{T_4} + \underbrace{\mathbb{E}\left[ \left\langle \nabla f(\mathbf{z}_t), \frac{\eta}{1-\beta_1} \hat{\mathbf{V}}_t^{-1/2} (\widehat{\bar{\theta}}_t - \theta_t) \right\rangle \right]}_{T_5} + \underbrace{\frac{\eta^2 L^2}{(1-\beta)^2} \mathbb{E}\left[ \left\| \hat{\mathbf{V}}_t^{-1/2} (\widehat{\bar{\theta}}_t - \theta_t) \right\|^2 \right]}_{T_6}.$$

$$\text{(E.5)}$$

here recall the notation $\hat{\mathbf{V}}_t = \mathrm{diag}(\widehat{\mathbf{v}}_t) = \mathrm{diag}(\max(\widehat{\mathbf{v}}_{t-1}, \mathbf{v}_t, \epsilon))$.

**Bounding $T_1$:**,

$$
\mathrm{T1} = \mathbb{E}\left[\left\langle \nabla f(\theta_t), \eta \frac{\Delta_t}{\sqrt{\hat{\mathbf{v}}_t}} \right\rangle\right]
$$

$$
\leq \eta\mathbb{E}\left[\left\langle \nabla f(\theta_t), \frac{\sqrt{2}\cdot\Delta_t}{\sqrt{\mathbf{v}_t}+\epsilon} \right\rangle\right]
$$

$$
= \sqrt{2}\eta\mathbb{E}\left[\left\langle \nabla f(\theta_t), \frac{\Delta_t}{\sqrt{\beta_2\mathbf{v}_{t-1}}+\epsilon} \right\rangle\right] + \sqrt{2}\eta\mathbb{E}\left[\left\langle \nabla f(\theta_t), \frac{\Delta_t}{\sqrt{\mathbf{v}_t}+\epsilon} - \frac{\Delta_t}{\sqrt{\beta_2\mathbf{v}_{t-1}}+\epsilon} \right\rangle\right],
\tag{E.6}
$$

where the first inequality follows by the fact that $\hat{v}_t \geq \frac{v_t+\epsilon}{2}$. For the second term in (E.6),

$$
\sqrt{2}\cdot\eta\mathbb{E}\left[\left\langle \nabla f(\theta_t), \frac{\Delta_t}{\sqrt{\mathbf{v}_t}+\epsilon} - \frac{\Delta_t}{\sqrt{\beta_2\mathbf{v}_{t-1}}+\epsilon} \right\rangle\right]
$$

$$
\leq \sqrt{2}\cdot\eta\cdot\mathbb{E}\|\nabla f(\theta_t)\|\mathbb{E}\left[\left\|\frac{1}{\sqrt{\mathbf{v}_t}+\epsilon} - \frac{1}{\sqrt{\beta_2\mathbf{v}_{t-1}}+\epsilon}\right\| \cdot \|\Delta_t\|\right]
$$

$$
\leq \frac{\eta\sqrt{2(1-\beta_2)}G}{\epsilon}\mathbb{E}[\|\Delta_t\|^2],
$$

where the second inequality follows from Lemma F.1 and F.5, and we will further apply the bound for $E[||\Delta_t||^2]$ by applying Lemma F.7. For the first term in (E.6),

$$
\sqrt{2}\cdot\eta\mathbb{E}\left[\left\langle \nabla f(\theta_t), \frac{\Delta_t}{\sqrt{\beta_2\mathbf{v}_{t-1}}+\epsilon} \right\rangle\right]
$$

$$
= \sqrt{2}\cdot\eta\mathbb{E}\left[\left\langle \frac{\nabla f(\theta_t)}{\sqrt{\beta_2\mathbf{v}_{t-1}}+\epsilon}, \Delta_t + \eta K\nabla f(\theta_t) - \eta_t K\nabla f(\theta_t) \right\rangle\right]
$$

$$
= -\sqrt{2}\eta\eta_l K\mathbb{E}\left[\left\|\frac{\nabla f(\theta_t)}{\sqrt[4]{\beta_2\mathbf{v}_{t-1}}+\epsilon}\right\|^2\right] + \sqrt{2}\eta\mathbb{E}\left[\left\langle \frac{\nabla f(\theta_t)}{\sqrt{\beta_2\mathbf{v}_{t-1}}+\epsilon}, \Delta_t + \eta_l K\nabla f(\theta_t) \right\rangle\right]
$$

$$
= -\sqrt{2}\eta\eta_t K\mathbb{E}\left[\left\|\frac{\nabla f(\theta_t)}{\sqrt[4]{\beta_2\mathbf{v}_{t-1}}+\epsilon}\right\|^2\right] + \sqrt{2}\eta\left\langle \frac{\nabla f(\theta_t)}{\sqrt{\beta_2\mathbf{v}_{t-1}}+\epsilon}, \mathbb{E}\left[-\frac{\eta_l}{m}\sum_{i=1}^{m}\sum_{k=0}^{K-1}\mathbf{g}_{t,k}^i + \eta_l K\nabla f(\theta_t)\right] \right\rangle
$$

$$
= -\sqrt{2}\eta\eta_l K\mathbb{E}\left[\left\|\frac{\nabla f(\theta_t)}{\sqrt[4]{\beta_2\mathbf{v}_{t-1}}+\epsilon}\right\|^2\right] + \sqrt{2}\eta\left\langle \frac{\nabla f(\theta_t)}{\sqrt{\beta_2\mathbf{v}_{t-1}}+\epsilon}, \mathbb{E}\left[-\frac{\eta_l}{m}\sum_{i=1}^{m}\sum_{k=0}^{K-1}\mathbf{g}_{t,k}^i + \frac{\eta_l K}{m}\sum_{i=1}^{m}\nabla F_i(\theta_t)\right] \right\rangle
\tag{E.7}
$$

For the last term in (E.7), we get

$$\sqrt{2}\eta \left\langle \frac{\nabla f(\theta_t)}{\sqrt{\beta_2 \mathbf{v}_{t-1} + \epsilon}}, \mathbb{E}\left[ -\frac{\eta_l}{m}\sum_{i=1}^{m}\sum_{k=0}^{K-1}\mathbf{g}_{t,k}^i + \frac{\eta_l K}{m}\sum_{i=1}^{m}\nabla F_i(\theta_t) \right] \right\rangle$$

$$= \sqrt{2}\eta \left\langle \frac{\sqrt{\eta K}}{\sqrt[4]{\beta_2 \mathbf{v}_{t-1} + \epsilon}}\nabla f(\theta_t), -\frac{\sqrt{\eta_n K}}{Km}\frac{1}{\sqrt[4]{\beta_2 \mathbf{v}_{t-1} + \epsilon}}\mathbb{E}\left[ \sum_{i=1}^{m}\sum_{k=0}^{K-1}(\nabla F_i(\theta_{t,k}^i) - \nabla F_i(\theta_t)) \right] \right\rangle$$

$$= \frac{\sqrt{2}\eta\eta_l K}{2}\left\| \frac{\nabla f(\theta_t)}{\sqrt[4]{\beta_2 \mathbf{v}_{t-1} + \epsilon}} \right\|^2 + \frac{\sqrt{2}\eta\eta_l}{2Km^2}\mathbb{E}\left\| \frac{1}{\sqrt[4]{\beta_2 \mathbf{v}_{t-1} + \epsilon}}\sum_{i=1}^{m}\sum_{k=0}^{K-1}(\nabla F_i(\theta_{t,k}^i) - \nabla F_i(\theta_t)) \right\|^2$$

$$- \frac{\sqrt{2}\eta\eta_l}{2Km^2}\mathbb{E}\left[ \left\| \frac{1}{\sqrt[4]{\beta_2 \mathbf{v}_{t-1} + \epsilon}}\sum_{i=1}^{m}\sum_{k=0}^{K-1}\nabla F_i(\theta_{t,k}^i) \right\|^2 \right]$$

$$\leq \frac{\sqrt{2}\eta\eta_l K}{2}\left\| \frac{\nabla f(\theta_t)}{\sqrt[4]{\beta_2 \mathbf{v}_{t-1} + \epsilon}} \right\|^2 + \frac{\sqrt{2}\eta\eta_l}{2m}\cdot\sum_{i=1}^{m}\sum_{k=0}^{K-1}\mathbb{E}\left[ \left\| \frac{\nabla F_i(\theta_{t,k}^i) - \nabla F_i(\theta_t)}{\sqrt[4]{\beta_2 \mathbf{v}_{t-1} + \epsilon}} \right\|^2 \right]$$

$$- \frac{\sqrt{2}\eta\eta_l}{2Km^2}\mathbb{E}\left[ \left\| \frac{1}{\sqrt[4]{\beta_2 \mathbf{v}_{t-1} + \epsilon}}\sum_{i=1}^{m}\sum_{k=0}^{K-1}\nabla F_i(\theta_{t,k}^i) \right\|^2 \right],$$

where the second equation follows from $\langle \mathbf{x}, \mathbf{y} \rangle = \frac{1}{2}[\|\mathbf{x}\|^2 + \|\mathbf{y}\|^2 - \|\mathbf{x} - \mathbf{y}\|^2]$, and the inequality holds by applying Cauchy-Schwarz inequality. Then by Assumption 3.1, then

$$\sqrt{2}\eta \left\langle \frac{\nabla f(\theta_t)}{\sqrt{\beta_2 \mathbf{v}_{t-1} + \epsilon}}, \mathbb{E}\left[ -\frac{\eta_t}{m}\sum_{i=1}^{m}\sum_{k=0}^{K-1}\mathbf{g}_{t,k}^i + \frac{\eta_l K}{m}\sum_{i=1}^{m}\nabla F_i(\theta_t) \right] \right\rangle$$

$$\leq \frac{\sqrt{2}\eta\eta_l K}{2}\left\| \frac{\nabla f(\theta_t)}{\sqrt[4]{\beta_2 \mathbf{v}_{t-1} + \epsilon}} \right\|^2 + \frac{\sqrt{2}\eta\eta_l L^2}{2m}\sum_{i=1}^{m}\sum_{k=0}^{K-1}\mathbb{E}\left[ \left\| \frac{\theta_{t,k}^i - \theta_t}{\sqrt[4]{\beta_2 \mathbf{v}_{t-1} + \epsilon}} \right\|^2 \right]$$

$$- \frac{\sqrt{2}\eta\eta_l}{2Km^2}\mathbb{E}\left[ \left\| \frac{1}{\sqrt[4]{\beta_2 \mathbf{v}_{t-1} + \epsilon}}\sum_{i=1}^{m}\sum_{k=0}^{K-1}\nabla F_i(\theta_{t,k}^i) \right\|^2 \right]$$

$$\leq \frac{3\sqrt{2}\eta\eta_l K}{4}\left\| \frac{\nabla f(\theta_t)}{\sqrt[4]{\beta_2 \mathbf{v}_{t-1} + \epsilon}} \right\|^2 + \frac{5\eta\eta_l^3 K^2 L^2}{\sqrt{2\epsilon}}(\sigma_l^2 + 6K\sigma_g^2)$$

$$- \frac{\sqrt{2}\eta\eta_l}{2Km^2}\mathbb{E}\left[ \left\| \frac{1}{\sqrt[4]{\beta_2 \mathbf{v}_{t-1} + \epsilon}}\sum_{i=1}^{m}\sum_{k=0}^{K-1}\nabla F_i(\theta_{t,k}^i) \right\|^2 \right],$$

where the last inequality holds by applying Lemma F.13 and the constraint of local learning rate $\eta_n \leq \frac{1}{8KL}$. Then

$$T_1 \leq -\frac{\sqrt{2} \cdot \eta\eta_l K}{4}\mathbb{E}\left[\left\|\frac{\nabla f(\theta_t)}{\sqrt[4]{\beta_2 \mathbf{v}_{t-1}+\epsilon}}\right\|^2\right] + \frac{5\eta\eta_l^3 K^2 L^2}{\sqrt{2}\epsilon}\left(\sigma_l^2 + 6K\sigma_g^2\right)$$

$$-\frac{\sqrt{2}\cdot\eta\eta_l}{2Km^2}\mathbb{E}\left[\left\|\frac{1}{\sqrt[4]{\beta_2\mathbf{v}_{t-1}+\epsilon}}\sum_{i=1}^{m}\sum_{k=0}^{K-1}\nabla F_i(\theta_{t,k}^i)\right\|^2\right] + \frac{\eta\sqrt{2(1-\beta_2)}G}{\epsilon}\mathbb{E}[\|\Delta_t\|^2]$$

$$\leq -\frac{\eta\eta_l K}{4}\mathbb{E}\left[\left\|\frac{\nabla f(\theta_t)}{\sqrt[4]{\beta_2\mathbf{v}_{t-1}+\epsilon}}\right\|^2\right] + \frac{5\eta\eta_l^3 K^2 L^2}{\sqrt{2}\epsilon}\left(\sigma_l^2 + 6K\sigma_g^2\right)$$

$$-\frac{\eta\eta_l}{2Km^2}\mathbb{E}\left[\left\|\frac{1}{\sqrt[4]{\beta_2\mathbf{v}_{t-1}+\epsilon}}\sum_{i=1}^{m}\sum_{k=0}^{K-1}\nabla F_i(\theta_{t,k}^i)\right\|^2\right] + \frac{\eta\sqrt{2(1-\beta_2)}G}{\epsilon}\mathbb{E}[\|\Delta_t\|^2]. \quad \text{(E.8)}$$

**Bounding** $T_2$: The bound for $T_2$ mainly follows by the update rule and definition of virtual sequence $\mathbf{z}_t$.

$$T_2 = -\eta\mathbb{E}\left[\left\langle\nabla f(\mathbf{z}_t), \frac{\beta_1}{1-\beta_1}\left(\widehat{\mathbf{V}}_{t-1}^{-1/2}-\widehat{\mathbf{V}}_t^{-1/2}\right)\mathbf{m}_{t-1}' + \left(\widehat{\mathbf{V}}_{t-1}^{-1/2}-\widehat{\mathbf{V}}_t^{-1/2}\right)\Gamma_t\right\rangle\right]$$

$$= \eta\mathbb{E}\left[\left\langle-\nabla f(\theta_t)+\nabla f(\theta_t)-\nabla f(\mathbf{z}_t), \left(\widehat{\mathbf{V}}_{t-1}^{-1/2}-\widehat{\mathbf{V}}_t^{-1/2}\right)\left(\frac{\beta_1}{1-\beta_1}\mathbf{m}_{t-1}'+\Gamma_t\right)\right\rangle\right]$$

$$\leq \eta\mathbb{E}\left[\|\nabla f(\theta_t)\|\left\|\left(\widehat{\mathbf{V}}_{t-1}^{-1/2}-\widehat{\mathbf{V}}_t^{-1/2}\right)\left(\frac{\beta_1}{1-\beta_1}\mathbf{m}_{t-1}'+\Gamma_t\right)\right\|\right]$$

$$+\eta^2 L\mathbb{E}\left[\left\|\widehat{\mathbf{V}}_{t-1}^{-1/2}\left(\frac{\beta_1}{1-\beta_1}\mathbf{m}_{t-1}'+\Gamma_t\right)\right\|\left\|\left(\widehat{\mathbf{V}}_{t-1}^{-1/2}-\widehat{\mathbf{V}}_t^{-1/2}\right)\left(\frac{\beta_1}{1-\beta_1}\mathbf{m}_{t-1}'+\Gamma_t\right)\right\|\right]$$

$$\leq \eta C_1\eta_l KG^2\mathbb{E}\left[\left\|\widehat{\mathbf{V}}_{t-1}^{-1/2}-\widehat{\mathbf{V}}_t^{-1/2}\right\|_1\right] + \eta^2 C_1^2 L\eta_l^2 K^2 G^2\epsilon^{-1/2}\mathbb{E}\left[\left\|\widehat{\mathbf{V}}_{t-1}^{-1/2}-\widehat{\mathbf{V}}_t^{-1/2}\right\|_1\right],$$
$$\text{(E.9)}$$

where the last inequality holds by Lemma C.4, here $C_1 = \frac{\beta_1}{1-\beta_1} + \sqrt{\frac{12q^2}{(1-q^2)^2} + \frac{(1-q^2)^2 C^2}{\alpha^2 m^2 q^2}}$ .

**Bounding**$T_3$ : It can be bounded as follows:

$$T_3 = \frac{\eta^2 L}{2}\mathbb{E}\left[\left\|\widehat{\mathbf{V}}_t^{-1/2}\Delta_t + \frac{\beta_1}{1-\beta_1}\left(\widehat{\mathbf{V}}_{t-1}^{-1/2}-\widehat{\mathbf{V}}_t^{-1/2}\right)\mathbf{m}_{t-1}' + \left(\widehat{\mathbf{V}}_{t-1}^{-1/2}-\widehat{\mathbf{V}}_t^{-1/2}\right)\Gamma_t\right\|^2\right]$$

$$\leq \eta^2 L\mathbb{E}\left[\left\|\widehat{\mathbf{V}}_t^{-1/2}\Delta_t\right\|^2\right] + \eta^2 L\mathbb{E}\left[\left\|\frac{\beta_1}{1-\beta_1}\left(\widehat{\mathbf{V}}_{t-1}^{-1/2}-\widehat{\mathbf{V}}_t^{-1/2}\right)\mathbf{m}_{t-1}' + \left(\widehat{\mathbf{V}}_{t-1}^{-1/2}-\widehat{\mathbf{V}}_t^{-1/2}\right)\Gamma_t\right\|^2\right]$$

$$\leq \eta^2 L\mathbb{E}\left[\left\|\widehat{\mathbf{V}}_t^{-1/2}\Delta_t\right\|^2\right] + \eta^2 L C_1^2\eta_l^2 K^2 G^2\mathbb{E}\left[\left\|\widehat{\mathbf{V}}_{t-1}^{-1/2}-\widehat{\mathbf{V}}_t^{-1/2}\right\|^2\right], \quad \text{(E.10)}$$

where the first inequality follows by Cauchy-Schwarz inequality, and the second one follows by Lemma C.4, here $C_1 = \frac{\beta_1}{1-\beta_1} + \sqrt{\frac{12q^2}{(1-q^2)^2} + \frac{(1-q^2)^2 C^2}{\alpha^2 m^2 q^2}}$. Bounding $T_4$:

$$T_4 = \mathbb{E}\left[\left\langle\nabla f(\mathbf{z}_t)-\nabla f(\theta_t), \eta\widehat{\mathbf{V}}_t^{-1/2}\Delta_t\right\rangle\right]$$

$$\leq \mathbb{E}\left[\|\nabla f(\mathbf{z}_t)-\nabla f(\theta_t)\|\left\|\eta\widehat{\mathbf{V}}_t^{-1/2}\Delta_t\right\|\right]$$

$$\leq L\mathbb{E}\left[\|\mathbf{z}_t-\theta_t\|\left\|\eta\widehat{\mathbf{V}}_t^{-1/2}\Delta_t\right\|\right]$$

$$\leq \frac{\eta^2 L}{2}\mathbb{E}\left[\left\|\widehat{\mathbf{V}}_t^{-1/2}\Delta_t\right\|^2\right] + \frac{\eta^2 L}{2}\mathbb{E}\left[\left\|\frac{\beta_1}{1-\beta_1}\widehat{\mathbf{V}}_{t-1}^{-1/2}\mathbf{m}_{t-1}' + \widehat{\mathbf{V}}_{t-1}^{-1/2}\Gamma_t\right\|^2\right],$$

where the first inequality holds by the fact of $\langle \mathbf{a}, \mathbf{b} \rangle \leq \|\mathbf{a}\|\|\mathbf{b}\|$, the second one follows from Assumption 3.1 and the third one holds by the definition of virtual sequence $\mathbf{z}_t$ and the fact of $\|\mathbf{a}\|\|\mathbf{b}\| \leq \frac{1}{2}\|\mathbf{a}\|^2 + \frac{1}{2}\|\mathbf{b}\|^2$. Then summing $T_4$ over $t = 1, \cdots, T$,

$$
\sum_{t=1}^{T} T_4 \leq \frac{\eta^2 L}{2} \sum_{t=1}^{T} \mathbb{E}\left[\left\|\widehat{\mathbf{V}}_t^{-1/2}\Delta_t\right\|^2\right] + \frac{\eta^2 L}{2\epsilon} \sum_{t=1}^{T} \mathbb{E}\left[\left\|\frac{\beta_1}{1-\beta_1}\mathbf{m}'_{t-1} + \Gamma_t\right\|^2\right]
$$

$$
\leq \frac{\eta^2 L}{2\epsilon} \sum_{t=1}^{T} \mathbb{E}[\|\Delta_t\|^2] + \frac{\eta^2 L}{\epsilon}\left[\frac{\beta_1^2}{(1-\beta_1)^2}\sum_{t=1}^{T}\mathbb{E}\|\mathbf{m}'_{t-1}\|^2 + \sum_{t=1}^{T}\mathbb{E}\|\Gamma_t\|^2\right].
$$

By Lemma F.11, then

$$
\sum_{t=1}^{T}\mathbb{E}[\|\mathbf{m}'_{t-1}\|^2] \leq \frac{TK\eta_l^2}{m}\sigma_l^2 + \frac{\eta_l^2}{m^2}\sum_{t=1}^{T}\mathbb{E}\left[\left\|\sum_{i=1}^{m}\sum_{k=0}^{K-1}\nabla F_i(\theta_{t,k}^i)\right\|^2\right],
$$

and

$$
\sum_{t=1}^{T}\mathbb{E}[\|\Gamma_t\|^2] \leq \frac{4T(q+\gamma)^2}{(1-q^2)^2}\frac{K\eta_l^2}{m}\sigma_l^2 + \frac{\eta_l^2}{m^2}\frac{4(q+\gamma)^2}{(1-q^2)^2}\sum_{t=1}^{T}\mathbb{E}\left[\left\|\sum_{i=1}^{m}\sum_{k=0}^{K-1}\nabla F_i(\theta_{t,k}^i)\right\|^2\right].
$$

Therefore, the $T_4$ term is bounded by

$$
\sum_{t=1}^{T} T_4 \leq \frac{\eta^2 L}{2\epsilon}\sum_{t=1}^{T}\mathbb{E}[\|\Delta_t\|^2] + \frac{C_2\eta^2 L}{\epsilon}\frac{\eta_l^2}{m^2}\sum_{t=1}^{T}\mathbb{E}\left[\left\|\sum_{i=1}^{m}\sum_{k=0}^{K-1}\nabla F_i(\theta_{t,k}^i)\right\|^2\right] + \frac{C_2\eta^2 L}{\epsilon}\frac{TK\eta_l^2}{m}\sigma_l^2,
$$

(E.11)

where $C_2 = \frac{4(q+\gamma+\frac{\lambda C}{\alpha m})^2}{(1-q^2)^2} + \frac{\beta_1^2}{(1-\beta_1)^2}$.

**Bound $T_5$**, there

$$
T_5 = \mathbb{E}\left[\left\langle \nabla f(\mathbf{z}_t), \frac{\eta}{1-\beta_1}\widehat{\mathbf{V}}_t^{-1/2}(\widehat{\bar{\theta}}_t - \theta_t)\right\rangle\right] = \frac{1}{m}\sum_{i}^{m}\mathbb{E}\left[\left\langle \nabla f(\mathbf{z}_t), \frac{\eta}{1-\beta_1}\widehat{\mathbf{V}}_t^{-1/2}(\widehat{\bar{\theta}}_t - \theta_t)\right\rangle\right]
$$

$$
= \frac{1}{m}\sum_{i}^{m}\mathbb{E}\left[\left\langle \nabla f(\theta_t), \frac{\eta}{1-\beta_1}\widehat{\mathbf{V}}_t^{-1/2}(\widehat{\bar{\theta}}_t - \theta_t)\right\rangle\right] + \frac{1}{m}\sum_{i}^{m}\mathbb{E}\left[\left\langle \nabla f(\mathbf{z}_t) - \nabla f(\theta_t), \frac{\eta}{1-\beta_1}\widehat{\mathbf{V}}_t^{-1/2}(\widehat{\bar{\theta}}_t - \theta_t)\right\rangle\right]
$$

$$
\leq \frac{1}{m}\sum_{i}^{m}\mathbb{E}\|\nabla f(\theta_t)\| \cdot \mathbb{E}\left\|\frac{\eta}{1-\beta_1}\widehat{\mathbf{V}}_t^{-1/2}(\widehat{\bar{\theta}}_t - \theta_t)\right\| + \frac{1}{m}\sum_{i}^{m}\mathbb{E}\|\nabla f(\mathbf{z}_t) - \nabla f(\theta_t)\| \cdot \frac{1}{m}\sum_{i}^{m}\mathbb{E}\left\|\frac{\eta}{1-\beta_1}\widehat{\mathbf{V}}_t^{-1/2}(\widehat{\bar{\theta}}_t - \theta_t)\right\|
$$

$$
\leq \frac{1}{m}\sum_{i}^{m}\mathbb{E}\|\nabla f(\theta_t)\| \cdot \frac{1}{m}\sum_{i}^{m}\mathbb{E}\left\|\frac{\eta}{1-\beta_1}\widehat{\mathbf{V}}_t^{-1/2}(\widehat{\bar{\theta}}_t - \theta_t)\right\| + L\frac{1}{m}\sum_{i}^{m}\mathbb{E}\|z_t - \theta_t\| \cdot \frac{1}{m}\sum_{i}^{m}\mathbb{E}\left\|\frac{\eta}{1-\beta_1}\widehat{\mathbf{V}}_t^{-1/2}(\widehat{\bar{\theta}}_t - \theta_t)\right\|
$$

$$
= \frac{1}{m}\sum_{i}^{m}\mathbb{E}\|\nabla f(\theta_t)\| \cdot \frac{1}{m}\sum_{i}^{m}\mathbb{E}\left\|\frac{\eta}{1-\beta_1}\widehat{\mathbf{V}}_t^{-1/2}(\widehat{\bar{\theta}}_t - \theta_t)\right\|
$$

$$
+ L\frac{1}{m}\sum_{i}^{m}\mathbb{E}\left\|\eta\widehat{\mathbf{V}}_t^{-1/2}\Delta_t - \eta\frac{\beta_1}{1-\beta_1}\left(\widehat{\mathbf{V}}_{t-1}^{-1/2} - \widehat{\mathbf{V}}_t^{-1/2}\right)\mathbf{m}'_{t-1} - \eta\left(\widehat{\mathbf{V}}_{t-1}^{-1/2} - \widehat{\mathbf{V}}_t^{-1/2}\right)\Gamma_t + \eta\frac{1}{1-\beta_1}\hat{\mathbf{V}}_t^{-1/2}(\widehat{\bar{\theta}}_t - \theta_t) - \theta_t\right\|
$$

$$
\cdot \frac{1}{m}\sum_{i}^{m}\mathbb{E}\left\|\frac{\eta}{1-\beta_1}\widehat{\mathbf{V}}_t^{-1/2}(\widehat{\bar{\theta}}_t - \theta_t)\right\|
$$

$$
\leq \frac{1}{m}\sum_{i}^{m}\mathbb{E}\|\nabla f(\theta_t)\| \cdot \mathbb{E}\left\|\frac{\eta}{1-\beta_1}\widehat{\mathbf{V}}_t^{-1/2}(\widehat{\bar{\theta}}_t - \theta_t)\right\| + L\frac{1}{m}\sum_{i}^{m}\mathbb{E}\left\|\eta\widehat{\mathbf{V}}_t^{-1/2}\Delta_t\right\| \cdot \frac{1}{m}\sum_{i}^{m}\mathbb{E}\left\|\frac{\eta}{1-\beta_1}\widehat{\mathbf{V}}_t^{-1/2}(\widehat{\bar{\theta}}_t - \theta_t)\right\|
$$

$$+ L\frac{1}{m}\sum_i^m \mathbb{E}\left\|\eta\frac{\beta_1}{1-\beta_1}\left(\widehat{\mathbf{V}}_{t-1}^{-1/2}-\widehat{\mathbf{V}}_t^{-1/2}\right)\mathbf{m}_{t-1}' + \eta\left(\widehat{\mathbf{V}}_{t-1}^{-1/2}-\widehat{\mathbf{V}}_t^{-1/2}\right)\Gamma_t\right\| \cdot \frac{1}{m}\sum_i^m \mathbb{E}\left\|\frac{\eta}{1-\beta_1}\widehat{\mathbf{V}}_t^{-1/2}(\widehat{\widehat{\theta}}_t-\theta_t)\right\|$$

$$+ L\frac{1}{m}\sum_i^m \mathbb{E}\left\|\eta\frac{1}{1-\beta_1}\widehat{\mathbf{V}}_t^{-1/2}(\widehat{\widehat{\theta}}_t-\theta_t)\right\| \cdot \frac{1}{m}\sum_i^m \mathbb{E}\left\|\frac{\eta}{1-\beta_1}\widehat{\mathbf{V}}_t^{-1/2}(\widehat{\widehat{\theta}}_t-\theta_t)\right\|$$

$$+ L\frac{1}{m}\sum_i^m \mathbb{E}\|\theta_t\| \cdot \frac{1}{m}\sum_i^m \mathbb{E}\left\|\frac{\eta}{1-\beta_1}\widehat{\mathbf{V}}_t^{-1/2}(\widehat{\widehat{\theta}}_t-\theta_t)\right\| \tag{E.12}$$

$$\leq \left[G + \frac{L\eta\eta_l KG}{\sqrt{\epsilon}} + L\eta\eta_l C_1 KG\mathbb{E}\|\widehat{\mathbf{V}}_{t-1}^{-1/2}-\widehat{\mathbf{V}}_t^{-1/2}\|\right] \cdot \frac{\eta(\gamma+\frac{C}{\alpha m})H}{(1-\beta)\sqrt{\epsilon}} + \frac{2L\eta^2(\gamma^2+\frac{C^2}{\alpha^2 m^2})H^2}{(1-\beta)^2\epsilon} + \frac{2L\eta^2(\gamma^2+\frac{C^2}{\alpha^2 m^2})H^2}{(1-\beta)\sqrt{\epsilon}} \tag{E.13}$$

here $C_1 = \frac{\beta_1}{1-\beta_1} + \sqrt{\frac{12q^2}{(1-q^2)^2} + \frac{(1-q^2)^2 C^2}{\alpha^2 m^2 q^2}}$ .

**Bound of** $T_6$ , there

$$T_6 = \frac{\eta^2 L^2}{(1-\beta)^2}\mathbb{E}\left[\left\|\widehat{\mathbf{V}}_t^{-1/2}(\widehat{\widehat{\theta}}_t-\theta_t)\right\|^2\right] = \frac{\eta^2 L^2}{(1-\beta)^2}\frac{1}{m}\sum_i^m \mathbb{E}\left[\left\|\widehat{\mathbf{V}}_t^{-1/2}(\widehat{\widehat{\theta}}_t-\theta_t)\right\|^2\right]$$

$$= \frac{\eta^2 L^2}{(1-\beta)^2}\frac{1}{m}\sum_i^m \mathbb{E}\left[\left\|\widehat{\mathbf{V}}_t^{-1/2}(\widehat{\theta}_t-\theta_t) - \widehat{\mathbf{V}}_t^{-1/2}\frac{1}{M_t}\sum_{i\in M_t} Q_t^i\right\|^2\right]$$

$$\leq \frac{2\eta^2 L^2}{(1-\beta)^2}\frac{1}{m}\sum_i^m \mathbb{E}\left[\left\|\widehat{\mathbf{V}}_t^{-1/2}(\widehat{\theta}_t-\theta_t)\right\|^2\right] + \frac{2\eta^2 L^2}{(1-\beta)^2}\frac{1}{m}\sum_i^m \mathbb{E}\left[\left\|\widehat{\mathbf{V}}_t^{-1/2}\frac{1}{M_t}\sum_{i\in M_t} Q_t^i\right\|^2\right]$$

$$\leq \frac{2L^2\eta^2(\gamma^2+\frac{C^2}{\alpha^2 m^2})H^2}{(1-\beta)^2\epsilon}(\frac{C^2}{\alpha^2 m^2}+1) \tag{E.14}$$

Merging pieces together: Substituting (E.8)), (E.9) , (E.10), (E.12),, (E.14) and , (E.11) into (E.5) , summing over from $t=1$ to $T$ , then

$$\mathbb{E}[f(\mathbf{z}_{T+1})] - f(\mathbf{z}_1) = \sum_{t=1}^T [T_1 + T_2 + T_3 + T_4 + T_5 + T_6]$$

$$\leq -\frac{\eta\eta K}{4}\sum_{t=1}^T \mathbb{E}\left[\left\|\frac{\nabla f(\theta_t)}{\sqrt[4]{\beta_2 \mathbf{v}_{t-1}+\epsilon}}\right\|^2\right] + \frac{5\eta\eta_l^3 K^2 L^2 T}{\sqrt{2\epsilon}}(\sigma_l^2 + 6K\sigma_g^2) + \frac{\sqrt{2(1-\beta_2)}\eta G}{\epsilon}\sum_{t=1}^T \mathbb{E}[\|\Delta_t\|^2]$$

$$- \frac{\eta m}{2Km^2}\sum_{t=1}^T \mathbb{E}\left[\left\|\frac{1}{\sqrt[4]{\beta_2\mathbf{v}_{t-1}+\epsilon}}\sum_{i=1}^m\sum_{k=0}^{K-1}\nabla F_i(\theta_{t,k}^i))\right\|^2\right] + C_1\eta\eta_t KG^2\sum_{t=1}^T \mathbb{E}\left[\left\|\widehat{\mathbf{V}}_{t-1}^{-1/2}-\widehat{\mathbf{V}}_t^{-1/2}\right\|^2\right]$$

$$+ \frac{C_1^2\eta^2\eta_l^2 K^2 G^2}{\sqrt{\epsilon}}\sum_{t=1}^T \mathbb{E}\left[\left\|\widehat{\mathbf{V}}_{t-1}^{-1/2}-\widehat{\mathbf{V}}_t^{-1/2}\right\|_1\right] + C_1^2\eta^2\eta_l^2 K^2 LG^2\sum_{t=1}^T \mathbb{E}\left[\left\|\widehat{\mathbf{V}}_{t-1}^{-1/2}-\widehat{\mathbf{V}}_t^{-1/2}\right\|^2\right]$$

$$+ \eta^2 L\sum_{t=1}^T \mathbb{E}\left[\left\|\widehat{\mathbf{V}}_t^{-1/2}\Delta_t\right\|^2\right] + \frac{\eta^2 L}{2}\sum_{t=1}^T \mathbb{E}\left[\left\|\widehat{\mathbf{V}}_t^{-1/2}\Delta_t\right\|^2\right] + \frac{\eta^2 L}{2}\frac{\beta_1^2}{(1-\beta_1)^2}\sum_{t=1}^T \mathbb{E}[\|\mathbf{m}_t'\|^2]$$

$$+ \frac{\eta^2 L}{2}\sum_{t=1}^T \mathbb{E}[\|\Gamma_t\|^2] + \sum_{t=1}^T T_5 + \sum_{t=1}^T T_6. \tag{E.15}$$

Hence by organizing and applying Lemmas, then

$$\mathbb{E}[f(\mathbf{z}_{T+1})] - f(\mathbf{z}_1)$$

$$\leq -\frac{\eta\eta_l K}{4}\sum_{t=1}^{T}\mathbb{E}\left[\left\|\frac{\nabla f(\theta_t)}{\sqrt[4]{\beta_2\mathbf{v}_{t-1}+\epsilon}}\right\|^2\right] + \frac{5\eta\eta_l^3 K^2 L^2 T}{\sqrt{2\epsilon}}(\sigma_l^2 + 6K\sigma_g^2)$$

$$-\frac{\eta\eta_l}{2Km^2}\sum_{t=1}^{T}\mathbb{E}\left[\left\|\frac{1}{\sqrt[4]{\beta_2\mathbf{v}_{t-1}+\epsilon}}\sum_{i=1}^{m}\sum_{k=0}^{K-1}\nabla F_i(\theta_{t,k}^i))\right\|^2\right] + \frac{C_1\eta\eta_l KG^2 d}{\sqrt{\epsilon}} + \frac{2C_1^2\eta^2\eta_l^2 K^2 LG^2 d}{\epsilon}$$

$$+\left(\eta^2 L + \frac{\eta^2 L}{2} + \sqrt{2(1-\beta_2)}\eta G\right)\left[\frac{KT\eta_l^2}{m\epsilon}\sigma_l^2 + \frac{\eta_l^2}{m^2\epsilon}\sum_{t=1}^{T}\mathbb{E}\left[\left\|\sum_{i=1}^{m}\sum_{k=0}^{K-1}\nabla F_i(\theta_{t,k}^i)\right\|^2\right]\right]$$

$$+\frac{\eta^2 L}{\epsilon}\frac{\eta_l^2 C_2}{m^2}\sum_{t=1}^{T}\mathbb{E}\left[\left\|\sum_{i=1}^{m}\sum_{k=0}^{K-1}\nabla F_i(\theta_{t,k}^i)\right\|^2\right] + \frac{\eta^2 L}{\epsilon}\frac{TK\eta_l^2 C_2}{m}\sigma_l^2 + \sum_{t=1}^{T}T_5 + \sum_{t=1}^{T}T_6,$$

by applying Lemma F.7 into all terms containing the second moment estimate of model difference $\Delta_t$ in (E.15), using the fact that $\left(\sqrt{\beta_2\frac{(1+q^2)^3}{(1-q^2)^2}K^2 G^2+\epsilon}\right)^{-1}\|\theta\| \leq \left(\sqrt{\beta_2\frac{(1+q^2)^3}{(1-q^2)}\eta_l^2 K^2 G^2+\epsilon}\right)^{-1}\|\theta\| \leq \|\frac{\theta}{\sqrt{\beta_2\mathbf{v}+\epsilon}}\| \leq \epsilon^{-1/2}\|\theta\|$, and applying Lemma F.3 and F.13, then

$$\mathbb{E}[f(\mathbf{z}_{T+1})] - f(\mathbf{z}_1)$$

$$\leq -\frac{\eta\eta_l K}{4\sqrt{4\beta_2\frac{(1+q^2)^3}{(1-q^2)^2}\eta_l^2 K^2 G^2+\epsilon}}\sum_{t=1}^{T}\mathbb{E}[\|\nabla f(\theta_t)\|^2] + \frac{5\eta\eta_l^3 K^2 L^2 T}{\sqrt{2\epsilon}}(\sigma_l^2 + 6K\sigma_g^2)$$

$$+\frac{C_1\eta\eta_l KG^2 d}{\sqrt{\epsilon}} + \frac{2C_1^2\eta^2\eta_l^2 K^2 LG^2 d}{\epsilon} + \left(\frac{3\eta^2 L}{2} + C_2\eta^2 L + \sqrt{2(1-\beta_2)}\eta G\right)\frac{KT\eta_l^2}{m\epsilon}\sigma_l^2 + \frac{\eta_l\eta TC^2 K^2 G^2}{\alpha^2 m^2\epsilon}$$

$$-\sum_{t=1}^{T}\mathbb{E}\left[\left\|\sum_{i=1}^{m}\sum_{k=0}^{K-1}\nabla F_i(\theta_{t,k}^i)\right\|^2\right]\left[\frac{\eta\eta_l}{2\sqrt{4\beta_2\frac{(1+q^2)^3}{(1-q^2)^2}\eta_l^2 K^2 G^2+\epsilon}Km^2} - \left(\frac{3\eta^2 L}{2} + C_2\eta^2 L + \sqrt{2(1-\beta_2)}\eta G\right)\frac{\eta_l^2}{m^2\epsilon}\right]$$

$$+\left[G + \frac{L\eta\eta_l KG}{\sqrt{\epsilon}} + L\eta\eta_l C_1 KG\mathbb{E}\|\widehat{\mathbf{V}}_{t-1}^{-1/2} - \widehat{\mathbf{V}}_t^{-1/2}\|\right] \cdot \frac{T\eta(\gamma + \frac{C}{\alpha m})H}{(1-\beta)\sqrt{\epsilon}}$$

$$+\frac{3TL\eta^2(\gamma^2 + \frac{C^2}{\alpha^2 m^2})H^2}{(1-\beta)^2\epsilon} + \frac{2TL\eta^2(\gamma^2 + \frac{C^2}{\alpha^2 m^2})H^2}{(1-\beta)\sqrt{\epsilon}}$$

$$\leq -\frac{\eta\eta_l K}{4C_0}\sum_{t=1}^{T}\mathbb{E}[\|\nabla f(\theta_t)\|^2] + \frac{5\eta\eta_l^3 K^2 L^2 T}{\sqrt{2\epsilon}}(\sigma_l^2 + 6K\sigma_g^2)$$

$$+\frac{C_1\eta\eta_l KG^2 d}{\sqrt{\epsilon}} + \frac{2C_1^2\eta^2\eta_l^2 K^2 LG^2 d}{\epsilon} + \left(\frac{3\eta^2 L}{2} + C_2\eta^2 L + \sqrt{2(1-\beta_2)}\eta G\right)\frac{KT\eta_l^2}{m\epsilon}\sigma_l^2$$

$$+\left[G + \frac{L\eta\eta_l KG}{\sqrt{\epsilon}} + \frac{L\eta\eta_l C_1 KGd}{\epsilon}\right] \cdot \frac{\eta T(\gamma + \frac{C}{\alpha m})H}{(1-\beta)\sqrt{\epsilon}} + \frac{2TL\eta^2(\gamma^2 + \frac{C^2}{\alpha^2 m^2})H^2}{(1-\beta)^2\epsilon} + \frac{2TL\eta^2(\gamma^2 + \frac{C^2}{\alpha^2 m^2})H^2}{(1-\beta)\sqrt{\epsilon}}$$

$$+\frac{2TL^2\eta^2(\gamma^2 + \frac{C^2}{\alpha^2 m^2})H^2}{(1-\beta)^2\epsilon}(\frac{C^2}{\alpha^2 m^2} + 1)$$

where the last inequality holds by $\eta_l \leq \frac{\epsilon}{\sqrt{4\beta_2(1+q^2)^3(1-q^2)^{-2}K^2 G^2+\epsilon}\cdot K(3\eta L+2C_2\eta L+2\sqrt{2(1-\beta_2)}G)}$.

Hence

$$\frac{\eta\eta_l K}{4\sqrt{4\beta_2 \frac{(1+q^2)^3}{(1-q^2)^2}\eta_l^2 K^2 G^2 + \epsilon} \cdot T} \sum_{t=1}^{T} \mathbb{E}[\|\nabla f(\theta_t)\|^2]$$

$$\leq \frac{f(\mathbf{z}_0) - \mathbb{E}[f(\mathbf{z}_T)]}{T} + \frac{5\eta\eta_l^3 K^2 L^2}{\sqrt{2\epsilon}}(\sigma_l^2 + 6K\sigma_g^2) + \frac{C_1\eta\eta_l KG^2 d}{T\sqrt{\epsilon}} + \frac{2C_1^2\eta^2\eta_l^2 K^2 LG^2 d}{T\epsilon}$$

$$+ \left[G + \frac{L\eta\eta_l KG}{\sqrt{\epsilon}} + \frac{L\eta\eta_l C_1 KGd}{\epsilon}\right] \cdot \frac{\eta(\gamma + \frac{C}{\alpha m})H}{(1-\beta)\sqrt{\epsilon}} + \frac{L\eta^2(\gamma^2 + \frac{C^2}{\alpha^2 m^2})H^2}{(1-\beta)^2\epsilon} + \frac{2L\eta^2(\gamma^2 + \frac{C^2}{\alpha^2 m^2})H^2}{(1-\beta)\sqrt{\epsilon}}$$

$$+ \left[3\eta^2 L + 2C_2\eta^2 L + 2\sqrt{2(1-\beta_2)}\eta G\right]\frac{K\eta_l^2}{2m\epsilon}\sigma_l^2 + \frac{2L^2\eta^2(\gamma^2 + \frac{C^2}{\alpha^2 m^2})H^2}{(1-\beta)^2\epsilon}(\frac{C^2}{\alpha^2 m^2} + 1),$$

where $C_1 = \frac{\beta_1}{1-\beta_1} + \sqrt{\frac{12q^2}{(1-q^2)^2} + \frac{(1-q^2)^2 C^2}{\alpha^2 m^2 q^2}}$ and $C_2 = \frac{\beta_1^2}{(1-\beta_1)^2} + \frac{4(q+\gamma+\frac{\lambda C}{\alpha m})^2}{(1-q^2)^2}$. then,

$$\min \mathbb{E}[\|\nabla f(\theta_t)\|^2] \leq 4\sqrt{4\beta_2 \frac{(1+q^2)^3}{(1-q^2)^2}\eta_l^2 K^2 G^2 + \epsilon}\left[\frac{f_0 - f_*}{\eta\eta_l KT} + \frac{\Xi}{T} + \Omega\right],$$

where $\Xi = \frac{C_1 G^2 d}{\sqrt{\epsilon}} + \frac{2C_1^2\eta\eta_l KLG^2 d}{\epsilon}, \Omega = \left[G + \frac{L\eta\eta_l KG}{\sqrt{\epsilon}} + \frac{L\eta\eta_l C_1 KGd}{\epsilon}\right] \cdot \frac{\eta(\gamma+\frac{C}{\alpha m})H}{(1-\beta)\sqrt{\epsilon}} + \frac{2L\eta^2(\gamma^2 + \frac{C^2}{\alpha^2 m^2})H^2}{(1-\beta)^2\epsilon} + \frac{2L\eta^2(\gamma^2 + \frac{C^2}{\alpha^2 m^2})H^2}{(1-\beta)\sqrt{\epsilon}}\frac{5\eta^2 KL^2}{\sqrt{2\epsilon}}(\sigma_l^2 + 6K\sigma_g^2) + [(3 + 2C_2)\eta L + 2\sqrt{2(1-\beta_2)}G]\frac{\eta_l}{2mn\epsilon}\sigma_l^2, C_1 = \frac{\beta_1}{1-\beta_1} + \sqrt{\frac{12q^2}{(1-q^2)^2} + \frac{(1-q^2)^2 C^2}{\alpha^2 m^2 q^2}} + \frac{2TL^2\eta^2(\gamma^2 + \frac{C^2}{\alpha^2 m^2})H^2}{(1-\beta)^2\epsilon}(\frac{C^2}{\alpha^2 m^2} + 1)$ and $C_2 = \frac{\beta_1^2}{(1-\beta_1)^2} + \frac{4(q+\gamma+\frac{\lambda C}{\alpha m})^2}{(1-q^2)^2}$.

The proof of Theorem 4.6 is similar to the above proof procedure and the detailed proof will not be given here.

### E.2 PROOF OF COROLLARY 4.3

Let $\eta_l = \Theta(\frac{1}{\sqrt{TK}}), T = \mathcal{O}(Km)$ and $\eta = \Theta(\sqrt{Km})$, the convergence rate under full participation scheme is $\mathcal{O}(\frac{1}{T})$.

### E.3 ANALYSIS ON THE PARTIAL PARTICIPATION SETTING FOR FEDBNLACA

Similar to partial participation scheme in Section 3, we have the following convergence analysis.

**Theorem E.1.** *Under Assumption 3.1-3.4, if the local learning rate $\eta_l$ satisfies the following condition:*
$\eta_l \leq \min\left\{\frac{1}{8KL}, \frac{n(m-1)\epsilon}{48m(n-1)}[K\sqrt{4\beta_2(1+q^2)^3(1-q^2)^{-2}K^2 G^2 + \epsilon}(\eta L + \sqrt{2(1-\beta_2)}G)]^{-1}\right\}$,
*then the iterates of Algorithm 2 under partial participation scheme satisfy*

$$\min \mathbb{E}[\|\nabla f(\theta_t)\|^2] \leq 8\sqrt{4\beta_2 \frac{(1+q^2)^3}{(1-q^2)^2}\eta_l^2 K^2 G^2 + \epsilon}\left[\frac{f_0 - f_*}{\eta\eta_l KT} + \frac{\Xi}{T} + \Omega\right]$$

*, where* $\Xi = \frac{C_1 G^3 d}{\sqrt{\epsilon}} + \frac{2C_1^2\eta\eta_l KLG^2 d}{\epsilon}, \Omega = \left[G + \frac{L\eta\eta_l KG}{\sqrt{\epsilon}} + \frac{L\eta\eta_l C_1 KGd}{\epsilon}\|\right] \cdot \frac{\eta(\gamma+\frac{C}{\alpha n})H}{(1-\beta)\sqrt{\epsilon}} + \frac{4L\eta^2(\gamma^2 + \frac{C^2}{\alpha^2 n^2})H^2}{(1-\beta)^2\epsilon} + \frac{2L\eta^2(\gamma^2 + \frac{C^2}{\alpha^2 n^2})H^2}{(1-\beta)\sqrt{\epsilon}} + \frac{2L^2\eta^2(\gamma^2 + \frac{C^2}{\alpha^2 n^2})H^2}{(1-\beta)^2\epsilon}(\frac{C^2}{\alpha^2 n^2} + 1) + \frac{C_1\eta\eta_l KLG^2}{\epsilon} + \frac{5\eta^2 KL^2}{\sqrt{2\epsilon}}(\sigma_l^2 + 6K\sigma_g^2) + [\eta L + \sqrt{2(1-\beta_2)}G]\frac{\eta_l}{\eta n\epsilon}\sigma_l^2 + [\eta L + \sqrt{2(1-\beta_2)}G]\frac{\eta_l(m-n)}{n(m-1)\epsilon}[15K^2 L^2\eta_l^2(\sigma_l^2 + 6K\sigma_g^2) + 3K\sigma_g^2]$ and $C_1 = \frac{\beta_1}{1-\beta_1} + \frac{m}{n}\sqrt{\frac{12q^2}{(1-q^2)^2} + \frac{(1-q^2)^2 C^2}{\alpha^2 n^2 q^2}}$.

**Theorem E.2.**

$$\min \mathbb{E}[\|\nabla f(\theta_t)\|^2] \leq 8\sqrt{4\beta_2 \frac{(1+q^2)^3}{(1-q^2)^2}\eta_l^2 K^2 G^2 + \epsilon}\left[\frac{f_0 - f_*}{\eta\eta_l KT} + \frac{\Xi}{T} + \Omega\right]$$

, *where* $\Xi = \frac{C_1 G^3 d}{\sqrt{\epsilon}} + \frac{2C_1^2 \eta \eta_l KLG^2 d}{\epsilon}, \Omega = \left[ G + \frac{L\eta\eta_l KG}{\sqrt{\epsilon}} + \frac{L\eta\eta_l C_1 KGd}{\epsilon} \| \right] \cdot \frac{\eta(\gamma+1)H}{(1-\beta)\sqrt{\epsilon}} +$

$\frac{4L\eta^2(\gamma^2+1)H^2}{(1-\beta)^2\epsilon} + \frac{2L\eta^2(\gamma^2+1)H^2}{(1-\beta)\sqrt{\epsilon}} + \frac{2L^2\eta^2(\gamma^2+1)H^2}{(1-\beta)^2\epsilon} (2) + \frac{C_1\eta\eta_l KLG^2}{\epsilon} + \frac{5\eta^2 KL^2}{\sqrt{2\epsilon}} (\sigma_l^2 + 6K\sigma_g^2) + [\eta L +$

$\sqrt{2(1-\beta_2)}G]\frac{\eta_l}{\eta n \epsilon}\sigma_l^2 + [\eta L + \sqrt{2(1-\beta_2)}G]\frac{\eta_l(m-n)}{n(m-1)\epsilon}[15K^2 L^2 \eta_l^2 (\sigma_l^2 + 6K\sigma_g^2) + 3K\sigma_g^2]$ *and*

$C_1 = \frac{\beta_1}{1-\beta_1} + \frac{m}{n}\sqrt{\frac{12q^2}{(1-q^2)^2} + \frac{(1-q^2)^2}{q^2}}.$

*Remark* E.1. When he parameters $C = D, \frac{C}{\alpha n} = 1$, the result of Theorem E.1 becomes the result of Theorem E.2. The upper bound for $\min_{t \in [T]} \mathbb{E}\|\nabla f(\theta_t)\|^2$ of partial participation is similar to full participation case but with a larger variance term $\Omega$. This is due to the fact that random sampling of participating workers introduces an additional variance during sampling.

**Proof of Theorem** E.1: Notations and equations: From the update rule of Algorithm 2, we have $\mathbf{e}_1 = 0, \mathbf{e}_t = \frac{1}{m}\sum_{i=1}^{m} \mathbf{e}_t^i$ and $\mathrm{m}_t = (1-\beta_1)\sum_{i=1}^{t}\beta_1^{t-i}\widehat{\Delta}_t^i$. Denote a global uncompressed difference $\Delta_t = \frac{1}{|\mathcal{S}_t|}\sum_{i\in\mathcal{S}_t}\Delta_t^i$. Denote a virtual momentum sequence: $\mathbf{m}_t' = \beta_1\mathbf{m}_{t-1}' + (1-\beta_1)\Delta_t$, hence we have $\mathrm{m}_t' = (1-\beta_1)\sum_{i=1}^{t}\beta_1^{t-i}\Delta_i$. Define additional two virtual sequences $\Delta_t' = \frac{1}{n}\sum_{i=1}^{m}\Delta_t^i$ and $\widehat{\Delta}_t' = \frac{1}{n}\sum_{i=1}^{m}\widehat{\Delta}_t^i$. Note that when the client $i$ does not take part in the round of participation at step $t$, we have $\Delta_t^i = \widehat{\Delta}_t^i = 0$, therefore, $\Delta_t' = \Delta_t$ and $\widehat{\Delta}_t' = \widehat{\Delta}_t$.

By the aforementioned definition and notation, define a subset $\mathcal{S}_t = \{w_1^t, w_2^t, ..., w_n^t\}$, then

$$\widehat{\Delta}_t - \Delta_t = \frac{1}{|\mathcal{S}_t|}\sum_{i\in\mathcal{S}_t}(\widehat{\Delta}_t^i - \Delta_t^i) = \frac{1}{n}\sum_{i=1}^{m}(\widehat{\Delta}_t^i - \Delta_t^i) = \frac{1}{n}\sum_{i=1}^{m}(\mathbf{e}_t^i - \mathbf{e}_{t+1}^i) = \mathbf{e}_t' - \mathbf{e}_{t+1}',$$

where the compression errors have the same structure, $\mathbf{e}_t' = \frac{1}{n}\sum_{i=1}^{m}\mathbf{e}_t^i$. Similar to the previous analysis, we define the following sequence:

$$\Gamma_{t+1} := (1-\beta_1)\sum_{\tau=1}^{t+1}\beta_1^{t+1-\tau}\mathbf{e}_\tau',$$

and keep using the Lyapunov function $\mathbf{z}_t$ from (E.4). For the expectation of model difference $\Delta_t$,

$$\mathbb{E}_{\mathcal{S}_t}[\Delta_t] = \frac{1}{n}\mathbb{E}_{\mathcal{S}_t}\left[\sum_{i=1}^{n}\Delta_t^{w_i}\right] = \mathbb{E}_{\mathcal{S}_t}[\Delta_t^{w_1}] = \frac{1}{m}\sum_{i=1}^{m}\Delta_t^i = \bar{\Delta}_t.$$

The proof of FedCAMS in partial participation settings has a similar outline combing the proof of partial participation in FedAMS and full participation in FedCAMS. By Assumption 3.1, then

$\mathbb{E}[f(\mathbf{z}_{t+1})] - f(\mathbf{z}_t)$

$\leq \underbrace{\mathbb{E}\left[\left\langle \nabla f(\theta_t), \eta\widehat{\mathbf{V}}_t^{-1/2}\Delta_t \right\rangle\right]}_{T_1'}$

$-\underbrace{\mathbb{E}\left[\left\langle \nabla f(\mathbf{z}_t), \eta\frac{\beta_1}{1-\beta_1}\left(\widehat{\mathbf{V}}_{t-1}^{-1/2} - \widehat{\mathbf{V}}_t^{-1/2}\right)m_{t-1}' + \left(\widehat{\mathbf{V}}_{t-1}^{-1/2} - \widehat{\mathbf{V}}_t^{-1/2}\right)\Gamma_t \right\rangle\right]}_{T_2'}$

$+\underbrace{\frac{\eta^2 L}{2}\mathbb{E}\left[\left\|\widehat{\mathbf{V}}_t^{-1/2}\Delta_t - \frac{\beta_1}{1-\beta_1}\left(\widehat{\mathbf{V}}_{t-1}^{-1/2} - \widehat{\mathbf{V}}_t^{-1/2}\right)m_{t-1}' - \left(\widehat{\mathbf{V}}_{t-1}^{-1/2} - \widehat{\mathbf{V}}_t^{-1/2}\right)\Gamma_t\right\|^2\right]}$

$+\underbrace{\mathbb{E}\left[\left\langle \nabla f(\mathbf{z}_t) - \nabla f(\theta_t), \eta\widehat{\mathbf{V}}_t^{-1/2}\Delta_t \right\rangle\right]}_{T_4'}$

Note that the bound for $T_2'$ is exactly the same as the bound for $T_2$. For the three corresponding terms, $T_1', T_3'$ and $T_4'$ which include the second-order momentum estimate of $\Delta_t$. For $T_1'$, similar to the full participation settings, we have

$$T_1' \le \sqrt{2}\mathbb{E}\left[\left\langle \nabla f(\theta_t), \eta\frac{\Delta_t}{\sqrt{\beta_2\mathbf{v}_{t-1}+\epsilon}}\right\rangle\right] + \sqrt{2}\eta\mathbb{E}\left[\left\langle \nabla f(\theta_t), \frac{\Delta_t}{\sqrt{\mathbf{v}_t+\epsilon}} - \frac{\Delta_t}{\sqrt{\beta_2\mathbf{v}_{t-1}+\epsilon}}\right\rangle\right].$$
(E.16)

The first term in (E.16) does not change in partial participation scheme. The second term is changed due to the variance of $\Delta_t$ changes. For the second term of $T_1'$,then

$$\sqrt{2}\eta\mathbb{E}\left[\left\langle \nabla f(\theta_t), \frac{\Delta_t}{\sqrt{\mathbf{v}_t+\epsilon}} - \frac{\Delta_t}{\sqrt{\beta_2\mathbf{v}_{t-1}+\epsilon}}\right\rangle\right] \le \frac{\sqrt{2(1-\beta_2)}\eta G}{\epsilon}\mathbb{E}[\|\Delta_t\|^2].$$

For $T_3'$,similar to the proof of $T_3$,we get

$$\sum_{t=1}^T T_3' \le \frac{\eta^2 L}{\epsilon}\sum_{t=1}^T \mathbb{E}[\|\Delta_t\|^2] + \eta^2 L C_1^2 \eta_l^2 K^2 G^2 \sum_{t=1}^T \mathbb{E}\left[\left\|\widehat{\mathbf{V}}_{t-1}^{-1/2} - \widehat{\mathbf{V}}_t^{-1/2}\right\|^2\right],$$

where $C_1 = \frac{\beta_1}{1-\beta_1} + \frac{m}{n}\sqrt{\frac{12q^2}{(1-q^2)^2} + \frac{(1-q^2)^2 C^2}{\alpha^2 n^2 q^2}}$ in partial participation,then

$$T_4' = \eta\mathbb{E}\left[\left\langle f(\mathbf{z}_t) - f(\theta_t), \widehat{\mathbf{V}}_t^{-1/2}\Delta_t\right\rangle\right]$$

$$\le \eta\mathbb{E}\left[\|f(\mathbf{z}_t) - f(\theta_t)\|\left\|\widehat{\mathbf{V}}_t^{-1/2}\Delta_t\right\|\right]$$

$$\le \eta^2 L\mathbb{E}\left[\left\|\frac{\beta_1}{1-\beta_1}\widehat{\mathbf{V}}_{t-1}^{-1/2}\mathbf{m}_{t-1}' + \widehat{\mathbf{V}}_{t-1}^{-1/2}\Gamma_t\right\|\left\|\widehat{\mathbf{V}}_t^{-1/2}\Delta_t\right\|\right]$$

$$\le \frac{C_1\eta^2\eta_l^2 K^2 L G^2}{\epsilon}.$$

**Bound** of $T_5'$ , there

$$T_5' = \mathbb{E}\left[\left\langle \nabla f(\mathbf{z}_t), \frac{\eta}{1-\beta_1}\widehat{\mathbf{V}}_t^{-1/2}(\widehat{\bar{\theta}}_t - \theta_t)\right\rangle\right] = \frac{1}{n}\sum_i^n \mathbb{E}\left[\left\langle \nabla f(\mathbf{z}_t), \frac{\eta}{1-\beta_1}\widehat{\mathbf{V}}_t^{-1/2}(\widehat{\bar{\theta}}_t - \theta_t)\right\rangle\right]$$

$$= \frac{1}{n}\sum_i^n \mathbb{E}\left[\left\langle \nabla f(\theta_t), \frac{\eta}{1-\beta_1}\widehat{\mathbf{V}}_t^{-1/2}(\widehat{\bar{\theta}}_t - \theta_t)\right\rangle\right] + \frac{1}{n}\sum_i^n \mathbb{E}\left[\left\langle \nabla f(\mathbf{z}_t) - \nabla f(\theta_t), \frac{\eta}{1-\beta_1}\widehat{\mathbf{V}}_t^{-1/2}(\widehat{\bar{\theta}}_t - \theta_t)\right\rangle\right]$$

$$\le \frac{1}{n}\sum_i^n \mathbb{E}\|\nabla f(\theta_t)\| \cdot \mathbb{E}\left\|\frac{\eta}{1-\beta_1}\widehat{\mathbf{V}}_t^{-1/2}(\widehat{\bar{\theta}}_t - \theta_t)\right\| + \frac{1}{n}\sum_i^n \mathbb{E}\|\nabla f(\mathbf{z}_t) - \nabla f(\theta_t)\| \cdot \frac{1}{n}\sum_i^n \mathbb{E}\left\|\frac{\eta}{1-\beta_1}\widehat{\mathbf{V}}_t^{-1/2}(\widehat{\bar{\theta}}_t - \theta_t)\right\|$$

$$\le \frac{1}{n}\sum_i^n \mathbb{E}\|\nabla f(\theta_t)\| \cdot \frac{1}{n}\sum_i^n \mathbb{E}\left\|\frac{\eta}{1-\beta_1}\widehat{\mathbf{V}}_t^{-1/2}(\widehat{\bar{\theta}}_t - \theta_t)\right\| + L\frac{1}{n}\sum_i^n \mathbb{E}\|z_t - \theta_t\| \cdot \frac{1}{n}\sum_i^n \mathbb{E}\left\|\frac{\eta}{1-\beta_1}\widehat{\mathbf{V}}_t^{-1/2}(\widehat{\bar{\theta}}_t - \theta_t)\right\|$$

$$= \frac{1}{n}\sum_i^n \mathbb{E}\|\nabla f(\theta_t)\| \cdot \frac{1}{n}\sum_i^n \mathbb{E}\left\|\frac{\eta}{1-\beta_1}\widehat{\mathbf{V}}_t^{-1/2}(\widehat{\bar{\theta}}_t - \theta_t)\right\|$$

$$+ L\frac{1}{n}\sum_i^n \mathbb{E}\left\|\eta\widehat{\mathbf{V}}_t^{-1/2}\Delta_t - \eta\frac{\beta_1}{1-\beta_1}\left(\widehat{\mathbf{V}}_{t-1}^{-1/2} - \widehat{\mathbf{V}}_t^{-1/2}\right)\mathbf{m}_{t-1}' - \eta\left(\widehat{\mathbf{V}}_{t-1}^{-1/2} - \widehat{\mathbf{V}}_t^{-1/2}\right)\Gamma_t + \eta\frac{1}{1-\beta_1}\hat{\mathbf{V}}_t^{-1/2}(\widehat{\bar{\theta}}_t - \theta_t) - \theta_t\right\|$$

$$\cdot \frac{1}{n}\sum_i^n \mathbb{E}\left\|\frac{\eta}{1-\beta_1}\widehat{\mathbf{V}}_t^{-1/2}(\widehat{\bar{\theta}}_t - \theta_t)\right\|$$

$$\le \frac{1}{n}\sum_i^n \mathbb{E}\|\nabla f(\theta_t)\| \cdot \mathbb{E}\left\|\frac{\eta}{1-\beta_1}\widehat{\mathbf{V}}_t^{-1/2}(\widehat{\bar{\theta}}_t - \theta_t)\right\| + L\frac{1}{n}\sum_i^n \mathbb{E}\left\|\eta\widehat{\mathbf{V}}_t^{-1/2}\Delta_t\right\| \cdot \frac{1}{n}\sum_i^n \mathbb{E}\left\|\frac{\eta}{1-\beta_1}\widehat{\mathbf{V}}_t^{-1/2}(\widehat{\bar{\theta}}_t - \theta_t)\right\|$$

$$+ L\frac{1}{n}\sum_i^n \mathbb{E}\left\|\eta\frac{\beta_1}{1-\beta_1}\left(\widehat{\mathbf{V}}_{t-1}^{-1/2} - \widehat{\mathbf{V}}_t^{-1/2}\right)\mathbf{m}'_{t-1} + \eta\left(\widehat{\mathbf{V}}_{t-1}^{-1/2} - \widehat{\mathbf{V}}_t^{-1/2}\right)\Gamma_t\right\| \cdot \frac{1}{n}\sum_i^n \mathbb{E}\left\|\frac{\eta}{1-\beta_1}\widehat{\mathbf{V}}_t^{-1/2}(\widehat{\bar{\theta}}_t - \theta_t)\right\|$$

$$+ L\frac{1}{n}\sum_i^n \mathbb{E}\left\|\eta\frac{1}{1-\beta_1}\widehat{\mathbf{V}}_t^{-1/2}(\widehat{\bar{\theta}}_t - \theta_t)\right\| \cdot \frac{1}{n}\sum_i^n \mathbb{E}\left\|\frac{\eta}{1-\beta_1}\widehat{\mathbf{V}}_t^{-1/2}(\widehat{\bar{\theta}}_t - \theta_t)\right\|$$

$$+ L\frac{1}{n}\sum_i^n \mathbb{E}\left\|\theta_t\right\| \cdot \frac{1}{n}\sum_i^n \mathbb{E}\left\|\frac{\eta}{1-\beta_1}\widehat{\mathbf{V}}_t^{-1/2}(\widehat{\bar{\theta}}_t - \theta_t)\right\|$$

$$\leq \left[G + \frac{L\eta\eta_l KG}{\sqrt{\epsilon}} + L\eta\eta_l C_1 KG\mathbb{E}\|\widehat{\mathbf{V}}_{t-1}^{-1/2} - \widehat{\mathbf{V}}_t^{-1/2}\|\right] \cdot \frac{\eta(\gamma + \frac{C}{\alpha m})H}{(1-\beta)\sqrt{\epsilon}} + \frac{2L\eta^2(\gamma^2 + \frac{C^2}{\alpha^2 n^2})H^2}{(1-\beta)^2\epsilon} + \frac{2L\eta^2(\gamma^2 + \frac{C^2}{\alpha^2 n^2})H^2}{(1-\beta)\sqrt{\epsilon}}$$

here $C_1 = \frac{\beta_1}{1-\beta_1} + \frac{m}{n}\sqrt{\frac{12q^2}{(1-q^2)^2} + \frac{(1-q^2)^2 C^2}{\alpha^2 n^2 q^2}}$ .

**Bound of** $T'_6$ , there

$$T'_6 = \frac{\eta^2 L^2}{(1-\beta)^2}\mathbb{E}\left[\left\|\widehat{\mathbf{V}}_t^{-1/2}(\widehat{\bar{\theta}}_t - \theta_t)\right\|^2\right] = \frac{\eta^2 L^2}{(1-\beta)^2}\frac{1}{n}\sum_i^n \mathbb{E}\left[\left\|\widehat{\mathbf{V}}_t^{-1/2}(\widehat{\bar{\theta}}_t - \theta_t)\right\|^2\right]$$

$$= \frac{\eta^2 L^2}{(1-\beta)^2}\frac{1}{n}\sum_i^n \mathbb{E}\left[\left\|\widehat{\mathbf{V}}_t^{-1/2}(\widehat{\theta}_t - \theta_t) - \widehat{\mathbf{V}}_t^{-1/2}\frac{1}{M_t}\sum_{i\in M_t}Q_t^i\right\|^2\right]$$

$$\leq \frac{2\eta^2 L^2}{(1-\beta)^2}\frac{1}{n}\sum_i^n \mathbb{E}\left[\left\|\widehat{\mathbf{V}}_t^{-1/2}(\widehat{\theta}_t - \theta_t)\right\|^2\right] + \frac{2\eta^2 L^2}{(1-\beta)^2}\frac{1}{n}\sum_i^n \mathbb{E}\left[\left\|\widehat{\mathbf{V}}_t^{-1/2}\frac{1}{M_t}\sum_{i\in M_t}Q_t^i\right\|^2\right]$$

$$\leq \frac{2L\eta^2(\gamma^2 + \frac{C^2}{\alpha^2 n^2})H^2}{(1-\beta)^2\epsilon}$$

Hence, the summation from $T'_1$ to $T'_6$ over total iteration T is:

$$\mathbb{E}[f(\mathbf{z}_{T+1})] - f(\mathbf{z}_1) = \sum_{t=1}^T [T'_1 + T'_2 + T'_3 + T'_4 + +T'_5 + T'_6]$$

$$\leq -\frac{\eta\eta_l K}{4}\sum_{t=1}^T \mathbb{E}\left[\left\|\frac{\nabla f(\theta_t)}{\sqrt[4]{\beta_2 \mathbf{v}_{t-1} + \epsilon}}\right\|^2\right] + \frac{5\eta\eta_l^3 K^2 L^2 T}{\sqrt{2\epsilon}}(\sigma_l^2 + 6K\sigma_g^2) + \frac{\sqrt{2(1-\beta_2)}\eta G}{\epsilon}\sum_{t=1}^T \mathbb{E}[\|\Delta_t\|^2]$$

$$- \frac{\eta\eta_l}{2Km^2}\sum_{t=1}^T \mathbb{E}\left[\left\|\frac{1}{\sqrt[4]{\beta_2 \mathbf{v}_{t-1} + \epsilon}}\sum_{i=1}^m\sum_{k=0}^{K-1}\nabla F_i(\theta_t))\right\|^2\right] + C_1\eta\eta_l KG^2 \sum_{t=1}^T \mathbb{E}\left[\left\|\widehat{\mathbf{V}}_{t-1}^{-1/2} - \widehat{\mathbf{V}}_t^{-1/2}\right\|_1\right]$$

$$+ C_1^2\eta^2\eta_l^2 K^2 LG^2 \epsilon^{-1/2}\sum_{t=1}^T \mathbb{E}\left[\left\|\widehat{\mathbf{V}}_{t-1}^{-1/2} - \widehat{\mathbf{V}}_t^{-1/2}\right\|_1\right] + C_1^2\eta^2\eta_l^2 K^2 LG^2 \sum_{t=1}^T \mathbb{E}\left[\left\|\widehat{\mathbf{V}}_{t-1}^{-1/2} - \widehat{\mathbf{V}}_t^{-1/2}\right\|^2\right]$$

$$+ \frac{\eta^2 L}{\epsilon}\sum_{t=1}^T \mathbb{E}[\|\Delta_t\|^2] + \frac{C_1 T\eta^2\eta_l^2 K^2 LG^2}{\epsilon}$$

$$+ \left[ G + \frac{L\eta\eta_l KG}{\sqrt{\epsilon}} + L\eta\eta_l C_1 KG \mathbb{E}\|\widehat{\mathbf{V}}_{t-1}^{-1/2} - \widehat{\mathbf{V}}_t^{-1/2}\| \right] \cdot \frac{\eta(\gamma + \frac{C}{\alpha m})H}{(1-\beta)\sqrt{\epsilon}} + \frac{4L\eta^2(\gamma^2 + \frac{C^2}{\alpha^2 n^2})H^2}{(1-\beta)^2 \epsilon} + \frac{2L\eta^2(\gamma^2 + \frac{C^2}{\alpha^2 n^2})H^2}{(1-\beta)\sqrt{\epsilon}}$$

$$\leq -\frac{\eta\eta_l K}{4\sqrt{4\beta_2 \frac{(1+q^2)^3}{(1-q^2)^2}\eta_l^2 K^2 G^2 + \epsilon}} \sum_{t=1}^{T} \mathbb{E}[\|\nabla f(\theta_t)\|^2] + \frac{5\eta\eta_l^3 K^2 L^2 T}{\sqrt{2\epsilon}}(\sigma_l^2 + 6K\sigma_g^2) + \frac{C_1 \eta\eta_h KG^2 d}{T\sqrt{\epsilon}}$$

$$+ \frac{2C_1^2 \eta^2 \eta_l^2 K^2 LG^2 d}{T\epsilon} - \frac{\eta\eta_l}{2\sqrt{4\beta_2 \frac{(1+q^2)^3}{(1-q^2)^2}\eta_l^2 K^2 G^2 + \epsilon}Km^2} \sum_{t=1}^{T} \mathbb{E}\left[\left\|\sum_{i=1}^{m}\sum_{k=0}^{K-1} \nabla F_i(\theta_t))\right\|^2\right]$$

$$+ \left( \frac{\eta^2 \eta_l^2 LKT}{n\epsilon} + \frac{\sqrt{2(1-\beta_2)}\eta\eta_l^2 KTG}{n\epsilon} \right) \sigma_l^2 + \frac{C_1 T\eta^2 \eta_l^2 K^2 LG^2}{\epsilon}$$

$$+ \left( \frac{\eta^2 \eta_l^2 L}{\epsilon} + \frac{\sqrt{2(1-\beta_2)}\eta\eta_l^2 G}{\epsilon} \right) \frac{m-n}{mn(m-1)} \left[ 15mK^3 L^3 \eta_l^2 (\sigma_l^2 + 6K\sigma_g^2)T \right]$$

$$+ (90mK^4 L^2 \eta_l^2 + 3mK^2) \sum_{t=1}^{T} \mathbb{E}[\|\nabla f(\theta_t)\|^2] + 3mK^2 T\sigma_g^2 \Big]$$

$$+ \left( \eta^2 \eta_l^2 L + \sqrt{2(1-\beta_2)}\eta\eta_l^2 G \right) \frac{n-1}{mn(m-1)} \sum_{t=1}^{T} \mathbb{E}\left[ \left\|\sum_{i=1}^{m}\sum_{k=0}^{K-1} \nabla F_i(\theta_t))\right\|^2 \right]$$

$$+ \left[ G + \frac{L\eta\eta_l KG}{\sqrt{\epsilon}} + \frac{L\eta\eta_l C_1 KGd}{\epsilon}\| \right] \cdot \frac{\eta(\gamma + \frac{C}{\alpha m})H}{(1-\beta)\sqrt{\epsilon}} + \frac{4L\eta^2(\gamma^2 + \frac{C^2}{\alpha^2 n^2})H^2}{(1-\beta)^2 \epsilon} + \frac{2L\eta^2(\gamma^2 + \frac{C^2}{\alpha^2 n^2})H^2}{(1-\beta)\sqrt{\epsilon}}.$$

The proof outline is similar with previous proof. We take the use of Lemma F.3,F.9,F.13 for corresponding terms. By additional constraints of local learning rate $\eta_n$ with the inequality $[\eta^2 L + \sqrt{2(1-\beta_2)}\eta G]\frac{\eta_l^2(n-1)}{mn(m-1)\epsilon} - \frac{\eta\eta_l}{2Km^2}\left[\sqrt{4\beta_2 \frac{(1+q^2)^3}{(1-q^2)^2}\eta_l^2 K^2 G^2 + \epsilon}\right]^{-1} \leq 0$, we obtain the constraint $\eta_l \leq \frac{n(m-1)}{m(n-1)}\frac{\epsilon}{2K\sqrt{4\beta_2(1+q^2)^3(1-q^2)^{-2}K^2 G^2 + \epsilon}[\eta L + \sqrt{2(1-\beta_2)G}]}$, and we further need $\eta_l$ satisfies $\frac{\eta\eta_l K}{4\sqrt{4\beta_2(1+q^2)^3(1-q^2)^{-2}\eta_l^2 K^2 G^2 + \epsilon}} - (\eta^2 L + \sqrt{2(1-\beta_2)}\eta G)\frac{\eta_l^2(m-n)}{mn(m-1)\epsilon}(90mK^4 L^2 \eta_l^2 + 3mK^2) \geq \frac{\eta\eta_l K}{8\sqrt{4\beta_2(1+q^2)^3(1-q^2)^{-2}\eta_l^2 K^2 G^2 + \epsilon}}$. Hence for the convergence rate, we have

$$\frac{\eta\eta_l K}{8\sqrt{4\beta_2 \frac{(1+q^2)^3}{(1-q^2)^2}\eta_l^2 K^2 G^2 + \epsilon} \cdot T} \sum_{i=1}^{T} \mathbb{E}[\|\nabla f(\theta_t)\|^2]$$

$$\leq \frac{f(\mathbf{z}_0) - \mathbb{E}[f(\mathbf{z}_T)]}{T} + \frac{5\eta\eta_l^3 K^2 L^2}{\sqrt{2\epsilon}}(\sigma_l^2 + 6K\sigma_g^2) + \left( \eta L + \sqrt{2(1-\beta_2)}G \right) \frac{\eta\eta_l^2 K}{n\epsilon}\sigma_l^2$$

$$+ \frac{C_1 \eta\eta_l KG^2 d}{T\sqrt{\epsilon}} + \frac{2C_1^2 \eta^2 \eta_l^2 K^2 LG^2 d}{T\epsilon} + \frac{C_1 \eta^2 \eta_l^2 K^2 LG^2}{\epsilon}$$

$$+ \left( \frac{\eta^2 \eta_l^2 L}{\epsilon} + \frac{\sqrt{2(1-\beta_2)}\eta\eta_l^2 G}{\epsilon} \right) \frac{m-n}{mn(m-1)}[15mK^3 L^2 \eta_l^2 (\sigma_l^2 + 6K\sigma_g^2) + 3mK^2 \sigma_g^2]$$

$$+ \left[ G + \frac{L\eta\eta_l KG}{\sqrt{\epsilon}} + \frac{L\eta\eta_l C_1 KGd}{\epsilon} \right] \cdot \frac{T\eta(\gamma + \frac{C}{\alpha m})H}{T(1-\beta)\sqrt{\epsilon}} + \frac{2TL\eta^2(\gamma^2 + \frac{C^2}{\alpha^2 n^2})H^2}{(1-\beta)^2 \epsilon} + \frac{2TL\eta^2(\gamma^2 + \frac{C^2}{T\alpha^2 n^2})H^2}{T(1-\beta)\sqrt{\epsilon}}$$

$$+ \frac{2TL^2 \eta^2(\gamma^2 + \frac{C^2}{\alpha^2 m^2})H^2}{(1-\beta)^2 \epsilon}(\frac{C^2}{\alpha^2 n^2} + 1).$$

Therefore

$$\min \mathbb{E}[\|\nabla f(\theta_t)\|^2] \leq 8\sqrt{4\beta_2 \frac{(1+q^2)^3}{(1-q^2)^2}\eta_l^2 K^2 G^2 + \epsilon}\left[ \frac{f_0 - f_*}{\eta\eta_l KT} + \frac{\Xi}{T} + \Omega \right]$$

, where $\Xi = \frac{C_1 G^3 d}{\sqrt{\epsilon}} + \frac{2C_1^2 \eta \eta_l K L G^2 d}{\epsilon}, \Omega = \left[ G + \frac{L \eta \eta_l K G}{\sqrt{\epsilon}} + \frac{L \eta \eta_l C_1 K G d}{\epsilon} \| \right] \cdot \frac{\eta (\gamma + \frac{C}{\alpha m}) H}{(1-\beta)\sqrt{\epsilon}} + \frac{4L\eta^2(\gamma^2 + \frac{C^2}{\alpha^2 n^2})H^2}{(1-\beta)^2 \epsilon} + \frac{2L\eta^2(\gamma^2 + \frac{C^2}{\alpha^2 n^2})H^2}{(1-\beta)\sqrt{\epsilon}} + \frac{2L^2\eta^2(\gamma^2 + \frac{C^2}{\alpha^2 m^2})H^2}{(1-\beta)^2 \epsilon}(\frac{C^2}{\alpha^2 n^2} + 1) = \frac{C_1 \eta \eta_l K L G^2}{\epsilon} + \frac{5\eta^2 KL^2}{\sqrt{2\epsilon}}(\sigma_l^2 + 6K\sigma_g^2) + [\eta L + \sqrt{2(1-\beta_2)}G]\frac{\eta_l}{\eta n \epsilon}\sigma_l^2 + [\eta L + \sqrt{2(1-\beta_2)}G]\frac{\eta_l(m-n)}{n(m-1)\epsilon}[15K^2 L^2 \eta_l^2(\sigma_l^2 + 6K\sigma_g^2) + 3K\sigma_g^2]$ and $C_1 = \frac{\beta_1}{1-\beta_1} + \frac{m}{n}\sqrt{\frac{12q^2}{(1-q^2)^2} + \frac{(1-q^2)^2 C^2}{\alpha^2 n^2 q^2}}$.

The proof of Theorem E.2 is similar to the above proof procedure and the detailed proof will not be given here.

### E.4 PROOF OF COROLLARY 4.3

If choose $\eta_l = \Theta(\frac{1}{\sqrt{TK}})$ and $\eta = \Theta(\sqrt{Kn})$,we get $\min_{t \in [T]} \mathbb{E}[\|\nabla f(\theta_t)\|^2] = \mathcal{O}(\frac{\sqrt{K}}{\sqrt{Tn}})$.

## F  LEMMAS

**Lemma F.1.** *For the element-wise difference,* $W_t = \frac{1}{\sqrt{\mathbf{v}_t + \epsilon}} - \frac{1}{\sqrt{\beta_2 \mathbf{v}_{t-1} + \epsilon}}, \|W_t\| \leq \frac{\sqrt{1-\beta_2}}{\epsilon}\|\tilde{\Delta}_t\|.$

*Proof.* Note that:

$$
\begin{aligned}
\|W_t\| &= \left\| \frac{1}{\sqrt{\mathbf{v}_t + \epsilon}} - \frac{1}{\sqrt{\beta_2 \mathbf{v}_{t-1} + \epsilon}} \right\| \\
&= \left\| \frac{(\sqrt{\beta_2 \mathbf{v}_{t-1} + \epsilon} - \sqrt{\mathbf{v}_t + \epsilon})(\sqrt{\beta_2 \mathbf{v}_{t-1} + \epsilon} + \sqrt{\mathbf{v}_t + \epsilon})}{\sqrt{\mathbf{v}_t + \epsilon}\sqrt{\beta_2 \mathbf{v}_{t-1} + \epsilon}(\sqrt{\beta_2 \mathbf{v}_{t-1} + \epsilon} + \sqrt{\mathbf{v}_t + \epsilon})} \right\| \\
&= \left\| \frac{\beta_2 \mathbf{v}_{t-1} - \mathbf{v}_t}{\sqrt{\mathbf{v}_t + \epsilon}\sqrt{\beta_2 \mathbf{v}_{t-1} + \epsilon}(\sqrt{\beta_2 \mathbf{v}_{t-1} + \epsilon} + \sqrt{\mathbf{v}_t + \epsilon})} \right\| \\
&= \left\| \frac{-(1-\beta_2)\tilde{\Delta}_t^2}{\sqrt{\mathbf{v}_t + \epsilon}\sqrt{\beta_2 \mathbf{v}_{t-1} + \epsilon}(\sqrt{\beta_2 \mathbf{v}_{t-1} + \epsilon} + \sqrt{\mathbf{v}_t + \epsilon})} \right\| \\
&\leq \left\| \frac{(1-\beta_2)\tilde{\Delta}_t^2}{\sqrt{\mathbf{v}_t + \epsilon}\sqrt{\beta_2 \mathbf{v}_{t-1} + \epsilon}\sqrt{1-\beta_2}\tilde{\Delta}_t} \right\| \\
&\leq \frac{\sqrt{1-\beta_2}}{\epsilon}\|\tilde{\Delta}_t\|,
\end{aligned}
\tag{F.1}
$$

where the forth equation holds by the update rule of $v_t$,i.e, $v_t = \beta_2 \mathbf{v}_{t-1} + (1-\beta_2)\tilde{\Delta}_t^2$,and the last inequality holds due $\sqrt{\mathbf{v}_t + \epsilon} \geq \sqrt{\mathbf{v}_t} \geq \sqrt{1-\beta_2}\tilde{\Delta}_t$ and $\sqrt{\beta_2 \mathbf{v}_{t-1} + \epsilon} \geq 0$.This is the end of the proof. $\square$

**Lemma F.2.** *For the element-wise difference,* $W_t = \frac{1}{\sqrt{\mathbf{v}_t + \epsilon}} - \frac{1}{\sqrt{\beta_2 \mathbf{v}_{t-1} + \epsilon}}, \|W_t\| \leq \frac{\sqrt{1-\beta_2}}{\epsilon}\|\Delta_t\|.$

*Proof.* Note that :

$$\|W_t\| = \left\| \frac{1}{\sqrt{\mathbf{v}_t + \epsilon}} - \frac{1}{\sqrt{\beta_2 \mathbf{v}_{t-1} + \epsilon}} \right\|$$

$$= \left\| \frac{(\sqrt{\beta_2 \mathbf{v}_{t-1} + \epsilon} - \sqrt{\mathbf{v}_t + \epsilon})(\sqrt{\beta_2 \mathbf{v}_{t-1} + \epsilon} + \sqrt{\mathbf{v}_t + \epsilon})}{\sqrt{\mathbf{v}_t + \epsilon}\sqrt{\beta_2 \mathbf{v}_{t-1} + \epsilon}(\sqrt{\beta_2 \mathbf{v}_{t-1} + \epsilon} + \sqrt{\mathbf{v}_t + \epsilon})} \right\|$$

$$= \left\| \frac{\beta_2 \mathbf{v}_{t-1} - \mathbf{v}_t}{\sqrt{\mathbf{v}_t + \epsilon}\sqrt{\beta_2 \mathbf{v}_{t-1} + \epsilon}(\sqrt{\beta_2 \mathbf{v}_{t-1} + \epsilon} + \sqrt{\mathbf{v}_t + \epsilon})} \right\|$$

$$= \left\| \frac{-(1 - \beta_2)\Delta_t^2}{\sqrt{\mathbf{v}_t + \epsilon}\sqrt{\beta_2 \mathbf{v}_{t-1} + \epsilon}(\sqrt{\beta_2 \mathbf{v}_{t-1} + \epsilon} + \sqrt{\mathbf{v}_t + \epsilon})} \right\|$$

$$\leq \left\| \frac{(1 - \beta_2)\Delta_t^2}{\sqrt{\mathbf{v}_t + \epsilon}\sqrt{\beta_2 \mathbf{v}_{t-1} + \epsilon}\sqrt{1 - \beta_2}\Delta_t} \right\|$$

$$\leq \frac{\sqrt{1 - \beta_2}}{\epsilon} \|\Delta_t\|, \tag{F.2}$$

where the forth equation holds by the update rule of $v_t$,i.e, $v_t = \beta_2 \mathbf{v}_{t-1} + (1 - \beta_2)\Delta_t^2$,and the first inequality holds due $\sqrt{\mathbf{v}_t + \epsilon} \geq \sqrt{\mathbf{v}_t} \geq \sqrt{1 - \beta_2}\Delta_t$ and $\sqrt{\beta_2 \mathbf{v}_{t-1} + \epsilon} \geq 0$.This concludes the proof. $\qquad\square$

**Lemma F.3.** *For the variance difference sequence* $\widehat{\mathbf{V}}_{t-1}^{-1/2} - \widehat{\mathbf{V}}_t^{-1/2}$,*then*

$$\sum_{t=1}^{T} \left\| \widehat{\mathbf{V}}_{t-1}^{-1/2} - \widehat{\mathbf{V}}_t^{-1/2} \right\|_1 \leq \frac{d}{\sqrt{\epsilon}}, \sum_{t=1}^{T} \left\| \widehat{\mathbf{V}}_{t-1}^{-1/2} - \widehat{\mathbf{V}}_t^{-1/2} \right\|^2 \leq \frac{d}{\epsilon}. \tag{F.3}$$

*Proof.* By the definition of variance matrix $\widehat{\mathbf{V}}_t$, and the non-decreasing update of FedCAMS, i.e., $\widehat{\mathbf{v}}_{t-1} \leq \widehat{\mathbf{v}}_t = \max(\widehat{\mathbf{v}}_{t-1}, \mathbf{v}_t, \epsilon)$,then

$$\sum_{t=1}^{T} \left\| \hat{\mathbf{V}}_{t-1}^{-1/2} - \hat{\mathbf{V}}_t^{-1/2} \right\|_1 = \sum_{t=1}^{T} \left\| \frac{1}{\sqrt{\widehat{\mathbf{v}}_{t-1}}} - \frac{1}{\sqrt{\widehat{\mathbf{v}}_t}} \right\|_1$$

$$= \sum_{t=1}^{T} \left[ \left\| \frac{1}{\sqrt{\widehat{\mathbf{v}}_{t-1}}} \right\|_1 - \left\| \frac{1}{\sqrt{\widehat{\mathbf{v}}_t}} \right\|_1 \right]$$

$$= \left\| \frac{1}{\sqrt{\widehat{\mathbf{v}}_0}} \right\|_1 - \left\| \frac{1}{\sqrt{\widehat{\mathbf{v}}_T}} \right\|_1$$

$$\leq \frac{d}{\sqrt{\epsilon}}, \tag{F.4}$$

where the inequality holds by the definition of $\widehat{v}_t \in \mathbb{R}^d$: For the sum of the variance difference under $\ell_2$ norm, then

$$\sum_{t=1}^{T} \left\| \hat{\mathbf{V}}_{t-1}^{-1/2} - \hat{\mathbf{V}}_t^{-1/2} \right\|^2 = \sum_{t=1}^{T} \left\| \frac{1}{\sqrt{\widehat{\mathbf{v}}_{t-1}}} - \frac{1}{\sqrt{\widehat{\mathbf{v}}_t}} \right\|^2$$

$$= \sum_{t=1}^{T} \left( \frac{1}{\sqrt{\widehat{\mathbf{v}}_{t-1}}} - \frac{1}{\sqrt{\widehat{\mathbf{v}}_t}} \right)^2$$

$$\leq \sum_{t=1}^{T} \left( \frac{1}{\widehat{\mathbf{v}}_{t-1}} - \frac{1}{\widehat{\mathbf{v}}_t} \right)$$

$$\leq \frac{1}{\widehat{\mathbf{v}}_0} - \frac{1}{\widehat{\mathbf{v}}_T}$$

$$\leq \frac{d}{\epsilon}, \tag{F.5}$$

where the first inequality holds by the element-wise operation: $\forall \mathbf{x}, \mathbf{y} \in \mathbb{R}^d, \mathbf{0} \leq \mathbf{y} \leq \mathbf{x}$, we have $(\mathbf{x} - \mathbf{y})^2 \leq (\mathbf{x} - \mathbf{y})(\mathbf{x} + \mathbf{y})) = \mathbf{x}^2 - \mathbf{y}^2$. It concludes the proof. $\square$

**Lemma F.4.** *The compression error has the following absolute bound*

$$\|\mathbf{e}_t^i\|^2 \leq \frac{4q^2}{(1-q^2)^2}\eta_l^2 K^2 G^2, \quad \|\mathbf{e}_t\|^2 \leq \frac{4q^2}{(1-q^2)^2}\eta_l^2 K^2 G^2. \tag{F.6}$$

*Proof.* For all $t \in [T]$, by Assumption 3.4 and Young's inequality, then if $i \notin M_t$

$$\begin{aligned}
\|\mathbf{e}_{t+1}^i\|^2 &= \|\Delta_t^i + \mathbf{e}_t^i - \mathcal{C}(\Delta_t^i + \mathbf{e}_t^i)\|^2 \\
&\leq q^2 \|\Delta_t^i + \mathbf{e}_t^i\|^2 \\
&\leq q^2(1+\rho)\|\mathbf{e}_t^i\|^2 + q^2\left(1 + \frac{1}{\rho}\right)\|\Delta_t^i\|^2 \\
&\leq \frac{1+q^2}{2}\|\mathbf{e}_t^i\|^2 + \frac{2q^2}{1-q^2}\|\Delta_t^i\|^2,
\end{aligned}$$

if $i \in M_t$

$$\begin{aligned}
\|\mathbf{e}_{t+1}^i\|^2 &= \|\Delta_t^i + \mathbf{e}_t^i - \mathcal{C}(\Delta_{t-1}^i + \mathbf{e}_{t-1}^i)\|^2 \\
&= \|\Delta_t^i + \mathbf{e}_t^i - \mathcal{C}(\Delta_t^i + \mathbf{e}_t^i) + \mathcal{C}(\Delta_t^i + \mathbf{e}_t^i) - \mathcal{C}(\Delta_{t-1}^i + \mathbf{e}_{t-1}^i)\|^2 \\
&\leq 2q^2\|\Delta_t^i + \mathbf{e}_t^i\|^2 + 2\|\mathcal{C}(\Delta_t^i + \mathbf{e}_t^i) - \mathcal{C}(\Delta_{t-1}^i + \mathbf{e}_{t-1}^i)\|^2 \\
&\leq 2q^2(1+\rho)\|\mathbf{e}_t^i\|^2 + \left(2q^2(1 + \frac{1}{\rho}) + \frac{2C^2}{\alpha^2 m^2}\right)\|\Delta_t^i\|^2 \\
&\leq \frac{2(1+q^2)}{2}\|\mathbf{e}_t^i\|^2 + (\frac{4q^2}{1-q^2} + \frac{2C^2}{\alpha^2 m^2})\|\Delta_t^i\|^2,
\end{aligned}$$

where the last inequality holds by choosing $\rho = \frac{1-q^2}{2q^2}$.

Thus obtain the absolute bound for the error terms

$$\|\mathbf{e}_t^i\|^2 \leq \frac{4q^2}{(1-q^2)^2}\eta_l^2 K^2 G^2$$

or

$$\|\mathbf{e}_t^i\|^2 \leq (\frac{8q^2}{(1-q^2)^2} + \frac{2C^2}{\alpha^2 m^2}\frac{1-q^2}{2q^2})\eta_l^2 K^2 G^2,$$

then

$$\begin{aligned}
\|\mathbf{e}_t\|^2 = \left\|\frac{1}{m}\sum_{i=1}^m \mathbf{e}_t^i\right\|^2 &\leq \frac{1}{m}\sum_{i=1}^m \|\Delta_t^i + \mathbf{e}_t^i - \mathcal{C}(\Delta_t^i + \mathbf{e}_t^i)\|^2 + \frac{1}{M_t}\sum_{i \in M_t} \|\Delta_t^i + \mathbf{e}_t^i - \mathcal{C}(\Delta_{t-1}^i + \mathbf{e}_{t-1}^i)\|^2 \\
&\leq \frac{4q^2}{(1-q^2)^2}\eta_l^2 K^2 G^2 + (\frac{8q^2}{(1-q^2)^2} + \frac{2C^2}{\alpha^2 m^2}\frac{1-q^2}{2q^2})\eta_l^2 K^2 G^2. \tag{F.7}
\end{aligned}$$

In the case of partial participation, suppose that client $i$ has the participated time set $T_i$, and we rewrite the $T_i = \{t_0, t_1, ..., t_{p_i}\}$, where $t_0 < t_1 < \cdots < t_{p_i}$. Since when client $i$ are not selected to participate local training, the error stay unchanged. Then for $t_s \in \mathcal{T}_i$

thus by the similar recursive approach, since $\mathbf{e}_{t_0}^i = 0$,

$$\mathbb{E}[\|\mathbf{e}_{t_{s+1}}^i\|^2] \leq \frac{2q^2}{1-q^2}\sum_{\tau=1}^s \left(\frac{1+q^2}{2}\right)^{s-\tau}\mathbb{E}[\|\Delta_{t_\tau}^i\|^2].$$

Thus obtain the absolute bound for the error terms,

$$\|\mathbf{e}_t^i\|^2 \leq \frac{4q^2}{(1-q^2)^2}\eta_l^2 K^2 G^2$$

or

$$\|\mathbf{e}_t^i\|^2 \leq (\frac{8q^2}{(1-q^2)^2} + \frac{2C^2}{\alpha^2 m^2}\frac{1-q^2}{2q^2})\eta_l^2 K^2 G^2,$$

$$\|\mathbf{e}_t\|^2 = \left\|\frac{1}{m}\sum_{i=1}^m \mathbf{e}_t^i\right\|^2 \leq \frac{1}{m}\sum_{i=1}^m \|\Delta_t^i + \mathbf{e}_t^i - \mathcal{C}(\Delta_t^i + \mathbf{e}_t^i)\|^2 + \frac{1}{M_t}\sum_{i\in M_t}\|\Delta_t^i + \mathbf{e}_t^i - \mathcal{C}(\Delta_{t-1}^i + \mathbf{e}_{t-1}^i)\|^2$$

$$\leq \frac{4q^2}{(1-q^2)^2}\eta_l^2 K^2 G^2 + (\frac{8q^2}{(1-q^2)^2} + \frac{2C^2}{\alpha^2 m^2}\frac{1-q^2}{2q^2})\eta_l^2 K^2 G^2. \tag{F.8}$$

This is the end of the proof. $\qquad\square$

**Lemma F.5.** *Under Assumptions 3.2 and 3.4, for FedAMS, we have* $\|\nabla f(\theta)\| \leq G, \|\tilde{\Delta}_t\| \leq \eta_l KG, \|\mathbf{m}_t\| \leq \eta_t KG$ *and* $\|\mathbf{v}_t\| \leq \eta_t^2 K^2 G^2$. *For FedCAMS, we have* $\|\nabla F_i(\theta)\| \leq G, \|\widehat{\Delta}_t\|^2 \leq \frac{4(1+q^2)^3}{(1-q^2)^2}\eta_l^2 K^2 G^2$ $\|\nabla f(\theta)\| \leq G, \|\widehat{\Delta}_t\|^2 \leq \frac{4(1+q^2)^3}{(1-q^2)^2}\eta_l^2 K^2 G^2, \|\mathbf{m}_t'\| \leq \eta_l KG$ *and* $\|\mathbf{v}_t\| \leq \frac{4(1+q^2)^3}{(1-q^2)^2}\eta_l^2 K^2 G^2$, *where* $\mathbf{m}_t' = \beta_1\mathbf{m}_{t-1}' + (1-\beta_1)\tilde{\Delta}_t$.

*Proof.* Since $f$ has $G$-bounded stochastic gradients, for any $\theta$ and $\xi$, we have $\|\nabla f(\theta, \xi)\| \leq G$, that

$$\|\nabla f(\theta)\| = \|\mathbb{E}_\xi \nabla f(\theta, \xi)\| \leq \mathbb{E}_\xi \|\nabla f(\theta, \xi)\| \leq G.$$

For Fed, the model difference $\tilde{\Delta}_t^i$, by definition, has the following formula, therefore,

$$\tilde{\Delta}_t^i = \theta_{t,K}^i - \theta_t = -\eta\sum_{k=1}^K \mathbf{g}_{t,k}^i \text{ or } \tilde{\Delta}_t^i = \theta_{t-1,K}^i - \theta_{t-1} = -\eta\sum_{k=1}^K \mathbf{g}_{t-1,k}^i,$$

$$\|\tilde{\Delta}_t\| \leq \frac{1}{|S_t|}\sum_{i\in S_t}\|\tilde{\Delta}_t^i\| \leq \eta_l KG.$$

Thus the bound for momentum $\mathbf{m}_t$ and variance $\mathbf{v}_t$ has the formula of

$$\|\mathbf{m}_t\| = (1-\beta_1)\sum_{\tau=1}^t \beta_1^{t-\tau}\|\tilde{\Delta}_t\| \leq \eta_l KG,$$

$$\|\mathbf{v}_t\| = (1-\beta_2)\sum_{\tau=1}^t \beta_2^{t-\tau}\|\tilde{\Delta}_t\|^2 \leq \eta_l^2 K^2 G^2.$$

For the compressed version, FedCAMS,

$$\Delta_t^i = \theta_{t,K}^i - \theta_t = -\eta\sum_{k=1}^K \mathbf{g}_{t,k}^i,$$

$$\|\Delta_t\| \leq \frac{1}{m}\sum_{i=1}^m\|\Delta_t^i\| \leq \eta_l KG.$$

Thus the bound for momentum $\mathbf{m}_t$ and variance $\mathbf{v}_t$ has the formula of

$$\|\mathbf{m}_t\| = (1-\beta_1)\sum_{\tau=1}^t \beta_1^{t-\tau}\|\Delta_t\| \leq \eta_l KG,$$

$$\|\mathbf{v}_t\| = (1-\beta_2)\sum_{\tau=1}^t \beta_2^{t-\tau}\|\Delta_t\|^2 \leq \eta_l^2 K^2 G^2.$$

$$\|\widehat{\Delta}_t^i\|^2 \le \|\mathcal{C}(\Delta_t^i + \mathbf{e}_t^i)\|^2$$
$$\le \|\mathcal{C}(\Delta_t^i + \mathbf{e}_t^i) - (\Delta_t^i + \mathbf{e}_t^i) + (\Delta_t^i + \mathbf{e}_t^i)\|^2$$
$$\le 2(q^2 + 1)\|\Delta_t^i + \mathbf{e}_t^i\|^2$$
$$\le 4(q^2 + 1)[\|\Delta_t^i\|^2 + \|\mathbf{e}_t^i\|^2],$$

if $i \notin M_t, \|\widehat{\Delta}_t^i\|^2 \le \frac{4(1+q^2)^3}{(1-q^2)^2}\eta_l^2 K^2 G^2$. if $i \in M_t, \|\widehat{\Delta}_t^i\|^2 \le (\frac{8(1+q^2)^3}{(1-q^2)^2} + \frac{4(1-q^2)(q^2+1)C^2}{\alpha^2 m^2})\eta_l^2 K^2 G^2$.

then

$$\|\widehat{\Delta}_t\|^2 = \left\|\frac{1}{m}\sum_{i=1}^{m}\widehat{\Delta}_t^i\right\|^2 \le \frac{4(1+q^2)^3}{(1-q^2)^2}\eta^2 K^2 G^2 + (\frac{8(1+q^2)^3}{(1-q^2)^2} + \frac{4(1-q^2)(q^2+1)C^2}{\alpha^2 m^2})\eta_l^2 K^2 G^2$$

where the third inequality holds due to Assumption 3.4,and the last inequality holds due to LemmaF.4 e virtual momentum sequence $\|\mathbf{m}_t'\|$ has the same bound as $\mathbf{m}_t$ of FedAMS. For the variance sequence of FedCAMS, we have

$$\|\mathbf{v}_t\| = (1 - \beta_2)\sum_{\tau=1}^{t}\beta_2^{t-\tau}\|\hat{\Delta}_t\|^2 \le \frac{4(1+q^2)^3}{(1-q^2)^2}\eta_l^2 K^2 G^2.$$

This concludes the proof.

□

**Lemma F.6.** *The global model difference* $\Delta_t = \sum_{i=1}^{m}\tilde{\Delta}_t^i$ *in full participation cases satisfy*

$$\mathbb{E}[\|\tilde{\Delta}_t\|^2] \le \frac{K\eta_l^2}{m}\sigma_l^2 + \frac{\eta_l^2}{m^2}\mathbb{E}\left\|\sum_{i=1}^{m}\sum_{k=0}^{K-1}\nabla F_i(\theta_{t,k}^i)\right\|^2 + \frac{1}{m^2}\mathbb{E}\left\|\frac{1}{M_t}\sum_{i\in M_t}q_t^i\right\|^2. \qquad \text{(F.9)}$$

*Proof.* For $\mathbb{E}[\|\tilde{\Delta}_t\|^2]$ in full participation case, then

$$\mathbb{E}[\|\tilde{\Delta}_t\|^2] = \mathbb{E}\left[\left\|\frac{1}{m}\sum_{i=1}^{m}\sum_{k=0}^{K-1}\eta_l\mathbf{g}_{t,k}^i - \frac{1}{M_t}\sum_{i\in M_t}q_t^i\right\|^2\right]$$

$$= \frac{\eta_l^2}{m^2}\mathbb{E}\left[\left\|\sum_{i=1}^{m}\sum_{k=0}^{K-1}\mathbf{g}_{t,k}^i - \frac{1}{\eta_l M_c}\sum_{i\in M_c}q_t^i\right\|^2\right]$$

$$= \frac{\eta_l^2}{m^2}\mathbb{E}\left[\left\|\sum_{i=1}^{m}\sum_{k=0}^{K-1}(\mathbf{g}_{t,k}^i - \nabla F_i(\theta_{t,k}^i))\right\|^2\right] + \frac{\eta_l^2}{m^2}\mathbb{E}\left[\left\|\sum_{i=1}^{m}\sum_{k=0}^{K-1}\nabla F_i(\theta_{t,k}^i) - \frac{1}{\eta_l M_t}\sum_{i\in M_t}q_t^i\right\|^2\right]$$

$$\le \frac{K\eta_l^2}{m}\sigma_l^2 + \frac{2\eta_l^2}{m^2}\left[\mathbb{E}\left\|\sum_{i=1}^{m}\sum_{k=0}^{K-1}\nabla F_i(\theta_{t,k}^i)\right\|^2 + \mathbb{E}\left\|\frac{1}{\eta_l M_t}\sum_{i\in M_t}q_t^i\right\|^2\right]$$

$$= \frac{K\eta_l^2}{m}\sigma_l^2 + \frac{2\eta_l^2}{m^2}\mathbb{E}\left\|\sum_{i=1}^{m}\sum_{k=0}^{K-1}\nabla F_i(\theta_{t,k}^i)\right\|^2 + \frac{2}{m^2}\mathbb{E}\left\|\frac{1}{M_t}\sum_{i\in M_t}q_t^i\right\|^2,$$

where the inequality holds by Assumption 3.2.The end of the proof.

□

**Lemma F.7.** *The global model difference $\Delta_t = \sum_{i=1}^m \Delta_t^i$ in full participation cases satisfy*

$$\mathbb{E}[\|\Delta_t\|^2] \leq \frac{K\eta_l^2}{m}\sigma_l^2 + \frac{\eta_l^2}{m^2}\mathbb{E}\left\|\sum_{i=1}^m \sum_{k=0}^{K-1} \nabla F_i(\theta_{t,k}^i)\right\|^2$$

.

*Proof.* For $\mathbb{E}[\|\Delta_t\|^2]$ in full participation case, then

$$\mathbb{E}[\|\Delta_t\|^2] = \mathbb{E}\left[\left\|\frac{1}{m}\sum_{i=1}^m \sum_{k=0}^{K-1} \eta_l \mathbf{g}_{t,k}^i\right\|^2\right]$$

$$= \frac{\eta_l^2}{m^2}\mathbb{E}\left[\left\|\sum_{i=1}^m \sum_{k=0}^{K-1} \mathbf{g}_{t,k}^i\right\|^2\right]$$

$$= \frac{\eta_l^2}{m^2}\mathbb{E}\left[\left\|\sum_{i=1}^m \sum_{k=0}^{K-1} (\mathbf{g}_{t,k}^i - \nabla F_i(\theta_{t,k}^i))\right\|^2\right] + \frac{\eta_l^2}{m^2}\mathbb{E}\left[\left\|\sum_{i=1}^m \sum_{k=0}^{K-1} \nabla F_i(\theta_{t,k}^i)\right\|^2\right]$$

$$\leq \frac{K\eta_l^2}{m}\sigma_l^2 + \frac{\eta_l^2}{m^2}\left[\mathbb{E}\left\|\sum_{i=1}^m \sum_{k=0}^{K-1} \nabla F_i(\theta_{t,k}^i)\right\|^2\right]$$

$$= \frac{K\eta_l^2}{m}\sigma_l^2 + \frac{\eta_l^2}{m^2}\mathbb{E}\left\|\sum_{i=1}^m \sum_{k=0}^{K-1} \nabla F_i(\theta_{t,k}^i)\right\|^2, \tag{F.10}$$

where the inequality holds by Assumption 3.2.The end of the proof. $\qquad\square$

**Lemma F.8.** *The global model difference $\tilde{\Delta}_t = \sum_{i \in S_t} \tilde{\Delta}_t^i$ in partial participation cases satisfy*

$$\mathbb{E}[\|\tilde{\Delta}_t\|^2] = \frac{K\eta_l^2}{n}\sigma_l^2 + \frac{\eta_l^2(m-n)}{mn(m-1)}[15mK^3L^3\eta_l^2(\sigma_l^2 + 6K\sigma_g^2) + 90mK^4L^2\eta_l^2 + 3mK^2\|\nabla f(\theta_t)\|^2$$

$$+3mK^2\sigma_g^2] + \frac{\eta_l^2(n-1)}{mn(m-1)}\mathbb{E}\left[\left\|\sum_{i=1}^m \sum_{k=0}^{K-1} \nabla F_i(\theta_{t,k}^i) - \frac{1}{M_t}\sum_{i \in M_t} q_t^i\right\|^2\right].$$

*Proof.* Then

$$\mathbb{E}[\|\tilde{\Delta}_t\|^2] = \mathbb{E}\left[\left\|\frac{1}{n}\sum_{i \in \mathcal{S}_t} \tilde{\Delta}_t^i\right\|^2\right]$$

$$= \frac{1}{n^2}\mathbb{E}\left[\left\|\sum_{i=1}^m \mathbb{I}\{i \in \mathcal{S}_t\}\tilde{\Delta}_t^i\right\|^2\right]$$

$$= \frac{1}{n^2}\mathbb{E}\left[\left\|\eta_l^2\sum_{i=1}^m \mathbb{I}\{i \in \mathcal{S}_t\}\sum_{k=0}^{K-1}[\mathbf{g}_{t,k}^i - \nabla F_i(\theta_{t,k}^i)]\right\|^2 + \left\|\eta_l^2\sum_{i=1}^m \mathbb{I}\{i \in \mathcal{S}_t\}\sum_{k=0}^{K-1}\nabla F_i(\theta_{t,k}^i) - \frac{1}{M_t}\sum_{i \in M_t} q_t^i\right\|^2\right]$$

$$= \frac{1}{n^2}\mathbb{E}\left[\left\|\eta_l^2\sum_{i=1}^m \mathbb{P}\{i \in \mathcal{S}_t\}\sum_{k=0}^{K-1}[\mathbf{g}_{t,k}^i - \nabla F_i(\theta_{t,k}^i)]\right\|^2 + \left\|\eta_l^2\sum_{i=1}^m \mathbb{P}\{i \in \mathcal{S}_t\}\sum_{k=0}^{K-1}\nabla F_i(\theta_{t,k}^i) - \frac{1}{M_t}\sum_{i \in M_t} q_t^i\right\|^2\right]$$

$$= \frac{\eta_l^2}{mn}\mathbb{E}\left[\left\|\sum_{i=1}^m \sum_{k=0}^{K-1}[\mathbf{g}_{t,k}^i - \nabla F_i(\theta_{t,k}^i)]\right\|^2\right] + \frac{1}{n^2}\mathbb{E}\left[\left\|\eta_l^2\sum_{i=1}^m \mathbb{P}\{i \in \mathcal{S}_t\}\sum_{k=0}^{K-1}\nabla F_i(\theta_{t,k}^i) - \frac{1}{M_t}\sum_{i \in M_t} q_t^i\right\|^2\right]$$

$$\leq \frac{K\eta_l^2}{n}\sigma_l^2 + \frac{2\eta_l^2}{n^2}\mathbb{E}\left\|\sum_{i=1}^m \mathbb{P}\{i \in \mathcal{S}_t\}\sum_{k=0}^{K-1}\nabla F_i(\theta_{t,k}^i)\right\|^2 + \frac{2}{n^2}\mathbb{E}\left\|\frac{1}{M_t}\sum_{i \in M_t} q_t^i\right\|^2, \quad \text{(F.11)}$$

where the fifth equation holds due to $\mathbb{P}\{i \in \mathcal{S}_t\} = \frac{n}{m}$. Note that

$$
\left\| \sum_{i=1}^{m} \sum_{k=0}^{K-1} \nabla F_i(\theta_{t,k}^i) \right\|^2 = \sum_{i=1}^{m} \left\| \sum_{k=0}^{K-1} \nabla F_i(\theta_{t,k}^i) \right\|^2 + \sum_{i \neq j} \left\langle \sum_{k=0}^{K-1} \nabla F_i(\theta_{t,k}^i), \sum_{k=0}^{K-1} \nabla F_j(\theta_{t,k}^j) \right\rangle
$$

$$
= \sum_{i=1}^{m} m \left\| \sum_{k=0}^{K-1} \nabla F_i(\theta_{t,k}^i) \right\|^2 - \frac{1}{2} \sum_{i \neq j} \left\| \sum_{k=0}^{K-1} \nabla F_i(\theta_{t,k}^i) - \sum_{k=0}^{K-1} \nabla F_j(\theta_{t,k}^j) \right\|^2,
$$

$$\tag{F.12}$$

where the second equation holds due to $\| \sum_{i=1}^{m} \theta_i \|^2 = \sum_{i=1}^{m} m \|\theta_i\|^2 - \frac{1}{2} \sum_{i \neq j} \|\theta_i - \theta_j\|^2$. By the sampling strategy (without replacement), $\mathbb{P}\{i \in \mathcal{S}_t\} = \frac{n}{m}$ and $\mathbb{P}\{i,j \in \mathcal{S}_t\} = \frac{n(n-1)}{m(m-1)}$, thus

$$
\left\| \sum_{i=1}^{m} \sum_{k=0}^{K-1} \mathbb{P}\{i \in \mathcal{S}_t\} \nabla F_i(\theta_{t,k}^i) \right\|^2
$$

$$
= \sum_{i=1}^{m} \mathbb{P}\{i \in \mathcal{S}_t\} \left\| \sum_{k=0}^{K-1} \nabla F_i(\theta_{t,k}^i) \right\|^2 + \sum_{i \neq j} \mathbb{P}\{i,j \in \mathcal{S}_t\} \left\langle \sum_{k=0}^{K-1} \nabla F_i(\theta_{t,k}^i), \sum_{k=0}^{K-1} \nabla F_j(\theta_{t,k}^j) \right\rangle
$$

$$
= \frac{n}{m} \sum_{i=1}^{m} \left\| \sum_{k=0}^{K-1} \nabla F_i(\theta_{t,k}^i) \right\|^2 + \frac{n(n-1)}{m(m-1)} \sum_{i \neq j} \left\langle \sum_{k=0}^{K-1} \nabla F_i(\theta_{t,k}^i), \sum_{k=0}^{K-1} \nabla F_j(\theta_{t,k}^j) \right\rangle
$$

$$
= \frac{n^2}{m} \sum_{i=1}^{m} \left\| \sum_{k=0}^{K-1} \nabla F_i(\theta_{t,k}^i) \right\|^2 - \frac{n(n-1)}{2m(m-1)} \sum_{i \neq j} \left\| \sum_{k=0}^{K-1} \nabla F_i(\theta_{t,k}^i) - \sum_{k=0}^{K-1} \nabla F_j(\theta_{t,k}^j) \right\|^2
$$

$$
= \frac{n(m-n)}{m(m-1)} \sum_{i=1}^{m} \left\| \sum_{k=0}^{K-1} \nabla F_i(\theta_{t,k}^i) \right\|^2 + \frac{n(n-1)}{m(m-1)} \left\| \sum_{i=1}^{m} \sum_{k=0}^{K-1} \nabla F_i(\theta_{t,k}^i) \right\|^2, \tag{F.13}
$$

where the third equation holds due to $\langle \mathbf{x}, \mathbf{y} \rangle = \frac{1}{2}[\|\mathbf{x}^2 + \mathbf{y}^2\| - \|\mathbf{x} - \mathbf{y}\|^2]$ and the last equation holds due to $\frac{1}{2} \sum_{i \neq j} \|\theta_i - \theta_j\|^2 = \sum_{i=1}^{m} m \|\theta_i\|^2 - \|\sum_{i=1}^{m} \theta_i\|^2$. Therefore, for the last term in F.11, then

$$
\mathbb{E}[\|\tilde{\Delta}_t\|^2] = \frac{K\eta_l^2}{n} \sigma_l^2 + \frac{2\eta_l^2(m-n)}{mn(m-1)} \sum_{i=1}^{m} \mathbb{E}\left[ \left\| \sum_{k=0}^{K-1} \nabla F_i(\theta_{t,k}^i) \right\|^2 \right]
$$

$$
+ \frac{2\eta_l^2(n-1)}{mn(m-1)} \mathbb{E}\left[ \left\| \sum_{i=1}^{m} \sum_{k=0}^{K-1} \nabla F_i(\theta_{t,k}^i) \right\|^2 \right] + \frac{2}{n^2} \mathbb{E}\left\| \frac{1}{M_t} \sum_{i \in M_t} q_t^i \right\|^2.
$$

The second term in (F.13) is bounded partially following Reddi et al. (2020),

$$
\sum_{i=1}^{m} \left\| \sum_{k=0}^{K-1} \nabla F_i(\theta_{t,k}^i) \right\|^2 = \sum_{i=1}^{m} \mathbb{E}\left\| \sum_{k=0}^{K-1} \left[ \nabla F_i(\theta_{t,k}^i) - \nabla F_i(\theta_t) + \nabla F_i(\theta_t) - \nabla f(\theta_t) + \nabla f(\theta_t) \right] \right\|^2
$$

$$
\leq 3 \sum_{i=1}^{m} \mathbb{E}\left\| \sum_{k=0}^{K-1} \left[ \nabla F_i(\theta_{t,k}^i) - \nabla F_i(\theta_t) \right] \right\|^2 + 3mK^2\sigma_g^2 + 3mK^2 \|\nabla f(\theta_t)\|^2
$$

$$
\leq 3KL^2 \sum_{i=1}^{m} \sum_{k=0}^{K-1} \mathbb{E}[\|\theta_{t,k}^i - \theta_t\|^2] + 3mK^2\sigma_g^2 + 3mK^2 \|\nabla f(\theta_t)\|^2
$$

$$
\leq 15mK^3L^3\eta_l^2(\sigma_l^2 + 6K\sigma_g^2) + (90mK^4L^2\eta_l^2 + 3mK^2)\|\nabla f(\theta_t)\|^2 + 3mK^2\sigma_g^2
$$

$$\tag{F.14}$$

where the last inequality holds by applying Lemma C.9 (also follows from Reddi et al. (2020)). Substituting (F.14) into (F.13), this concludes the proof.

$\square$

**Lemma F.9.** *The global model difference $\Delta_t = \sum_{i \in S_t} \Delta_t^i$ in partial participation cases satisfy*

$$\mathbb{E}[\|\Delta_t\|^2] = \frac{K\eta_l^2}{n}\sigma_l^2 + \frac{\eta_l^2(m-n)}{mn(m-1)}[15mK^3L^3\eta_l^2(\sigma_l^2 + 6K\sigma_g^2) + 90mK^4L^2\eta_l^2 + 3mK^2\|\nabla f(\theta_t)\|^2$$

$$+3mK^2\sigma_g^2] + \frac{\eta_l^2(n-1)}{mn(m-1)}\mathbb{E}\Big[\Big\|\sum_{i=1}^{m}\sum_{k=0}^{K-1}\nabla F_i(\theta_{t,k}^i)\Big\|^2\Big].$$

*Proof.* we have

$$\mathbb{E}[\|\Delta_t\|^2] = \mathbb{E}\left[\Big\|\frac{1}{n}\sum_{i \in S_t}\Delta_t^i\Big\|^2\right]$$

$$= \frac{1}{n^2}\mathbb{E}\left[\Big\|\sum_{i=1}^{m}\mathbb{I}\{i \in S_t\}\Delta_t^i\Big\|^2\right]$$

$$= \frac{1}{n^2}\mathbb{E}\left[\Big\|\eta_l^2\sum_{i=1}^{m}\mathbb{I}\{i \in S_t\}\sum_{k=0}^{K-1}[\mathbf{g}_{t,k}^i - \nabla F_i(\theta_{t,k}^i)]\Big\|^2 + \Big\|\eta_l^2\sum_{i=1}^{m}\mathbb{I}\{i \in S_t\}\sum_{k=0}^{K-1}\nabla F_i(\theta_{t,k}^i)\Big\|^2\right]$$

$$= \frac{1}{n^2}\mathbb{E}\left[\Big\|\eta_l^2\sum_{i=1}^{m}\mathbb{P}\{i \in S_t\}\sum_{k=0}^{K-1}[\mathbf{g}_{t,k}^i - \nabla F_i(\theta_{t,k}^i)]\Big\|^2 + \Big\|\eta_l^2\sum_{i=1}^{m}\mathbb{P}\{i \in S_t\}\sum_{k=0}^{K-1}\nabla F_i(\theta_{t,k}^i)\Big\|^2\right]$$

$$= \frac{\eta_l^2}{mn}\mathbb{E}\left[\Big\|\sum_{i=1}^{m}\sum_{k=0}^{K-1}[\mathbf{g}_{t,k}^i - \nabla F_i(\theta_{t,k}^i)]\Big\|^2\right] + \frac{1}{n^2}\mathbb{E}\left[\Big\|\eta_l^2\sum_{i=1}^{m}\mathbb{P}\{i \in S_t\}\sum_{k=0}^{K-1}\nabla F_i(\theta_{t,k}^i)\Big\|^2\right]$$

$$\leq \frac{K\eta_l^2}{n}\sigma_l^2 + \frac{2\eta_l^2}{n^2}\mathbb{E}\Big\|\sum_{i=1}^{m}\mathbb{P}\{i \in S_t\}\sum_{k=0}^{K-1}\nabla F_i(\theta_{t,k}^i)\Big\|^2, \tag{F.15}$$

where the fifth equation holds due to $\mathbb{P}\{i \in S_t\} = \frac{n}{m}$. Note that

$$\Big\|\sum_{i=1}^{m}\sum_{k=0}^{K-1}\nabla F_i(\theta_{t,k}^i)\Big\|^2 = \sum_{i=1}^{m}\Big\|\sum_{k=0}^{K-1}\nabla F_i(\theta_{t,k}^i)\Big\|^2 + \sum_{i \neq j}\Big\langle\sum_{k=0}^{K-1}\nabla F_i(\theta_{t,k}^i), \sum_{k=0}^{K-1}\nabla F_j(\theta_{t,k}^j)\Big\rangle$$

$$= \sum_{i=1}^{m}m\Big\|\sum_{k=0}^{K-1}\nabla F_i(\theta_{t,k}^i)\Big\|^2 - \frac{1}{2}\sum_{i \neq j}\Big\|\sum_{k=0}^{K-1}\nabla F_i(\theta_{t,k}^i) - \sum_{k=0}^{K-1}\nabla F_j(\theta_{t,k}^j)\Big\|^2, \tag{F.16}$$

where the second equation holds due to $\|\sum_{i=1}^{m}\theta_i\|^2 = \sum_{i=1}^{m}m\|\theta_i\|^2 - \frac{1}{2}\sum_{i \neq j}\|\theta_i - \theta_j\|^2$. By the sampling strategy (without replacement), we have $\mathbb{P}\{i \in S_t\} = \frac{n}{m}$ and $\mathbb{P}\{i, j \in S_t\} = \frac{n(n-1)}{m(m-1)}$, thus

$$\Big\|\sum_{i=1}^{m}\sum_{k=0}^{K-1}\mathbb{P}\{i \in S_t\}\nabla F_i(\theta_{t,k}^i)\Big\|^2$$

$$= \sum_{i=1}^{m}\mathbb{P}\{i \in S_t\}\Big\|\sum_{k=0}^{K-1}\nabla F_i(\theta_{t,k}^i)\Big\|^2 + \sum_{i \neq j}\mathbb{P}\{i, j \in S_t\}\Big\langle\sum_{k=0}^{K-1}\nabla F_i(\theta_{t,k}^i), \sum_{k=0}^{K-1}\nabla F_j(\theta_{t,k}^j)\Big\rangle$$

$$= \frac{n}{m}\sum_{i=1}^{m}\Big\|\sum_{k=0}^{K-1}\nabla F_i(\theta_{t,k}^i)\Big\|^2 + \frac{n(n-1)}{m(m-1)}\sum_{i \neq j}\Big\langle\sum_{k=0}^{K-1}\nabla F_i(\theta_{t,k}^i), \sum_{k=0}^{K-1}\nabla F_j(\theta_{t,k}^j)\Big\rangle$$

$$= \frac{n^2}{m} \sum_{i=1}^{m} \left\| \sum_{k=0}^{K-1} \nabla F_i(\theta_{t,k}^i) \right\|^2 - \frac{n(n-1)}{2m(m-1)} \sum_{i \neq j} \left\| \sum_{k=0}^{K-1} \nabla F_i(\theta_{t,k}^i) - \sum_{k=0}^{K-1} \nabla F_j(\theta_{t,k}^j) \right\|^2$$

$$= \frac{n(m-n)}{m(m-1)} \sum_{i=1}^{m} \left\| \sum_{k=0}^{K-1} \nabla F_i(\theta_{t,k}^i) \right\|^2 + \frac{n(n-1)}{m(m-1)} \left\| \sum_{i=1}^{m} \sum_{k=0}^{K-1} \nabla F_i(\theta_{t,k}^i) \right\|^2,$$

where the third equation holds due to $\langle \mathbf{x}, \mathbf{y} \rangle = \frac{1}{2}[\|\mathbf{x}^2 + \mathbf{y}^2\| - \|\mathbf{x} - \mathbf{y}\|^2]$ and the last equation holds due to $\frac{1}{2} \sum_{i \neq j} \|\theta_i - \theta_j\|^2 = \sum_{i=1}^{m} m\|\theta_i\|^2 - \|\sum_{i=1}^{m} \theta_i\|^2$. Therefore, for the last term in F.15, then

$$\mathbb{E}[\|\Delta_t\|^2] = \frac{K\eta_l^2}{n}\sigma_l^2 + \frac{\eta_l^2(m-n)}{mn(m-1)} \sum_{i=1}^{m} \mathbb{E}\left[ \left\| \sum_{k=0}^{K-1} \nabla F_i(\theta_{t,k}^i) \right\|^2 \right] + \frac{\eta_l^2(n-1)}{mn(m-1)} \mathbb{E}\left[ \left\| \sum_{i=1}^{m} \sum_{k=0}^{K-1} \nabla F_i(\theta_{t,k}^i) \right\|^2 \right].$$
$$\text{(F.17)}$$

The second term in (F.17) is bounded partially following (Reddi et al., 2020),

$$\sum_{i=1}^{m} \left\| \sum_{k=0}^{K-1} \nabla F_i(\theta_{t,k}^i) \right\|^2 = \sum_{i=1}^{m} \mathbb{E} \left\| \sum_{k=0}^{K-1} \left[ \nabla F_i(\theta_{t,k}^i) - \nabla F_i(\theta_t) + \nabla F_i(\theta_t) - \nabla f(\theta_t) + \nabla f(\theta_t) \right] \right\|^2$$

$$\leq 3 \sum_{i=1}^{m} \mathbb{E} \left\| \sum_{k=0}^{K-1} \left[ \nabla F_i(\theta_{t,k}^i) - \nabla F_i(\theta_t) \right] \right\|^2 + 3mK^2\sigma_g^2 + 3mK^2\|\nabla f(\theta_t)\|^2$$

$$\leq 3KL^2 \sum_{i=1}^{m} \sum_{k=0}^{K-1} \mathbb{E}[\|\theta_{t,k}^i - \theta_t\|^2] + 3mK^2\sigma_g^2 + 3mK^2\|\nabla f(\theta_t)\|^2$$

$$\leq 15mK^3L^3\eta_l^2(\sigma_l^2 + 6K\sigma_g^2) + (90mK^4L^2\eta_l^2 + 3mK^2)\|\nabla f(\theta_t)\|^2 + 3mK^2\sigma_g^2,$$
$$\text{(F.18)}$$

where the last inequality holds by applying Lemma C.9 (also follows from Reddi et al. (2020)). Substituting F.18 into F.17, this concludes the proof. □

**Lemma F.10.** *Under Assumptions 3.1-3.4, for the momentum sequence* $\mathbf{m}_t = (1-\beta_1) \sum_{\tau=1}^{t} \beta_1^{t-\tau} \tilde{\Delta}_\tau$
*and accumulated error sequence* $\Gamma_t = (1-\beta_1) \sum_{\tau=1}^{t} \beta_1^{t-\tau} \mathbf{e}_\tau$ *in full participation settings, then*

$$\sum_{t=1}^{T} \mathbb{E}[\|\mathbf{m}_t\|^2] \leq \frac{TK\eta_l^2}{m}\sigma_l^2 + \frac{2\eta_l^2}{m^2} \sum_{t=1}^{T} \mathbb{E}\left[ \left\| \sum_{i=1}^{m} \sum_{k=0}^{K-1} \nabla F_i(\theta_{t,k}^i) \right\| \right] + \frac{2}{m^2} \sum_{t=1}^{T} \mathbb{E} \left\| \frac{1}{M_t} \sum_{i \in M_t} q_t^i \right\|^2$$

*and*

$$\sum_{t=1}^{T} \mathbb{E}[\|\Gamma_t\|^2] \leq \frac{4Tq^2}{(1-q^2)^2}\frac{K\eta_l^2}{m}\sigma_l^2 + \frac{\eta_l^2}{m^2}\frac{4q^2}{(1-q^2)^2} \sum_{t=1}^{T} \mathbb{E} \left\| \sum_{i=1}^{m} \sum_{k=0}^{K-1} \nabla F_i(\theta_{t,k}^i) \right\|^2 + \frac{1}{m^2}\frac{4q^2}{(1-q^2)^2} \sum_{t=1}^{T} \mathbb{E} \left\| \frac{1}{M_t} \sum_{i \in M_t} q_t^i \right\|^2.$$

*Proof.* By the updating rule, we get

$$\mathbb{E}[\|\mathbf{m}_t\|^2] = \mathbb{E}\left[\|(1-\beta_1)\sum_{\tau=1}^t \beta_1^{t-\tau}\tilde{\Delta}_\tau\|^2\right]$$

$$\leq (1-\beta_1)^2 \mathbb{E}\left[\left(\sum_{\tau=1}^t \beta_1^{t-\tau}\tilde{\Delta}_{\tau,i}\right)^2\right]$$

$$\leq (1-\beta_1)^2 \mathbb{E}\left[\left(\sum_{\tau=1}^t \beta_1^{t-\tau}\right)\left(\sum_{\tau=1}^t \beta_1^{t-\tau}\tilde{\Delta}_{\tau,i}^2\right)\right]$$

$$\leq (1-\beta_1)\sum_{\tau=1}^t \beta_1^{t-\tau}\mathbb{E}[\|\tilde{\Delta}_\tau\|^2]$$

$$\leq \frac{K\eta_l^2}{m}\sigma_l^2 + \frac{\eta_l^2}{m^2}(1-\beta_1)\sum_{\tau=1}^t \beta_1^{t-\tau}\mathbb{E}\left[\left\|\sum_{i=1}^m\sum_{k=0}^{K-1}\nabla F_i(\theta_{t,k}^i) - \frac{1}{M_t}\sum_{i\in M_t}q_t^i\right\|^2\right]$$

$$\frac{2\eta_l^2}{m^2}(1-\beta_1)\sum_{\tau=1}^t \beta_1^{t-\tau}\mathbb{E}\left\|\sum_{i=1}^m\sum_{k=0}^{K-1}\nabla F_i(\theta_{t,k}^i)\right\|^2 + \frac{2}{m^2}(1-\beta_1)\sum_{\tau=1}^t \beta_1^{t-\tau}\mathbb{E}\left\|\frac{1}{M_t}\sum_{i\in M_t}q_t^i\right\|^2,$$

where the second inequality holds by applying Cauchy-Schwarz inequality, and the third inequality holds by summation of series. The last inequality holds by Lemma F.6. Hence summing over $t = 1, \cdots, T$, we get

$$\sum_{t=1}^T \mathbb{E}[\|\mathbf{m}_t\|^2] \leq \frac{TK\eta_l^2}{m}\sigma_l^2 + \frac{2\eta_l^2}{m^2}\sum_{t=1}^T \mathbb{E}\left[\|\sum_{i=1}^m\sum_{k=0}^{K-1}\nabla F_i(\theta_{t,k}^i)\right] + \frac{2}{m^2}\sum_{t=1}^T \mathbb{E}\left\|\frac{1}{M_t}\sum_{i\in M_t}q_t^i\right\|^2$$

$$\square$$

**Lemma F.11.** *Under Assumptions 3.1-3.4,for the momentum sequence* $\mathbf{m}_t = (1-\beta_1)\sum_{r=1}^t \beta_1^{t-\tau}\Delta_\tau$ *and accumulated error sequence* $\Gamma_t = (1-\beta_1)\sum_{\tau=1}^t \beta_1^{t-\tau}\mathbf{e}_\tau$ *in full participation settings, we have*

$$\sum_{t=1}^T \mathbb{E}[\|\mathbf{m}_t\|^2] \leq \frac{TK\eta_l^2}{m}\sigma_l^2 + \frac{\eta_l^2}{m^2}\sum_{t=1}^T \mathbb{E}\left[\|\sum_{i=1}^m\sum_{k=0}^{K-1}\nabla F_i(\theta_{t,k}^i)\right]$$

*and*

$$\sum_{t=1}^T \mathbb{E}[\|\Gamma_t\|^2] \leq \frac{4Tq^2}{(1-q^2)^2}\frac{K\eta_i^2}{m}\sigma_l^2 + \frac{\eta_l^2}{m^2}\frac{4q^2}{(1-q^2)^2}\sum_{t=1}^T \mathbb{E}\left\|\sum_{i=1}^m\sum_{k=0}^{K-1}\nabla F_i(\theta_{t,k}^i)\right\|^2.$$

*Proof.* By the updating rule, we get

$$\mathbb{E}[\|\mathbf{m}_t\|^2] = \mathbb{E}\left[\|(1-\beta_1)\sum_{\tau=1}^t \beta_1^{t-\tau}\Delta_\tau\|^2\right]$$

$$\leq (1-\beta_1)^2\mathbb{E}\left[\left(\sum_{\tau=1}^t \beta_1^{t-\tau}\Delta_{\tau,i}\right)^2\right]$$

$$\leq (1-\beta_1)^2\mathbb{E}\left[\left(\sum_{\tau=1}^t \beta_1^{t-\tau}\right)\left(\sum_{\tau=1}^t \beta_1^{t-\tau}\Delta_{\tau,i}^2\right)\right]$$

$$\leq (1-\beta_1)\sum_{\tau=1}^t \beta_1^{t-\tau}\mathbb{E}[\|\Delta_\tau\|^2]$$

$$\leq \frac{K\eta_l^2}{m}\sigma_l^2 + \frac{\eta_l^2}{m^2}(1-\beta_1)\sum_{\tau=1}^{t}\beta_1^{t-\tau}\mathbb{E}\left[\left\|\sum_{i=1}^{m}\sum_{k=0}^{K-1}\nabla F_i(\theta_{t,k}^i)\right\|^2\right]$$

$$\frac{\eta_l^2}{m^2}(1-\beta_1)\sum_{\tau=1}^{t}\beta_1^{t-\tau}\mathbb{E}\left\|\sum_{i=1}^{m}\sum_{k=0}^{K-1}\nabla F_i(\theta_{t,k}^i)\right\|^2,$$

where the second inequality holds by applying Cauchy-Schwarz inequality, and the third inequality holds by summation of series. The last inequality holds by Lemma F.7. Hence summing over $t = 1, \cdots, T$, we have

$$\sum_{t=1}^{T}\mathbb{E}[\|\mathbf{m}_t\|^2] \leq \frac{TK\eta_l^2}{m}\sigma_l^2 + \frac{\eta_l^2}{m^2}\sum_{t=1}^{T}\mathbb{E}\left[\left\|\sum_{i=1}^{m}\sum_{k=0}^{K-1}\nabla F_i(\theta_{t,k}^i)\right\|\right].$$

For the compress error $e_t$, by Assumption3.4-C.1, then

$$\|\mathbf{e}_{t+1}\| = \left\|\frac{1}{m}\sum_{i=1}^{m}\mathbf{e}_{t+1}^i\right\|$$

$$= \left\|\frac{1}{m}\sum_{i=1}^{m}[\Delta_t^i + \mathbf{e}_t^i] - \frac{1}{m}\sum_{i=1}^{m}\mathcal{C}(\Delta_t^i + \mathbf{e}_t^i) + \frac{1}{M_t}\sum_{i\in M_t}\mathcal{C}(q_t^i)\right\|$$

$$\leq \left\|\frac{1}{m}\sum_{i=1}^{m}[\Delta_t^i + \mathbf{e}_t^i] - \mathcal{C}\left(\frac{1}{m}\sum_{i=1}^{m}[\Delta_t^i + \mathbf{e}_t^i]\right)\right\| + \left\|\mathcal{C}\left(\frac{1}{m}\sum_{i=1}^{m}[\Delta_t^i + \mathbf{e}_t^i]\right) - \frac{1}{m}\sum_{i=1}^{m}\mathcal{C}(\Delta_t^i + \mathbf{e}_t^i)\right\| + \left\|\frac{1}{M_t}\sum_{i\in M_t}\mathcal{C}(q_t^i)\right\|$$

$$\leq q\left\|\frac{1}{m}\sum_{i=1}^{m}[\Delta_t^i + \mathbf{e}_t^i]\right\| + \gamma\left\|\frac{1}{m}\sum_{i=1}^{m}\Delta_t^i\right\| + \frac{C}{\alpha m}\left\|\frac{1}{M_t}\sum_{i\in M_t}\Delta_t^i\right\|$$

$$\leq q\|\Delta_t\| + q\|\mathbf{e}_t\| + \gamma\|\Delta_t\| + \frac{\lambda C}{\alpha m}\|\Delta_t\|$$

$$= q\|\mathbf{e}_t\| + (q + \gamma + \frac{\lambda C}{\alpha m})\|\Delta_t\|,$$

where the first equation holds by the definition for error $\mathbf{e}_{t+1}$, and the second one holds by the update rule for $\mathbf{e}_{t+1}^i$. The first inequality holds by $\|\mathbf{a} + \mathbf{b}\| \leq \|\mathbf{a}\| + \|\mathbf{b}\|$, and the second one holds by Assumption 3.4-C.1. Thus by Young's inequality, then

$$\|\mathbf{e}_{t+1}\|^2 \leq \left(q\|\mathbf{e}_t\| + (q + \gamma + \frac{\lambda C}{\alpha m})\|\Delta_t\|\right)^2$$

$$\leq q^2(1+\rho)\|\mathbf{e}_t\|^2 + (q + \gamma + \frac{\lambda C}{\alpha m})^2(1+\rho^{-1})\|\Delta_t\|^2$$

$$= \frac{1+q^2}{2}\|\mathbf{e}_t\|^2 + \frac{(q + \gamma + \frac{\lambda C}{\alpha m})^2}{1-q^2}\|\Delta_t\|^2, \tag{F.19}$$

where the equation holds by letting $\rho = \frac{1-q^2}{2q^2}$, and $1 + \rho^{-1} = \frac{1+q^2}{1-q^2} \leq \frac{2}{1-q^2}$, then by the similar recursive approach in the proof of Lemma F.3, we have

$$\mathbb{E}[\|\mathbf{e}_{t+1}\|^2] \leq \frac{2(q + \gamma + \frac{\lambda C}{\alpha m})^2}{1-q^2}\sum_{\tau=1}^{t}\left(\frac{1+q^2}{2}\right)^{t-\tau}\mathbb{E}[\|\Delta_\tau\|^2]$$

$$\leq \frac{4(q + \gamma + \frac{\lambda C}{\alpha m})^2}{(1-q^2)^2}\frac{K\eta_l^2}{m}\sigma_l^2 + \frac{\eta_l^2}{m^2}\frac{2(q + \gamma + \frac{\lambda C}{\alpha m})^2}{1-q^2}\sum_{\tau=1}^{t}\left(\frac{1+q^2}{2}\right)^{t-\tau}\mathbb{E}\left[\left\|\sum_{i=1}^{m}\sum_{k=0}^{K-1}\nabla F_i(\mathbf{x}_{\tau,k}^i)\right\|^2\right].$$

For the sequence $\boldsymbol{\Gamma}_t$,similar as the previous analysis, we have

$$
\mathbb{E}[\|\boldsymbol{\Gamma}_\ell\|^2] = \mathbb{E}\left[\left\|(1-\beta_1)\sum_{\tau=1}^{t}\beta_1^{t-\tau}\mathbf{e}_\tau\right\|^2\right]
$$

$$
\leq (1-\beta_1)\sum_{\tau=1}^{t}\beta_1^{t-\tau}\mathbb{E}[\|\mathbf{e}_\tau\|^2]
$$

$$
\leq \frac{4(q+\gamma+\frac{\lambda C}{\alpha m})^2}{(1-q^2)^2}\frac{K\eta_l^2}{m}\sigma_l^2 + \frac{\eta_l^2}{m^2}\frac{2(q+\gamma+\frac{\lambda C}{\alpha m})^2(1-\beta_1)}{1-q^2}\sum_{\tau=1}^{t}\beta_1^{t-\tau}\sum_{j=1}^{\tau}\left(\frac{1+q^2}{2}\right)^{\tau-j}\mathbb{E}\left[\left\|\sum_{i=1}^{m}\sum_{k=0}^{K-1}\nabla F_i(\theta_{j,k}^i)\right\|^2\right]
$$

Summing over $t=1,\cdots,T$,then

$$
\sum_{t=1}^{T}\mathbb{E}[\|\boldsymbol{\Gamma}_t\|^2] \leq \frac{4T(q+\gamma+\frac{\lambda C}{\alpha m})^2}{(1-q^2)^2}\frac{K\eta_i^2}{m}\sigma_i^2 + \frac{\eta_l^2}{m^2}\frac{2(q+\gamma+\frac{\lambda C}{\alpha m})^2}{1-q^2}\sum_{t=1}^{T}\sum_{\tau=1}^{t}\left(\frac{1+q^2}{2}\right)^{t-\tau}\mathbb{E}\left[\left\|\sum_{i=1}^{m}\sum_{k=0}^{K-1}\nabla F_i(\theta_{\tau,k}^i)\right\|^2\right]
$$

$$
+ \frac{\eta_l^2}{m^2}\frac{2(q+\gamma+\frac{\lambda C}{\alpha m})^2(1-\beta_1)}{1-q^2}\sum_{\tau=1}^{t}\beta_1^{t-\tau}\sum_{j=1}^{\tau}\left(\frac{1+q^2}{2}\right)^{\tau-j}\mathbb{E}\left\|\frac{1}{M_t}\sum_{i\in M_t}q_t^i\right\|^2
$$

$$
\leq \frac{4T(q+\gamma+\frac{\lambda C}{\alpha m})^2}{(1-q^2)^2}\frac{K\eta_l^2}{m}\sigma_l^2 + \frac{\eta_l^2}{m^2}\frac{4(q+\gamma+\frac{\lambda C}{\alpha m})^2}{(1-q^2)^2}\sum_{t=1}^{T}\mathbb{E}\left[\left\|\sum_{i=1}^{m}\sum_{k=0}^{K-1}\nabla F_i(\theta_{t,k}^i)\right\|^2\right].
$$

The end of the proof. $\qquad\square$

**Lemma F.12.** *Under Assumptions 3.1-3.4,for the momentum sequence* $\mathbf{m}_t = (1-\beta_1)\sum_{\tau=1}^{t}\beta_1^{t-\tau}\tilde{\Delta}_\tau$ *inpartialparticipationsettings, we have*

$$
\sum_{t=1}^{T}\mathbb{E}[\|\mathbf{m}_t\|^2] \leq \frac{KT\eta_l^2}{n}\sigma_l^2 + \frac{2\eta_l^2}{n^2}\sum_{t=1}^{T}\mathbb{E}\left[\left\|\sum_{i\in\mathcal{S}_t}\sum_{k=0}^{K-1}\nabla F_i(\mathbf{x}_{t,k}^i)\right\|^2\right] + \frac{2}{n^2}\mathbb{E}\left\|\frac{1}{M_t}\sum_{i\in M_t}q_t^i\right\|^2.
$$

*Proof.* The proof outline is the same as the proof of Lemma F.10, the main difference is $E[\|\tilde{\Delta}_t\|^2]$ has changed, so we need to apply Lemma F.8 instead of Lemma F.6 during the proof. $\qquad\square$

**Lemma F.13.** *Under Assumptions 3.1-3.4,for the momentum sequence* $\mathbf{m}_t = (1-\beta_1)\sum_{\tau=1}^{t}\beta_1^{t-\tau}\Delta_\tau$ *in partial participation settings,then*

$$
\sum_{t=1}^{T}\mathbb{E}[\|\mathbf{m}_t\|^2] \leq \frac{KT\eta_l^2}{n}\sigma_l^2 + \frac{\eta_l^2}{n^2}\sum_{t=1}^{T}\mathbb{E}\left[\left\|\sum_{i\in\mathcal{S}_t}\sum_{k=0}^{K-1}\nabla F_i(\mathbf{x}_{t,k}^i)\right\|^2\right].
$$

*Proof.* The proof outline is the same as the proof of Lemma F.11, the main difference is $E[\|\Delta_t\|^2]$ has changed, so we need to apply Lemma F.9 instead of Lemma F.7 during the proof. $\qquad\square$

**Lemma F.14.** *(This lemma directly follows from Lemma 3 in Reddi et al. (2020)). For local learning rate which satisfying* $\eta_l \leq \frac{1}{8KL}$,*the local model difference after* $k$ *($\forall k \in \{0,1,...,K-1\}$) steps local updates satisfies,*

$$
\frac{1}{m}\sum_{i=1}^{m}\mathbb{E}[\|\mathbf{x}_{t,k}^i - \mathbf{x}_t\|^2] \leq 5K\eta_l^2(\sigma_l^2 + 6K\sigma_g^2) + 30K^2\eta_l^2\mathbb{E}[\|\nabla f(\mathbf{x}_t)\|^2].
$$

*Proof.* The proof of Lemma F.14 is exactly same as the proof of Lemma 3 in (Reddi et al. (2020)). $\qquad\square$

