# OpenReview forum: "Bidirectional Communication-Efficient Non-Convex Adaptive Federated Learning"
_ICLR.cc/2025/Conference — ICLR 2025 Conference Withdrawn Submission_

### Official Review · Reviewer_sXXw · 2024-10-24

**Soundness:** 1
**Presentation:** 1
**Contribution:** 1
**Rating:** 1
**Confidence:** 5

**Summary:**

The paper studies methods that apply compression in both directions: uplink and downling (mentioned in the title but not in the pseudocode of Algorithm 1, 2, or 3). The authors provide some convergence guarantees under some set of assumptions for the proposed methods. The methods combine lazy aggregation and adaptive stepsize into FedAvg framework.

**Strengths:**

- I couldn't find any strengths in this work. I believe neither the idea nor the results are new.

**Weaknesses:**

- The paper is really hard to read. Even the two proposed algorithms are written in such a way that didn't allow me to understand what is actually happening.

- Important related works are missing. The idea of lazy updates was also explored in several works [1] but those papers are not even mentioned in the text (and consequently the results are compared). There is almost nothing regarding the related works on bidirectional compression, important papers like [2,3] are not cited (and many more).

- The idea of combining FedAvg with adaptive stepsizes is not new, see for example [4]. It is not clear how the methods from this work improve on top of existing results of adaptive stepsize methods.

- The convergence results are incorrectly reported in the text. The authors mention that they achieve O(1/T) rate but in theorems 4.2 and 4.3 the term proportional to $\Omega$ does not improve with $T$. Therefore, it is not clear how the results in Corollary 4.1 and 4.2 can be true.

- The convergence is provided under strong assumptions (e.g., bounded gradients and deterministic biased compressors). Moreover, the example of a scaled sign compressor doesn't satisfy assumption 3.4 since the parameter $q$ depends on the input $\theta$.

- Some claims in the text look weird to me. For example, lines 36-38 mention that distributed SGD is unsuitable for FL due to the heterogeneity of the data. It is clear that distributed SGD doesn't suffer from heterogeneity but rather from expensive communication. Another example is regarding the discussion around FedAvg. The authors mention that FedAvg has "unaffordable communication" but later mention local computations as a way to reduce the communication cost (which FedAvg does).

- Line 188 mentions "prior to the presentation of my strategy...". In my opinion, this is a leak of information about the authors (that it is a single-author work) which contradicts the rules of the conference.


[1] Peter Richtárik, Igor Sokolov, Ilyas Fatkhullin, Elnur Gasanov, Zhize Li and Eduard Gorbunov
3PC: Three point compressors for communication-efficient distributed training and a better theory for lazy aggregation, (ICML 2022)

[2] Kaja Gruntkowska, Alexander Tyurin, and Peter Richtárik Improving the worst-case bidirectional communication complexity for nonconvex distributed optimization under function similarity, NeurIPS 2024.

[3] Alexander Tyurin and Peter Richtárik, 2Direction: Theoretically faster distributed training with bidirectional communication compression, NeurIPS 2023

[4] Reddi, Sashank and Charles, Zachary and Zaheer, Manzil and Garrett, Zachary and Rush, Keith and Kone{\v{c}}n{\`y}, Jakub and Kumar, Sanjiv and McMahan, H Brendan, Adaptive federated optimization, arXiv preprint arXiv:2003.00295, 2020

**Questions:**

- What is $\epsilon$ in the theorems? It is not mentioned anywhere. What is $M_t$ in the examples? What are the constants $C$ and $D$ in the experiment section, where are they introduced?

---

### Official Review · Reviewer_2mt6 · 2024-10-26

**Soundness:** 4
**Presentation:** 4
**Contribution:** 2
**Rating:** 6
**Confidence:** 3

**Summary:**

In this paper, the authors proposed two communication strategies to reduce the communication cost of Federated Learning. The two strategies, stemming from the LAG strategy, check the difference between the delta of model parameters in current and previous round to determine whether communication can be skipped. Combining the two strategies with FedAMS, the authors proposed FedNLAA and FedAA. And combining the two strategies with FedCAMS, the authors proposed FedBNLACA and FedBACA, which leverages error feedback. The authors then derived convergence results for the proposed algorithms under both full and partial participation settings. Finally, they conducted experiments to show that the proposed algorithms require less communication cost to achieve comparable performance with existing methods.

**Strengths:**

- The proposed algorithms are novel.
- The theoretical results are thorough, which cover a number of settings including compression and partial participation.
- The experimental results are promising. The proposed algorithms require only a fraction of the communication cost of existing methods to achieve the same test accuracy.

**Weaknesses:**

- The major concern I have is the contribution of this work. The proposed algorithms do not have improved convergence rate compared with existing methods, and the proof follow that of FedAMS and FedCAMS. Though the experiment results are good, the setting is too small.
- The authors claimed that LAG requires storing $R$ models. This should be elaborated since from my understanding, LAG only requires storing the model from the previous round and the norm square of the difference between the current and previous model (which is a scalar).
- In Equations (1) and (2), the authors used the notations in LAG paper to present the LAG strategy (e.g., $m$ is the index of the client and is in the subscript). While different notations are used in the main text (e.g., $i$ is the index of the client and is in the superscript). I suggest the authors to use consistent notations, or provide additional explanation for the notations used in the equations.

**Questions:**

- Please refer to Weaknesses
- Typos:
  - Corollary 4.1: "Algorithm 2" -> "Algorithm 1"
  - Corollary 4.3: "Algorithm 1" -> "Algorithm 2", "full participation" -> "partial participation"

---

### Official Review · Reviewer_hcjC · 2024-10-27

**Soundness:** 2
**Presentation:** 1
**Contribution:** 2
**Rating:** 3
**Confidence:** 4

**Summary:**

The paper proposes two strategies for Federated Learning. The NLA strategy achieves communication cost reduction by reducing the amount of information passed and the AA strategy reduces the communication cost by accelerating computation. The author designed FedNLAA and FedAA algorithms. After that authors extend them to bidirectional algorithms (FedBNLACA and FedBACA).

**Strengths:**

- Touching bidirectional compression in Federated Learning is still a valid line of research. However, that this question has not been studied (ies 066-068) is overstatement. Ideas behind the proposed algorithm are pretty novel.
- The convergence rate in terms of T is not improvable in the worst-case sense for finite sum structure (if individual functions are non-convex but smooth).
- The work provides a good overview of some existing works in the realm of distributed training.
- The work uses standards assumptions for convergence analysis

**Weaknesses:**

**Issues with Readability:**

----

1. Incorrect use of "Etal": The correct form is "et al." with a space and a period, as it stands for "and others."

2. Citation error: The paper "T. Li, A. K. Sahu, M. Zaheer, ,andetal. Feddane" is missing content between two commas (Line 603)

3. Redundant "and" with "et al.": The use of "and" alongside "et al." is redundant since "et al." already means "and others."

4. Move notation to the appendix: While the notation in lines 131-135 is standard, I suggest moving it to a glossary in the appendix to save space. In addition, it's worthwhile to add more information about all employed quantities in the rates.

5. Unreadable pseudocode: The algorithms in lines 204 and 225 need to be properly typeset using LaTeX packages like algpseudocode, algorithmic, or algcompatible.

6. Unreadable $\eta_l$ in Theorem 4.3: In line 335, the notation $\eta_l$ isn’t clearly readable. Please reformat.

7. Inappropriate personal language: The phrase "Prior to the presentation of my strategy" in line 188 is too informal for ICLR. Also, "my" is inappropriate when multiple authors are involved (as "Anonymous authors" indicate more than one author).

8. Weak statement in Assumption 3.1: The statement "The smoothness of $F_i$ also means the L-gradient Lipschitz condition" is correct but weak. It’s an equivalence. Please either add a reference, provide proof, or state that it can be proved.

9. Assumption 3.4 is a definition: Assumption 3.4 should be labeled as a definition. Also, line 167 contains a redundant colon.

----

**Issues with Assumptions. All of them are extremely weak in the realm of not parameter-free optimization:**

----
10. Assumption 3.1 is weak in the following sense: It doesn’t account for scenarios where clients have different $L_i$.

11. Typo in Assumption 3.2: I believe In line 162, "\nabla f_i(.)" should be "\nabla f(.)".

12. First appearance of Assumption 3.2. Similar assumptions were proposed in 2013 by Saeed Ghadimi and Guanghui Lan in Stochastic First and Zeroth-Order Methods for Nonconvex Stochastic Programming (SIAM Journal on Optimization, 23(4):2341–2368). But it has not been cited. Assumption 3.2 is weak in general. This assumption does not hold for even basic cases, such as minimizing over real numbers ($f(\theta) = \theta^2$). Provide an example if you believe otherwise.

13. \textbf{Assumption 3.3 is impractical}: The assumption about gradient estimation is impractical and does not hold even for data-points subsampling. The contribution of individual data points to the loss is not an additive random variable with bounded variance. It depends on the position of the iterate, curvature, and more.

To the best of my knowledge, the ABC is the most relaxed which exists
"Better Theory for SGD in the Nonconvex World" (https://arxiv.org/abs/2002.03329)

All previous assumptions before this work are weaker and they include:
a) Bounded variance by
  Saeed Ghadimi and Guanghui Lan.
  Stochastic First- and Zeroth-Order Methods for Nonconvex Stochastic Programming.
  SIAM Journal on Optimization, 23(4):2341–2368, 2013

 b) Maximal strong growth by
   Mark Schmidt and Nicolas Le Roux. Fast Convergence of Stochastic Gradient Descent under a Strong Growth Condition.
   arXiv preprint arXiv:1308.6370, 2013.

c) Expected strong growth by
  Sharan Vaswani, Francis Bach, and Mark Schmidt.
  Fast and Faster Convergence of SGD for Over-Parameterized Models and an Accelerated Perceptron.

d) Relaxed growth condition by
  Leon. Bottou, Frank E. Curtis, and Jorge. Nocedal.
  Optimization Methods for Large-Scale Machine Learning.
  SIAM Review, 60(2):223–311, 2018.

e) Gradient confusion by
  Karthik A. Sankararaman, Soham De, Zheng Xu, W. Ronny Huang, and Tom Goldstein. The Impact of Neural Network Overparameterization on
  Gradient Confusion and Stochastic Gradient Descent

f) Sure smoothness by
  Yunwei Lei, Ting Hu, Guiying Li, and Ke Tang.
  Stochastic Gradient Descent for Nonconvex Learning Without Bounded Gradient
  Assumptions

Assumption 3.3 is extremely weak. If you disagree, provide an example demonstrating its practicality.

----

**Major Issues:**

----

14. Limited novelty of motivation: Bidirectional compression has been studied extensively, and your motivation in lines 066-069 is misleading. Please consider works like EF21-P and Friends (https://arxiv.org/abs/2209.15218), EF21 with Bells & Whistles (https://arxiv.org/abs/2110.03294), and Personalized Federated Learning with Communication Compression (https://openreview.net/pdf?id=dZugyhbNFY). The novelty of your research question may be limited.

15. Convergence issue with Algorithms 1 & 2 compared to current SOTA: Algorithms 1 and 2 only converge to a neighborhood, not the exact solution. For a rate of 1/T without a neighborhood for \epsilon^2 stationary point, you can refer to Algorithm 5 in "EF21 with Bells & Whistles" (Theorem 9, p.51, https://arxiv.org/pdf/2110.03294).

16. Unclear function bounding in Theorems 4.3, 4.4, and 4.5: The assumption that the function is bounded below isn’t explicitly stated. It's unclear what exactly f^* in the realm of non-convex settings refers to - presumably, the value where the non-convex function is bounded below by this value.

17. The source code for experiments to verify reproducibility has not been provided.

----

**Overall Evaluation:**

----
18. Weak comparison to EF21: Your results are weaker than existing work like EF21 with Bells & Whistles (NeurIPS 2021, oral). Specifically:
- Your rates do not converge exactly to the solution without extra assumptions which are hidden in Lines 453-456.
- The theorems are ineffective when G is large
- The paper does not discuss Ω. But if epsilon is 10*{-12}, the Ω due to the the third term will already be in order of 10^{-6}
- EF21-BC requires only one assumption (the Li smoothness of the clients' loss functions), which is even weaker than Assumption 3.1 in your paper.

19. Missing SOTA comparison: You have not compared your results to state-of-the-art biased compressors -- EF21-BC (EF21 with bidirectional biased compression, https://arxiv.org/pdf/2110.03294)

20. Unfortunately, in its current form, I cannot recommend this work for ICLR 2025, because, in terms of assumptions, convergence of the EF21-BC  is strictly better.

**Questions:**

**Recommendation:**
---

I suggest revisiting EF21 with Bells & Whistles (https://arxiv.org/abs/2110.03294), identifying its weaknesses, and focusing on scenarios where your approach can outperform it. Additionally, a comprehensive revision of the minor issues will greatly improve the clarity and readability of the paper.

**Details Of Ethics Concerns:**

Inappropriate personal language. The phrase 'Prior to the presentation of my strategy' in line 188 is too informal for ICLR. Additionally, the use of 'my' is inappropriate in multi-author papers (as 'Anonymous authors' implies more than one author). The use of 'my' can create a biased impression, suggesting sole ownership of the idea, which is neither polite nor accurate in the context of collaborative research. The research is the result of joint effort, and language should reflect that.

---

### Official Review · Reviewer_6bB3 · 2024-10-29

**Soundness:** 2
**Presentation:** 2
**Contribution:** 2
**Rating:** 1
**Confidence:** 4

**Summary:**

This paper proposes two algorithms, FedABNLACA and FedBACA, aimed at reducing communication and computational costs through two novel strategies: New Lazy Aggregation (NLA) and Accelerated Aggregation (AA). The authors establish the convergence of these algorithms and analyze the impact of data heterogeneity and local update steps on convergence speed under both full and partial participation settings. Experiments are conducted to verify the communication cost reduction achieved by these algorithms.

**Strengths:**

1. The authors propose two novel algorithms to reduce communication costs.
2. They provide a convergence analysis for these algorithms, with complete proofs included in the appendix.
3. Experiments are conducted to compare these algorithms with others, demonstrating that their methods effectively reduce communication costs.

**Weaknesses:**

1. The writing lacks a clear logical flow, and terminology is inconsistent, especially in the introduction. For instance, the statement “In FL, traditional distributed stochastic gradient descent (SGD) is unsuitable” does not explain why it is unsuitable, and “To address this issue, many federated optimization methods use local client updates” does not specify which issue it aims to address. The terms "edge client," "device," "local client," and "worker" are used inconsistently, creating confusion.
2. The authors list three current methods for reducing communication costs but do not explain why they chose lazy aggregation as a baseline, particularly since they acknowledge its practical challenges.
3. The paper claims to reduce communication costs in both directions but does not clarify why downlink reduction is necessary, given that many prior works assume downlink communication is fast and negligible.
4. Both algorithms claim to reduce communication costs, but it is unclear why they are presented together in a single paper rather than separately.
5. There is redundant content in the “Main Contributions” section.
6. In Assumption 3.1, it should be "a constant L" rather than “an L.” The authors do not provide explanations for equations and examples in 3.1.
7. The authors state that larger local steps can accelerate convergence with full participation but lead to slower convergence with partial participation; however, they do not explain why this difference occurs.
8. The authors claim in the abstract that their algorithms reduce computation costs, yet no explanation is provided on how this reduction is achieved. The authors do not explain how their algorithms specifically reduce downlink communication costs, also comparing with other algorithms.
9. All figures in the experimental section are incomplete. For example, Figure 1,2,3 and 4 contain only three and six lines, whereas the compared algorithms are total four and six, respectively.

**Questions:**

Same as weaknesses. Please explain all questions above.

---

### Note · Authors · 2024-11-13

**Comment:**

Dear Editor,

We have decided to withdraw this article. Thank you!

**Withdrawal Confirmation:**

I have read and agree with the venue's withdrawal policy on behalf of myself and my co-authors.